

# Mitigation of satellite OCO-2 $CO_2$ biases in the vicinity of clouds with 3D calculations using the Education and Research 3D Radiative Transfer Toolbox (EaR$^3$T)

**Yu-Wen Chen[a,b], K. Sebastian Schmidt[a,b], Hong Chen[b], Steven T. Massie[b], Susan S. Kulawik[c], and Hironobu Iwabuchi[d]**

[a] *Department of Atmospheric and Oceanic Science, University of Colorado Boulder, Boulder, CO, US*

[b] *Laboratory for Atmospheric and Space Physics, University of Colorado, Boulder, CO, US*

[c] *Bay Area Environmental Research Institute, Earth Science Division, NASA Ames Research Center, Moffett Field, CA, US*

[d] *Center for Atmospheric and Oceanic Studies, Graduate School of Science, Tohoku University, Sendai, Miyagi, Japan*

Correspondence: Yu-Wen Chen (Yu-Wen.Chen@colorado.edu) and K. Sebastian Schmidt (Sebastian.Schmidt@lasp.colorado.edu)

**Abstract.**

Accurate and continuous measurements of atmospheric carbon dioxide ($CO_2$) are essential for climate change research and monitoring of emission reduction efforts. NASA's Orbiting Carbon Observatory (OCO-2/3) satellites have been deployed to measure the column-averaged $CO_2$ dry air mixing ratio ($X_{CO2}$) with very high precision. Although cloudy measurements are screened out, nearby clouds can still cause retrieval biases because the forward one-dimensional (1D) radiative transfer (RT) model used in the OCO retrieval algorithm does not account for the scattering induced by clouds in the vicinity of the OCO-2/3 footprints. These biases, referred to as the three-dimensional (3D) effects, can be quantified effectively using 3D-RT calculations, but these are computationally expensive, especially for hyperspectral applications (e.g., OCO-2/3). To reduce the prohibitive computational demands of 3D-RT radiance simulations across all three OCO spectral bands, this paper employs a linear approximation with two metrics (called slope and intercept) for each of the OCO bands that represent the 3D-RT perturbations on the OCO-2 spectra and accelerate the radiative transfer by a factor of 100. This is implemented by the Education and Research 3D Radiation Transfer Toolbox for OCO (EaR$^3$T-OCO). EaR$^3$T-OCO estimates OCO-2 satellite radiances using all available footprint-level data and imagery from the Aqua satellite, which orbits in close proximity to the OCO-2 satellite. EaR$^3$T-OCO can calculate 3D-RT spectral perturbations for any OCO-2 footprint. These calculations can be used to spectrally adjust the OCO-2 radiance measurements with scene-dependent EaR$^3$T-OCO perturbation calculations prior to the actual retrieval to undo cloud vicinity effects in the radiance spectra, which can subsequently be processed with the standard OCO-2 retrieval code. We find that this adjustment largely mitigates $X_{CO2}$ retrieval biases in proximity to clouds over land – the first physics-based correction of 3D-RT effects on OCO-2/3 retrievals. Although the accelerated 3D-RT radiance adjustment step is faster than full 3D-RT calculations for all OCO spectral bands, it still requires at least as much computational effort as the $X_{CO2}$ retrieval itself. To bypass 3D-RT altogether, the slope and intercept metrics are parameterized as a function of the weighted cloud distance of a footprint and several other scene parameters, all of which can be derived directly from Aqua-MODIS imagery. While this method is



fastest and thus feasible for operational use, it requires careful validation for various surface and atmospheric conditions. For the case we analyzed, both the 3D-RT calculation method and the parametric bypass method successfully corrected $X_{CO_2}$ biases, which exceeded 2 ppm at the footprint level, and reached up to 0.7 ppm in the regional average. We find that the biases depend most strongly on the cloud field morphology and surface reflectance, but also on secondary factors such as aerosol layers and sun-sensor geometry.

## 1.    Introduction

Precise global carbon dioxide ($CO_2$) measurements are becoming increasingly important as climate change intensifies. They are necessary to gain a deeper understanding of surface $CO_2$ sources and sinks and their response to climate change, emissions reductions, and other mitigation strategies. Reducing $CO_2$ emissions is imperative for slowing down the pace of climate change. The Greenhouse Gases Observing Satellites (GOSAT, GOSAT-2, Nakajima et al., 2010; Imasu et al., 2023) and the Orbiting Carbon Observatory (OCO-2, OCO-3, Chris, 2015; Eldering et al., 2019), launched by the Japan Aerospace Exploration Agency (JAXA) and NASA, respectively, are currently in space to observe $CO_2$ and other greenhouse gases. They have been designed to precisely measure $CO_2$ column dry air mixing ratios ($X_{CO_2}$) through the analysis of reflected solar radiances in the oxygen A-band at 765 nm ($O_2$-A), as well as the weak and strong $CO_2$ bands near 1.61 µm ($WCO_2$) and 2.06 µm ($SCO_2$).

For remote sensing measurements of $X_{CO_2}$ to effectively contribute to carbon flux (sources and sinks) studies, high accuracy is imperative. Miller et al. (2007) suggest that the regional uncertainty should be within 0.3-0.5% (1 to 2 ppm) to meaningfully contribute to carbon flux estimates. Deng et al. (2016) and Crowell et al. (2018) also emphasize the significance of the level of $X_{CO_2}$ accuracy for reliable $CO_2$ flux determination. Precision and accuracy in $CO_2$ remote sensing are contingent on factors including spectroscopy, calibration, aerosol scattering and absorption, and atmospheric water vapor (Nelson et al., 2022; Worden et al., 2017; Connor et al., 2016). The OCO missions employ an $X_{CO_2}$ retrieval algorithm that integrates these elements along with observational conditions, such as solar zenith angle (SZA), viewing zenith angle, and geolocation (OCO-2 L2 ATBD, 2020) and uses a priori data such as $CO_2$ vertical profiles and surface reflectance to initialize spectral calculations via a one-dimensional (1D) radiative transfer (RT) model. The retrieval process iteratively refines the initial a priori estimates through optimal estimation methods (Rogers, 2013) until convergence between calculated and observed spectra is achieved.

While the 1D-RT model facilitates efficient computation, it neglects lateral photon transfer between atmospheric columns as a trade-off. It has been observed that the three-dimensional (3D) cloud bias, which stems from cloud scattering, continues to affect the accuracy of trace gas retrievals. Recent studies (Massie et al., 2017, 2021, 2023; Kylling et al., 2022) have highlighted the presence of 3D cloud bias in trace gas retrievals, including OCO and TROPOspheric Monitoring Instrument (TROPOMI) nitrogen dioxide ($NO_2$) retrievals. The cloud-related bias is also evident when examining individual footprints. With clouds covering roughly 70% of the globe (Wylie et al., 2005; King et al., 2013) and 40% of the OCO-2 measurements within 4 km of clouds (Massie et al., 2021), addressing cloud-induced bias is crucial for refining $X_{CO_2}$ retrieval accuracy.

Schmidt et al. (2024) explain that lateral photon transport can be understood as missing physics in the operational OCO algorithm, and any adjustments for discrepancies between 1D-RT and 3D-RT could introduce additional inaccuracies in $X_{CO_2}$ retrieval. Although advances have been made in expediting high-



resolution 3D-RT simulations by using the same photon paths for various wavelengths (Emde et al., 2011; Iwabuchi and Okamura, 2017), the computational demands of such models have still hindered their operational application. Schmidt et al. (2024) introduced the 3D-RT radiance perturbation as the percentage difference between the 3D and 1D radiance simulations. This radiance perturbation is found to be linear over the relevant dynamic range of reflectance, which allows a simple representation of the perturbation as slope and intercept for each of the three OCO-2 bands. The details will be described in Section 2.

Although the physical mechanism of the $X_{CO2}$ 3D cloud retrieval bias is now largely understood, practical strategies for applying these insights to a bias correction have not been developed thus far. Mauceri et al. (2023) employed machine learning techniques to correct for 3D cloud biases using observations from the Total Carbon Column Observing Network (TCCON). This correction based on machine learning offers a dynamic means of addressing both linear and non-linear cloud-induced biases.

In this paper, we introduce the direct application of the scene-dependent slope and intercept parameters to the correction of 3D-RT biases, using a modified version of the Education and Research 3D Radiative Transfer Toolbox (EaR[3]T; Chen et al., 2023), tailored specifically for OCO (EaR[3]T-OCO). This tool simulates the radiance for OCO-2 footprints, using, among other data (Section 3), imagery from the MODIS on the Aqua satellite, which is approximately 6 minutes behind OCO-2 within the NASA A-Train (afternoon) satellite constellation. From these, the slope and intercept parameters for the OCO-2 footprints of a given scene are derived, then used to undo the 3D-RT perturbation in the observed radiance spectra, and subsequently in the $X_{CO2}$ retrieval. The spectral dimensionality (3x1024 for the three OCO-2 channels), and thus computational effort, are thereby greatly reduced because our methodology (Section 4) only requires a few selected wavelengths. From our results for a few scenes in different regions of the world, we develop a parameterization of slope and intercept as a function of effective cloud distance and other scene variables (Section 5). We then show that the correction of 3D-RT biases in the spectroscopy and $X_{CO2}$ retrievals works both on the footprint-by-footprint basis, and by way of the new parameterization. This parameterization not only enhances our physics-based understanding of the $X_{CO2}$ retrieval biases introduced by clouds, but also offers a computationally efficient pathway for applying these insights globally across extensive datasets. Conclusions are drawn in Section 6, and future work is discussed in Section 7. The appendix explains the functionality of EaR[3]T-OCO.

## 2.    Background information

This study builds on Schmidt et al. (2024), which found a linear relationship between radiance perturbations and reflectance due to 3D-RT effects. They define the 3D-RT radiance perturbation as the percent difference between the radiances calculated by 3D and 1D radiative transfer models, as formulated in Eq. (1).

$$Perturbation_\lambda = \frac{I_\lambda^{3D} - I_\lambda^{IPA}}{I_\lambda^{IPA}} \times 100\% = \left(\frac{I_\lambda^{3D}}{I_\lambda^{IPA}} - 1\right) \times 100\% \tag{1}$$

The magnitude of this perturbation is not uniform across the observed wavelength spectrum, but depends on the reflectance ($R_\lambda$), defined as follows:

$$R_\lambda = \frac{I_\lambda \times \pi}{S_0{}_\lambda^{TOA} \times \cos\theta_s} \tag{2}$$




where $S_{0\,\lambda}^{\,TOA}$ in the denominator denotes the solar irradiance at the top of the atmosphere (TOA) for a given
wavelength λ, and $\theta_s$ is the solar zenith angle.

Within the dynamic range of interest for reflectance, the dependence of the perturbation on the

reflectance is linear. This is illustrated in Fig. 1, which shows simulated observations in the $O_2$-A band
(Schmidt et al., 2016). For small reflectance, the scatter increases, which is due to a limited number of
photons in the calculations (Section 4.2.4). A line is fitted to the data to represent the first-order dependence
of the 3D-RT perturbation on the reflectance. This can be done with either all wavelengths (grey dots) or a
subset (blue), which is strategically chosen to encompass the full reflectance range.

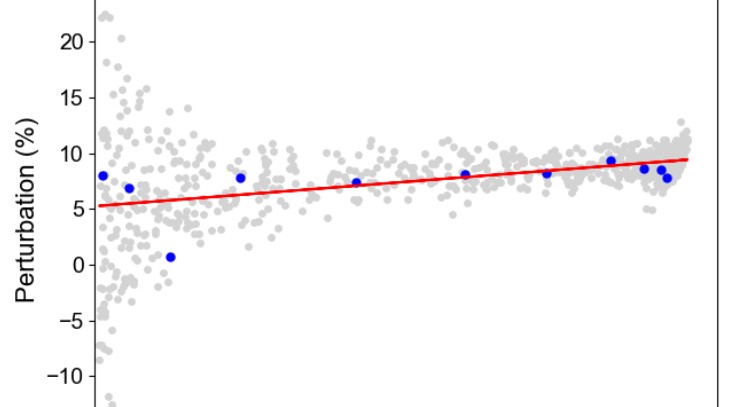

Figure 1. Example of the linear relationship between perturbation and reflectance. The grey dots represent
the complete wavelength range, while the blue dots indicate the subset selected for the $O_2$-A band
simulation.

The slope and intercept parameters, $s_{xy}$ and $i_{xy}$, are obtained through weighted linear regression as shown
in Eq. (3), where the weights are the inverse of the perturbation uncertainty (computational noise, see
Section 4.2.4). The slope and intercept are indicative of distinct physical phenomena: a non-zero slope
corresponds to wavelength-dependent variations and differences in 1D and 3D radiances, photon path
lengths, and absorption. Increased photon path lengths from multiple scattering in 3D-RT produce non-zero
perturbations (percentage differences in 1D and 3D radiances) expressed in Eq. (1). Since wavelengths with
higher absorption are attenuated more than those with lower absorption, the Eq. (1) perturbations are a
function of reflectance (line absorption depth), referred to later as spectral distortion. The intercept is related
to the often-reported increase of reflectance near clouds, or decrease in shadows, whereas the slope accounts
for spectroscopic effects.

$Perturbation_\lambda = i_{xy} + s_{xy} \times R^{3D}_\lambda \quad (3)$





## 3.  DATA

### 3.1.  OCO-2 data

Version 10r OCO-2 data, hosted by NASA's Goddard Earth Science Data and Information Services Center (GES DISC) data archive (https://oco2.gesdisc.eosdis.nasa.gov/data/OCO2_DATA), are used in this research. The Level 1B calibrated and geolocated science radiance spectra (L1bScND) are specified in all three OCO-2 bands, facilitating simulation comparison and retrieval adjustment. Additionally, solar zenith and azimuth angles, as well as viewing zenith and azimuth angles from this product, serve as inputs for the simulation. The standard Level 2 geolocated $X_{CO2}$ retrieval results (L2StdND) provide the retrieved $CO_2$ dry air mixing ratio and surface reflectance information for the three bands. Note that the $X_{CO2}$ in L2StdND files is raw $X_{CO2}$ before the bias correction provided by the algorithm team. We also employed Level 2 meteorological parameters interpolated from the global assimilation model for each sounding (L2MetND) and Level 2 $CO_2$ prior based on the $CO_2$ monthly flask record, global meteorology, and age of air (L2CO2Prior) to construct the atmosphere for the simulation (refer to Section 4.1.1.1).

### 3.2.  MODIS Aqua data

The MODIS Aqua satellite, launched in May 2002, is part of NASA's A-Train constellation but is exiting the formation due to fuel issues. As MODIS Aqua shared the same orbit as OCO-2 and arrived at the same scene approximately six minutes after OCO-2, the collocated information from MODIS Aqua offers valuable insights into the meteorological and surface conditions of the OCO-2 footprints and the spatial distances of clouds to the footprints. Several products derived from MODIS Aqua observations are used in this study, including MODIS level 1B radiance products at the quarter, half, and one-kilometer scales (channels 1 to 7, MYD02QKM, MYD02HKM, and MYD021KM, MODIS Characterization Support Team (MCST), 2017a-c), the geolocation product (MYD03, MODIS Characterization Support Team (MCST), 2017d), the level 2 cloud product (MYD06, Platnick et al., 2015), the level 2 aerosol product (MYD04_L2, Levy et al., 2015) and the surface reflectance product (MCD43A3, Schaaf et al., 2021) from data collection 6.1. These various products contribute to a comprehensive understanding of the atmospheric and surface conditions relevant to the OCO-2 measurements, thereby enhancing the accuracy and reliability of our analysis.

## 4.  Methods

### 4.1.  Case Description

In order to investigate the $X_{CO2}$ retrieval biases resulting from cloud scattering, we have selected a case that features high $X_{CO2}$ anomalies in close proximity to clouds, as shown in Fig. 2. The chosen case is located in Central Asia, spanning from 33.85° N, 55.15° E to 34.30° N, 55.45° E. The study focuses on the conditions observed on October 18th, 2018, which have stronger reflectance for all three OCO-2 bands. The average surface level is about 758 m based on the MODIS MYD03 file (see Fig. A1), while the OCO-2 Met file specifies an altitude near 790 m. The average solar zenith angle and observation zenith angle for OCO-2 footprints are 48.5° and 0.31°, and the mean surface albedo for $O_2$-A, $WCO_2$, and $SCO_2$ bands are 0.288, 0.375, and 0.370, respectively.  The average aerosol optical depth (AOD) at 550 nm from the MODIS





MYD04 file is 0.179 over the domain. The appendix specifies the surface altitude level from the MODIS
MYD03 data file, plus the atmosphere profile for this case, derived by the method described in Section
4.2.1.1.

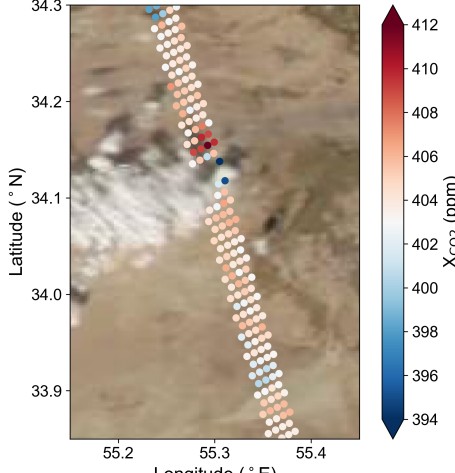

Figure 2. Satellite true-color imagery of MODIS Aqua from NASA Worldview on 18 October 2018, with
OCO-2 retrieved $X_{CO_2}$ overlaid.

**4.2.    Radiative transfer model simulation**


The RT model simulates the photon-environment interactions based on the understood physical
mechanisms, such as absorption, scattering, and reflection. Since this research aims to investigate the
differences between 1D and 3D radiances, both 1D and 3D RT calculations are utilized. The atmospheric
environment used for the RT simulation also significantly impacts the results. This subsection outlines the
details of the relevant model setup.

4.2.1.    Vertical atmospheric structure

The atmosphere profile for the simulation is constructed based on the OCO-2 Met and $CO_2$ prior data, and
it is vertically divided into 29 layers. The surface altitude and pressure are determined by the average surface
height of the footprints analyzed. The heights of other layers are then linearly interpolated from the surface
to 5 km for 11 points, with 0.5 km intervals from 5 km to 10 km, 1 km intervals from 10 km to 14 km, and
5 km intervals from 20 km to 40 km, which was the top height of the simulation. The pressure profile
corresponding to these heights is calculated using a method similar to the MERRA-2 reanalysis product
(Bosilovich et al., 2015) from NASA's Global Modeling Assimilation Office (GMAO). The temperature
and horizontal wind profiles are then retrieved from the OCO-2 Met data by linear interpolation between
pressure and temperature/wind.
Accurate number densities of $O_2$, $CO_2$, and water vapor are crucial for calculating the absorption
coefficients. We assumed that the atmosphere followed the ideal gas law to calculate the number density of



each layer. $O_2$ number density is determined by multiplying its dry-air mixing ratio (0.20935, OCO-2 L2
ATBD) with the dry-air number density. $CO_2$ number density is calculated similarly but using the $CO_2$ prior
profile. We use the specific humidity in the OCO-2 Met data as well as the temperature and pressure to
derive the water vapor volume mixing ratio (VMR) profile. $H_2O$ VMR is essential to obtain more accurate
absorption coefficients to consider the water vapor broadening, which is discussed further in Session 3.2.2.
The $H_2O$ VMR profile is also converted into number density for further absorption calculation. The derived
profiles are displayed in Fig. A2. These steps ensure that the atmospheric parameters used in the simulation
are as accurate as possible.
4.2.2.    Absorption Coefficients
For missions with high spectral resolution, such as GOSAT and OCO, accurate absorption coefficients
within the observed range are indispensable for meticulously modeling the absorption process. This study
utilizes the same precalculated lookup tables of absorption coefficients (ABSCO tables) employed in
version 10 of the OCO retrieval algorithm (ABSCO V5.1, Payne et al., 2020). These ABSCO tables furnish
line-by-line absorption cross-sections for $O_2$, $CO_2$, and $H_2O$ within the observed wavelength range. They
also account for the line mixing, speed dependence of molecular collisions, and collision-induced
absorption (OCO L2 ATBD, 2021). Due to the design of OCO instruments and the varying viewing angles
of the eight footprints within the same swath, the wavelengths of each band exhibit slight discrepancies
(OCO L1B ATBD, 2021). To mitigate excessive computational demands, we opt to use solely the
wavelengths of the first footprint.
The ABSCO tables are functions of the pressure, temperature, and water $H_2O$ VMR. Because the
grid points of pressure, temperature, and $H_2O$ VMR are discrete, we calculate the absorption coefficients
by applying trilinear interpolation to approximate the cross-section of each line. The instrument line shape
provided in the OCO L1B file was used to weigh various lines when calculating the absorption coefficient
for each wavelength. With the molecule number densities of $O_2$, $CO_2$, and water vapor established during
step 3.2.1, we compute the absorption coefficients in $km^{-1}$ for $O_2$-A, $WCO_2$, and $SCO_2$ for lines whose
relative cross-section exceeded 0.05 within the instrument lineshape range. The clear-sky transmittance for
each wavelength can be calculated using the derived absorption coefficients.
4.2.3.    Cloud detection and properties
MODIS products provide cloud mask information with a cloud identification accuracy of about 90% over
land between 60°N and 60°S (Frey et al., 2020). However, undetected clouds can lead to a significant
radiance inconsistency in RT simulation for a small footprint. To address this, we detect clouds based on
the reflectance difference between the observation and white-sky surface albedo provided by the MODIS
43 product. We use various reflectance thresholds for different cases to ensure that most clouds are detected.
This cloud detection approach is distinct from the detection method used in Chen et al. (2023) and designed
for this study specifically.
Once the cloudy pixels are identified, we retrieve the cloud top height (CTH) and cloud effective
radius (CER) of the nearest location from the MODIS MYD02 cloud file and assign them to each cloudy
grid point. To determine the cloud optical thickness (COT) of each pixel, we run the RT model over several
COT and derive the COT-radiance relationship by ourselves to ensure the radiance consistency in 1D-RT





simulation. The COT of each pixel is then determined by applying the COT-radiance relationship. To adjust
for the projection and observation time difference between OCO-2 and MODIS Aqua, we apply a cloud
position adjustment as described by Chen et al. (2023). The adjustment includes a geometry parallax shift
of the cloud position due to the time shift and the wind speed and direction, as determined by the CTH.
4.2.4.    RT model and tools
In this research, a modified version of the Education and Research 3D Radiative Transfer Toolbox v0.1.1
(Chen et al., 2023) for OCO (EaR$^3$T-OCO) is utilized to model the 1D and 3D radiances of the scene. The
Monte Carlo Atmospheric Radiative Transfer Simulator version 0.10.4 (MCARaTS, Iwabuchi, 2006)
serves as the core engine for this simulator, which automatically ingests satellite products and simulates 1D
and 3D spectral radiances. MCARaTS iteratively traces the path of each photon and calculates the
distribution of photons based on the final probability. Chen et al. (2023) demonstrate the ability of EaR$^3$T
to simulate the radiance observed by OCO-2. We utilize the framework and example application 1 outlined
in Chen et al. (2023) to develop a specialized version of the application, which is described in Appendix B.
We improve the atmospheric structure based on the OCO-2 level 2 products and the absorption coefficient
derivation method, as described in Sections 4.2.1 and 4.2.2. For the simulation of each wavelength, $1 \times 10^9$
photons are used and distributed to various absorption lines for a single run. The mean radiance and the
standard deviation are then calculated from three runs to estimate the uncertainty.
**4.3.    Perturbations, Reflectance, Slopes, and Intercept Derivation**
Building on the foundational concepts of perturbation and reflectance introduced in Section 2 (refer to Eq.
1 and Eq. 2), we run the EaR$^3$T-OCO simulator in 1D and 3D mode to calculate $I_\lambda^{IPA}$ and $I_\lambda^{3D}$. From these
simulated radiances, we obtain the reflectances and perturbations. These are used to derive the slope and
intercept parameters that are used for quantifying the 3D effect. We apply a weighted linear regression (see
Eq. 3) to ensure that more accurate data exerts a greater influence on the parameter estimation. This
approach yields not only the values of the slope and intercept but also their respective uncertainties,
providing a comprehensive picture of the 3D effect's variability and reliability. The obtained slope/intercept
parameters are used for quantifying the magnitude of the 3D effect, a detailed discussion of which is
presented in Sections 5.2 to 5.4. Additionally, the parameters play a crucial role in the offline mitigation
strategies explored in Section 5.5.
**4.4.    OCO retrieval algorithm and spectra mitigation**
The retrieval algorithm plays a vital role in determining $X_{CO2}$ based on the radiances of the three bands.
Notably, the retrieval algorithm accounts for various processes, and post-retrieval processing calculates
linear bias corrections. Different retrieval versions may yield diverse outcomes even with identical inputs.
In this study, we utilize the corresponding OCO retrieval algorithm version B10.04 to compare with version
10r    $X_{CO2}$.    The    retrieval    code    is    publicly    available    on    NASA's    GitHub    repository
(https://github.com/nasa/RtRetrievalFramework).



Given that the 1D-RT model does not account for additional scattered photons, the mitigation
strategy proposed in this study involves modifying the observed spectra to eliminate radiance changes
induced by the 3D effect. This adjustment process is referred to as "radiance adjustment" and is derived
from Eq. (1-3). Upon deriving the 3D parameters in Section 4.3, we can convert the OCO-2 spectra using
Eq. (4) with the observed radiance spectra and corresponding reflectance, slope, and intercept. Assuming
the absence of 3D effects in the adjusted 1D radiance, we can employ the B10.04 retrieval algorithm with
un-perturbed spectra to obtain mitigated $X_{CO2}$.

$$I_\lambda^{IPA(adjusted)}(x,y) = \frac{I_\lambda^{IPA(obs)}(x,y)}{\left(\dfrac{i_{xy}+s_{xy}\times R_\lambda^{obs}}{100\%}+1\right)}$$

(4)

## 5.   RESULTS

### 5.1.   3D-RT simulation radiance closure

In order to derive the slope (*s*) and intercept (*i*) parameters that accurately represent the 3D cloud effect in
the OCO-2 observations, it is crucial to perform realistic radiance simulations near the satellite's footprint.
Chen et al. (2023) show that the extent of radiance closure (the percent difference in the forward radiance
model and observed radiances) can indicate the correctness of the retrieved cloud properties. Fig. 3a-b
presents the 3D-RT simulation and MODIS observation of 650 nm using the COT, CER, and CTH shown
in Fig. A3. The heat map in Fig. 3c shows a good agreement between the simulation and observation. As a
result, we are confident that the simulation at other wavelengths is able to approach the actual condition.
We then used the same COT, CER, and COT settings to model the wavelengths of $O_2$-A, $WCO_2$, and $SCO_2$
bands.

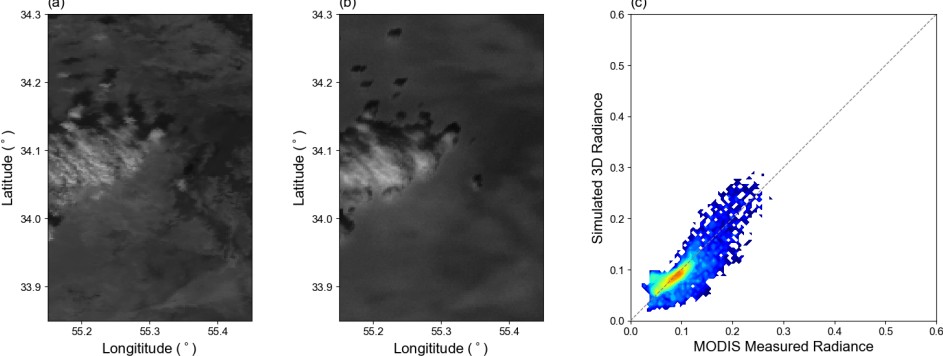

Figure 3. MODIS observation at 650 nm (a) and 3D radiance simulation at 650 nm by EaR$^3$T (b). A heatmap
comparison between (a) and (b) is depicted in (c).

Multiple reflectances or wavelengths are needed to derive *s* and *i* for the linear expression of the
perturbations and reflectances of three bands. To balance computational demands with accuracy, we





selected 11 wavelengths evenly distributed over the high 60% transmittance based on sorted clear-sky transmittance for further RT simulation (depicted in Fig. 4 as an example, the transmittance of full spectra is presented in Fig. A4). The transmittance is calculated based on the atmosphere profile, $\theta_s$, etc. This reduction in simulated wavelengths is feasible due to the linear relationship between perturbations and reflectance. Employing a reduced number of wavelengths, uniformly distributed across the reflectance space, effectively minimizes computational demands while still permitting the derivation of $s$ and $i$ for the linear relationships within each band. Note that the number of wavelengths (11) utilized for determining the 3D parameters is adaptable. Employing additional wavelengths could decrease the uncertainty of the derived $s$ and $i$, albeit at the expense of increased computation time.

Since the radiance simulation is cyclic at the edges, leaking radiance from one edge of the simulation to the other edge could introduce unrealistic artifacts from RT simulations. We extended the margin by 0.15° on each side but excluded the additional margins during the analysis. Fig. A5 illustrates the simulation and the analysis domains with cloud distribution and cloud distance serving as the background.

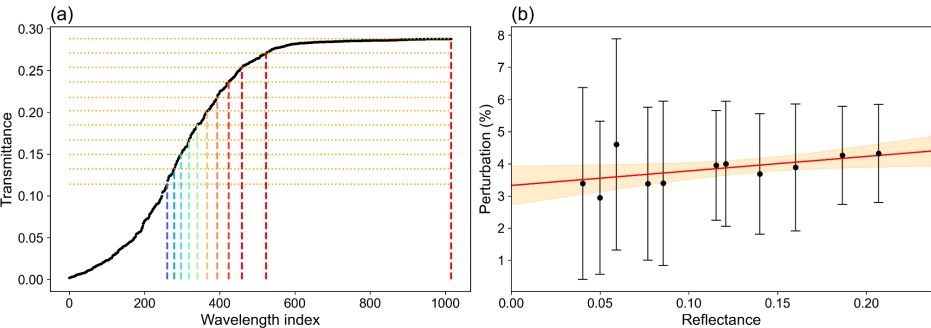

Figure 4. (a) Sorted clear-sky transmittance as per Section 4.2.2, the wavelength index presents the lowest to highest transmittance. (b) Illustration of the selected wavelength distribution in reflectance space versus the Eq. (3) perturbation for 34.08° N, 55.31° E.

## 5.2. 3D cloud effect parameters analysis

We employed simulated radiance data across three distinct bands to quantify the 3D cloud effect perturbation: $s$ and $i$. With the horizontal grid cell size being around 0.25 km, we derived an approximate mean radiance for a 1 km$^2$ area by calculating the average radiance of the 5×5 nearest grid points, thereby approximating the OCO-2 footprint. We excluded the data if the 25 nearest grid points contained cloud pixels used in the RT simulation. The distribution of $s$ and $i$ for the O$_2$-A band ($s_{O_2-A}, i_{O_2-A}$) is illustrated in Fig. 5, with the cloud positions denoted by red dots.



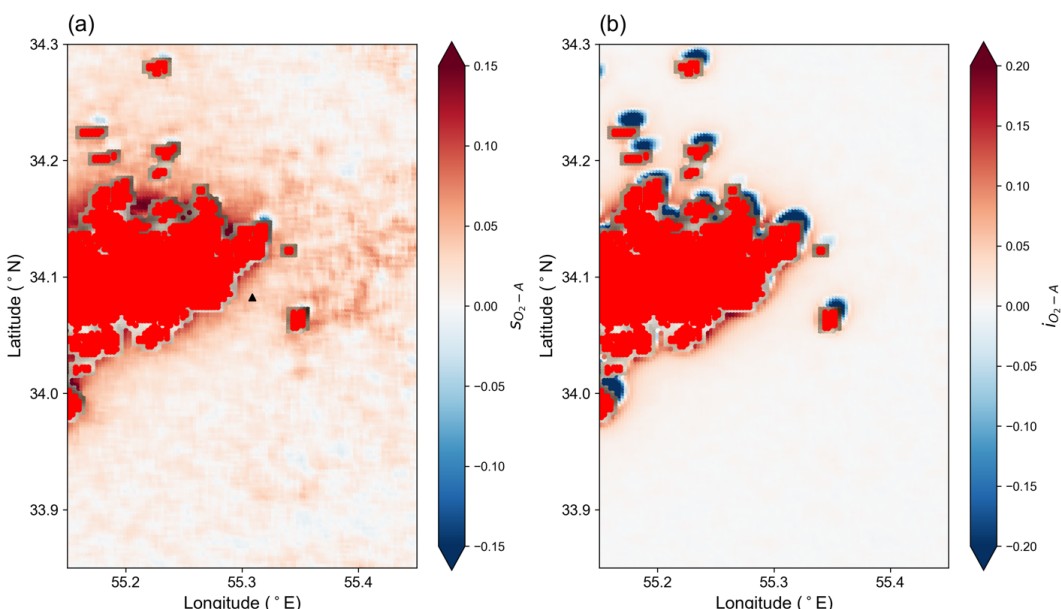

Figure 5. Distribution of (a) $s$ and (b) $i$ of $O_2$-A band. Red dots denote the cloud pixels employed in the RT
simulation.
The analysis shows the magnitudes of the 3D cloud effect parameters diminish ($s$ and $i$ get close to 0) as
the grid point distance from the cloud increases. This decrease in magnitude corresponds to the smaller 3D
cloud effect when the cloud is not in close proximity. Additionally, we divided the clear sky area into two
distinct categories: bright and shadow areas. The bright area represents grid points that receive more
scattered photons, whereas the shadow area encompasses grid points within the cloud shadow region.
Separating these categories is necessary due to the negative and positive intercepts associated with shadow
and bright areas, respectively. Notably, both categories exhibit positive $s$ for the three different bands,
exhibiting characteristics consistent with those described by Schmidt et al. (2024). Though it is instructive
to discuss both cloud brightening and cloud shadowing effects, Massie et al. (2023) determined that
there are relatively few cloud shadow retrievals in the OCO-2 Lite files, since many observations
impacted by shadowing are screened by the OCO-2 pre-retrieval cloud screening algorithms.
Thereafter, bright area analyses are the primary focus of our study.
Upon plotting $s$ and $i$ as a function of various definitions of the distance of a given pixel to the
surrounding clouds, we identified an exponential decay relationship between the 3D cloud effect parameters
and the *effective* horizontal cloud distance ($D_e$, Fig. 6), which is defined as the average distance of the pixel
to the surrounding cloudy pixels, weighted by the inverse square distance to the cloudy pixel (Eq. 5):

$$D_e = \frac{\sum_{i \in \{\text{surrounding clouds}\}} w_i D_i}{\sum_{i \in \{\text{surrounding clouds}\}} w_i} \quad (5)$$



where $D_i$ is the distance of a given pixel to a pixel location of a surrounding cloud, and $w_i = D_i^{-2}$ is the
weight. The exponential drop-off shown in Fig. 6 aligns with the result shown in Fig. 6 of Massie et al.
(2021), although they used the nearest cloud distance.
The effective cloud distance ($D_e$) helps minimize the effect of one small isolated cloud versus multiple
scattered clouds, as displayed in Fig. A5. This exponential decay relationship can be attributed to
atmospheric attenuation proportional to their current values. Subsequently, we fitted these parameters and
$D_e$ using Eq. (6-7);
$$\boldsymbol{s} = a_s \ \times \ \exp(-\frac{D_e}{d_s}) \ (6)$$
$$\boldsymbol{i} = a_i \ \times \ \exp(-\frac{D_e}{d_i}) \ (7)$$
where amplitude ($a_s$, $a_i$) and e-folding distance ($d_s$, $d_i$) are the fitting parameters (separate sets for $\boldsymbol{s}$ and $\boldsymbol{i}$,
the slope ($\boldsymbol{s}$) and intercept ($\boldsymbol{i}$) parameters that represent the 3D cloud effect). The data are partitioned into
multiple columns, employing a bin size of 0.05 reflectance, and we utilize the median of each bin for the
fitting procedure. However, we observed that the median $\boldsymbol{s}$ and $\boldsymbol{i}$ values did not approach zero as the cloud
distance increased, possibly due to an inadequate number of grid points at larger cloud distances. To rectify
this issue, we optimized the fitting coefficients by iteratively increasing the number of points employed in
the fitting process until the maximum $R^2$ value was attained.



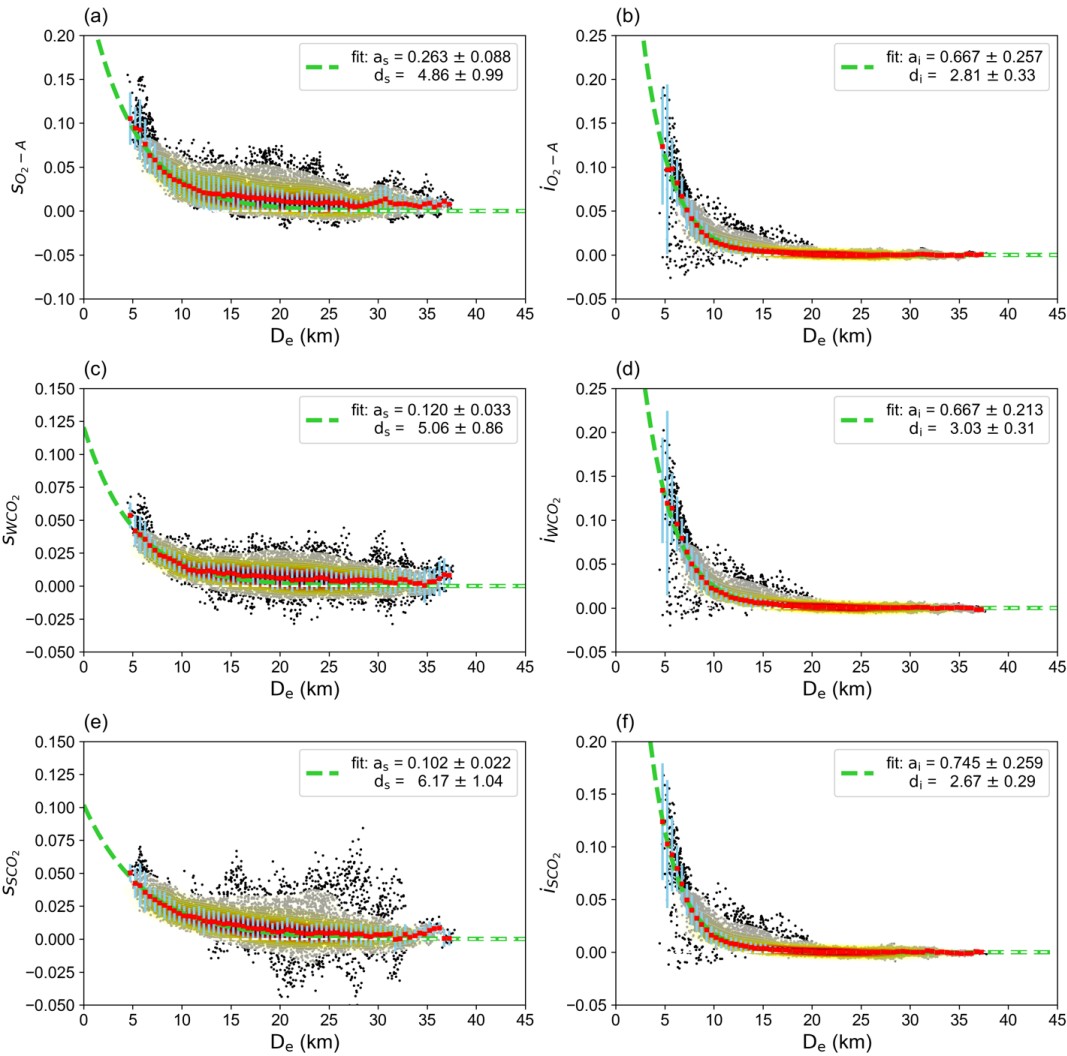

Figure 6. Exponential fitting (green dashed lines) of three bands in the bright area. The black dots present data from each pixel, while the background shading indicates the density of the black dots' distribution. The red points denote the median of each bin, and the blue error bars indicate the first and third quantiles for each bin.

The intercept parameter relates to what is traditionally known as the 3D-RT effect in spectrometry for a wavelength range with minimal gas absorption, whereas the slope is its spectroscopic equivalent, representing spectral distortion due to strongly varying gas absorption cross-sections over a wavelength range. Even slight changes in absorption may result in substantial changes in the retrieved trace gas concentration. The disparity in $d_s$ for each band is not statistically significant, suggesting a similarity in the photon path histories across the different spectral bands. The initial method, denoted the baseline method, uses only about 1% of the available wavelengths, resulting in a substantial reduction in computation time



for 3D-RT calculations. By using these exponential relationships, we can parameterize six 3D effect
parameters, thereby eliminating the need for 3D-RT simulations altogether. This parametric approach will
be denoted as the *bypass* method.
Table 1. Amplitude and e-folding distances for $s$ and $i$ fittings in the $O_2$-A, $WCO_2$, and $SCO_2$ bands.

|  | Slope | | | Intercept | | |
|---|---|---|---|---|---|---|
|  | $s_{O_2-A}$ | $s_{WCO_2}$ | $s_{SCO_2}$ | $i_{O_2-A}$ | $i_{WCO_2}$ | $i_{SCO_2}$ |
| $a_s$ or $a_i$ | $0.263 \pm 0.088$ | $0.120 \pm 0.033$ | $0.102 \pm 0.022$ | $0.667 \pm 0.257$ | $0.667 \pm 0.213$ | $0.745 \pm 0.259$ |
| $d_s$ or $d_i$ (km) | $4.86 \pm 0.99$ | $5.06 \pm 0.86$ | $6.17 \pm 1.04$ | $2.81 \pm 0.33$ | $3.03 \pm 0.31$ | $2.67 \pm 0.29$ |

## 5.3.    Impact of aerosol
Upon establishing the relationship between the 3D cloud effect parameters and the cloud distance, we
proceeded to analyze the impact of aerosols on this phenomenon since aerosols play an important role in
shortwave radiation. In the presence of aerosols, photons near clouds experience increased extinction
(scattering or absorption depending on aerosol radiative properties), making them travel shorter horizontal
distances. Consequently, the e-folding distance is expected to be smaller when there is a higher aerosol
loading. To maintain consistency with the previous section, we kept several variables constant, including
cloud optical thickness (COT), cloud effective radius (CER), cloud top height (CTH), cloud position, and
atmospheric conditions. However, we introduced a homogeneous aerosol layer into the scenario to
investigate its effect on the fitting amplitude and e-folding distance, as detailed in Section 5.2. The aerosol
optical depth (AOD) data were obtained from the MODIS MYD04 data file using the aerosol optical depth
and angstrom exponent at 550 nm. The top height of the aerosol layer was determined by the prevailing
cloud of cloud top heights below 4 km, and we assumed uniform AOD values for layers beneath this top
height. Aerosol optical depths in the $O_2$-A, $WCO_2$, and $SCO_2$ bands are 0.098, 0.038, and 0.024, respectively.
In the simulation incorporating aerosols, analysis was conducted utilizing the methodology
discussed in Section 5.2. Analogous correlations were identified between the 3D cloud parameters and $D_e$,
as shown in Fig. 7 and Table 2. Notably, the presentation of an aerosol layer exhibited a pronounced impact
on the $O_2$-A and $SCO_2$ bands. This observation aligns with the absorption strength of each spectral band.
Consequently, $d_s$ associated with the $O_2$-A and $SCO_2$ bands witness a reduction while $a_s$ of those two bands
increases. This suggests that, in the presence of aerosols, the spectral distortion processes within the strong
absorption bands are predominantly localized in proximity to the cloud. These findings underscore the
pronounced influence of aerosols on the spectral distortion of bands with strong absorptivity.
It's important to note that our simulation relied on the assumption of uniform aerosol distribution
within the boundary layer, derived from the mean AOD obtained from the MODIS product. However, this
assumption may not always hold true in real-world scenarios. We illustrate that the presence of aerosols
can lead to alterations in both the $s$ and $i$ of the $O_2$-A and $SCO_2$ bands, potentially increasing the uncertainty
associated with the derivation of 3D effect parameters.



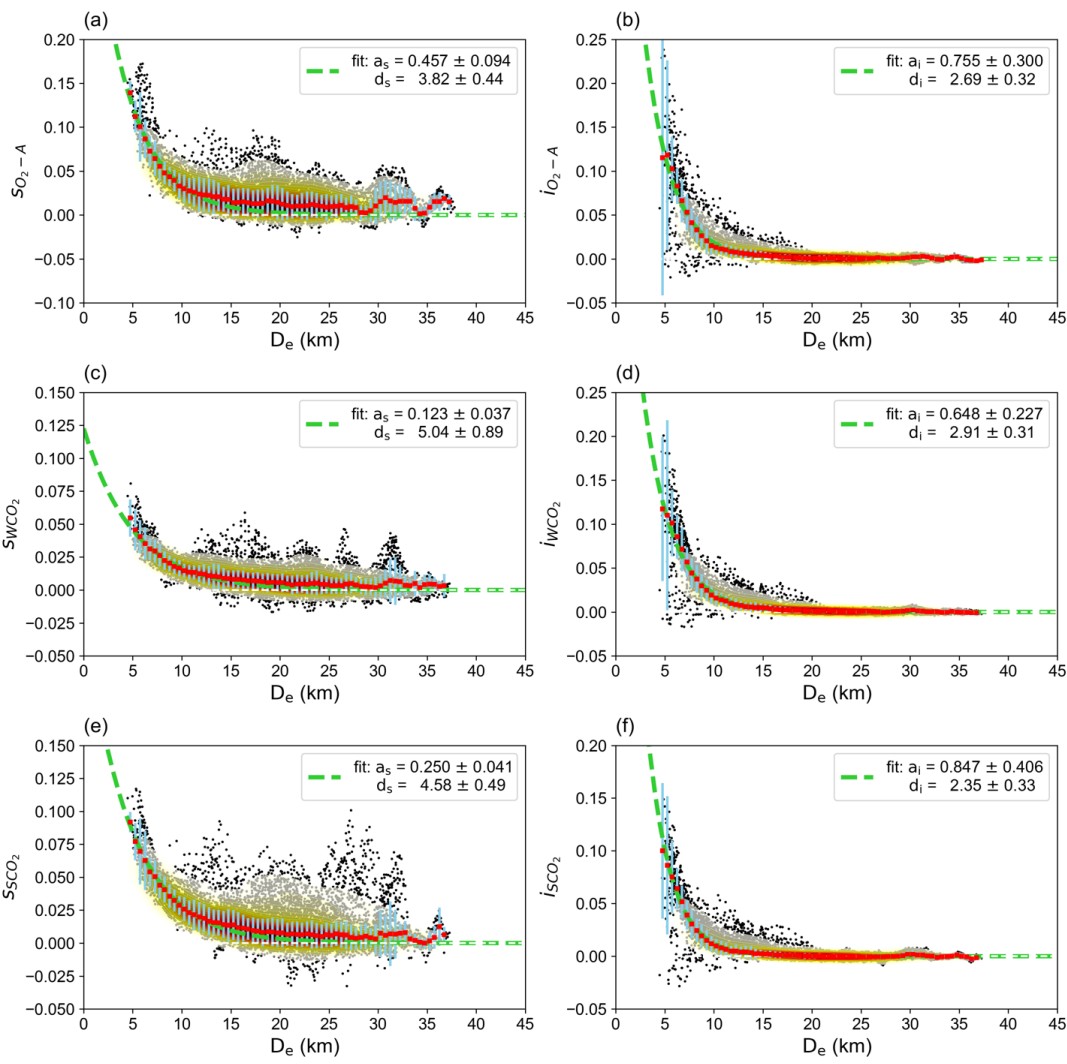

Figure 7. Exponential fitting (green dashed lines) of the three OCO-2 bands in the bright area of the simulation with a homogeneous aerosol layer. The black dots present data from each pixel, while the background shading indicates the density of the black dots' distribution. The red points denote the median of each bin, and the blue error bars indicate the first and third quantiles for each bin.

Table 2. Amplitude and e-folding distances for $s$ and $i$ fittings of the simulation with a homogeneous aerosol layer in the O$_2$-A, WCO$_2$, and SCO$_2$ bands.

| | Slope | | | Intercept | | |
|---|---|---|---|---|---|---|
| | $s_{O_2-A}$ | $s_{WCO_2}$ | $s_{SCO_2}$ | $i_{O_2-A}$ | $i_{WCO_2}$ | $i_{SCO_2}$ |
| $a_s$ or $a_i$ | $0.457 \pm 0.094$ | $0.123 \pm 0.037$ | $0.250 \pm 0.041$ | $0.755 \pm 0.327$ | $0.648 \pm 0.227$ | $0.847 \pm 0.406$ |





| | | | | | | |
|---|---|---|---|---|---|---|
| $d_s$ or $d_i$ (km) | 3.82 ± 0.44 | 5.04 ± 0.89 | 4.58 ± 0.78 | 2.69 ± 0.32 | 2.91 ± 0.31 | 2.35 ± 0.33 |

### 5.4. Impact of Footprint Size

Since the OCO instrument series have a narrow field of view (FOV) of 1.3 km × 2.3 km, compared to 78 km² (10.5 km in diameter) for the GOSAT series, the 3D cloud bias is considered more significant for the OCO retrieval when small footprints are in close proximity to clouds. Numerous upcoming satellites for $CO_2$ remote sensing will adopt similar retrieval algorithms but feature varying footprint sizes in accordance with their specific mission objectives. Thus, exploring the influence of footprint size on 3D effect parameters is vital. We performed an analysis analogous to the one described in Section 5.3 but expanded the average domain from the closest 5×5 grid points (approximately 1×1 km²) to 9×9, and 13×13 grid points (approximately 2×2, and 3×3 km²). Fig. A6 displays the updated distributions of $s$ and $i$. The results of the $s/i$ and cloud distance fitting in the bypass method, presented in Table 3, indicate a decrease in $a_s$ across all three bands. This decline aligns with the expectation that larger footprints would mitigate the spectral distortion effect, reducing the prevalence of pronounced biases. Notably, there is no statistically significant change in $a_i$ and $d_i$ of the intercept.

In addition, $d_s$ exhibit an increase when compared to the smaller footprint size. This implies that even though the baseline radiance change may demonstrate a minor deviation compared to the 1D-RT simulation as the footprint size expands, the perturbation difference in relation to reflectance might persist due to the increasing $d_s$. In conclusion, future satellite missions with any footprint size must account for 3D biases to ensure accurate remote sensing of $X_{CO2}$. For missions utilizing larger footprint sizes to achieve broader global coverage, the 3D cloud effect may be diminished but distributed over a more extensive area. Conversely, missions designed with smaller footprint sizes than OCO-2, particularly those targeting enhanced data acquisition in cloud-prone regions such as the Amazon Basin (Frankenberg et al., 2024) will be susceptible to substantial 3D cloud biases. It is therefore imperative that these missions integrate 3D-RT mitigation strategies (such as the ones we proposed in Section 5.5) from the initial planning stages.

Table 3. Amplitude and e-folding distances for $s$ and $i$, determined using different average grid points in simulations with a homogeneous aerosol layer for the $O_2$-A, $WCO_2$, and $SCO_2$ bands.

| | grid points | Slope | | | Intercept | | |
|---|---|---|---|---|---|---|---|
| | | $s_{O_2-A}$ | $s_{WCO_2}$ | $s_{SCO_2}$ | $i_{O_2-A}$ | $i_{WCO_2}$ | $i_{SCO_2}$ |
| $a_s$ or $a_i$ | 1 x 1 | 0.457 ± 0.094 | 0.123 ± 0.037 | 0.250 ± 0.041 | 0.755 ± 0.327 | 0.648 ± 0.227 | 0.847 ± 0.406 |
| | 2 x 2 | 0.355 ± 0.110 | 0.097 ± 0.025 | 0.217 ± 0.044 | 0.758 ± 0.483 | 0.698 ± 0.360 | 1.138 ± 0.785 |
| | 3 x 3 | 0.180 ± 0.044 | 0.079 ± 0.031 | 0.173 ± 0.058 | 0.971 ± 0.738 | 0.922 ± 0.551 | 1.768 ± 1.548 |
| $d_s$ or $d_i$ (km) | 1 x 1 | 3.82 ± 0.44 | 5.04 ± 0.89 | 4.58 ± 0.78 | 2.69 ± 0.32 | 2.91 ± 0.31 | 2.35 ± 0.33 |
| | 2 x 2 | 4.24 ± 0.68 | 5.82 ± 0.95 | 4.94 ± 0.62 | 2.61 ± 0.45 | 2.78 ± 0.40 | 2.16 ± 0.36 |
| | 3 x 3 | 6.20 ± 1.03 | 6.46 ± 1.57 | 5.46 ± 1.07 | 2.47 ± 0.46 | 2.61 ± 0.40 | 2.00 ± 0.36 |

### 5.5. 3D effect mitigation




Utilizing the derived **s** and **i** for the 3D effect, we can mitigate the 3D effect through the "radiance
adjustment" process, as elaborated in Section 4.4. The assumption is that the 3D effect is removed in the
adjusted 1D radiance, allowing us to retrieve the mitigated $X_{CO2}$ using existing operational retrieval
algorithms. Nevertheless, both the *bypass* method and the *baseline* method can be used to determine the 3D
effect parameters for each footprint. The bypass method uses a generalized parameterization derived from
Eq. (6-7) to conserve computational resources, which eliminates the need for any 3D simulation and could
be feasibly implemented in the operational retrieval algorithm. However, the bypass method is less precise
than conducting a 3D-RT simulation with our baseline approach to derive **s** and **i** on a pixel-by-pixel basis.
This bypass approach also disregards the presence of cloud shadows. We investigate the parameterization
method considering the practicality of integrating this approach with the current operational retrieval
algorithm.
To examine the efficiency of the bypass approach in mitigation, we apply the exponential
relationships between the 3D parameters and cloud distance to footprints in our study case. Initially, the
cloud distance of each footprint is computed based on the cloud position, incorporating parallax and wind
correction (refer to Fig. A5). Subsequently, we determine the **s** and **i** of the three bands using the coefficients
in Table 2 for all footprints that are the best quality (Quality Flag = 0 or 1) data points. The adjusted
spectra are derived accordingly using Eq. (6). Fig. 8 presents an example of the original and corresponding
adjusted spectra of the $O_2$-A band.

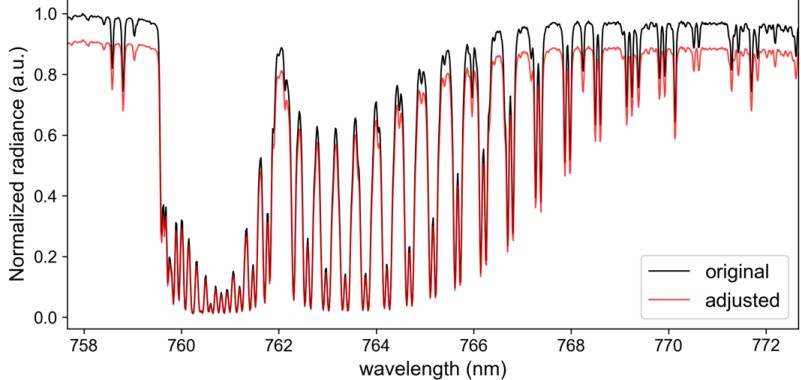

Figure 8. Example of an $O_2$-A spectrum before and after radiance adjustment.
Utilizing the B10.04 retrieval algorithm (refer to Section 4.4), we calculate the newly retrieved $X_{CO2}$ values
for each footprint. Fig. 9 displays the distribution of retrieved $X_{CO2}$ before and after the spectral adjustment
process, superimposed on the collocated MODIS Aqua image. The elevated $X_{CO2}$ near the cloud decreases
after the adjustment approximation, with the newly retrieved $X_{CO2}$ values reduced by approximately 0 to 3
ppm for $D_e$ exceeding 5 km. We perform a similar calculation using **s** and **i** close to the footprint (the pixel-
by-pixel method) and calculate $X_{CO2}$ differences (defined as $\Delta X_{CO2}$, which represents the difference
between the newly retrieved $X_{CO2}$ and level 2 $X_{CO2}$).
Figure 10 compares $\Delta X_{CO2}$ and $D_e$ by using the bypass method and baseline method. The bypass
method calculates **s** and **i** for each grid point based solely on $D_e$, without considering the effects of cloud
shadowing. In contrast, the baseline method derives the 3D parameters from simulation data, accounting
for shadowing effects. Preliminary observations suggest that both methods exhibit analogous $\Delta X_{CO2}$



patterns and offer similar average reductions. Specifically, the bypass method registers an average $\Delta X_{CO2}$
of -0.778 ppm, in contrast to the baseline method which delineates an average $\Delta X_{CO2}$ of -0.876 ppm. Both
methodologies illustrate that pronounced $\Delta X_{CO2}$ usually occurs near clouds and attenuates with an
increasing cloud distance. This trend aligns coherently with the observed distribution of $s$ and $i$. Notably,
the pronounced negative values predominantly arise at shorter cloud distances, an aspect overlooked by the
bypass method. Moreover, the accurate $s$ and $i$ near the cloud edge in the baseline method give a larger
$X_{CO2}$ deduction than the bypass approach.

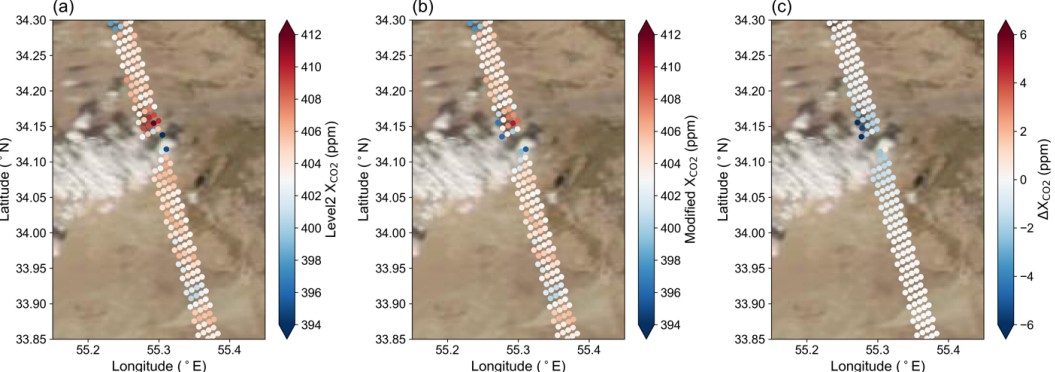

Figure 9. Satellite true-color imagery of MODIS Aqua from NASA Worldview on 18 October 2018 with
(a) $X_{CO2}$ in OCO-2 level 2 data, (b) mitigated $X_{CO2}$ retrieved from the adjusted spectra and (c) difference
between the mitigated and original $X_{CO2}$ values.

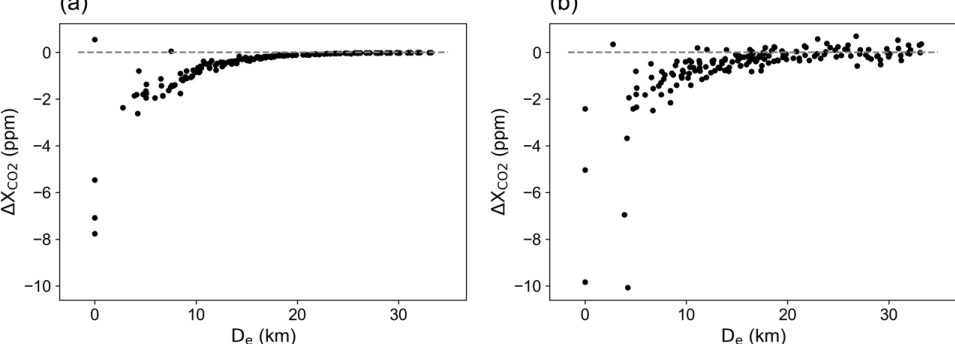

Figure 10. (a) Relationship of $\Delta X_{CO2}$ with $D_e$ as depicted in Fig. 7c, based on parameterized slopes and
intercepts from the bypass method. $\Delta X_{CO2}$ is defined as the difference between the newly retrieved $X_{CO2}$
and level 2 $X_{CO2}$. (b) Corresponding relationship using slopes and intercepts derived from the baseline
approach.





The observed variance of $X_{CO2}$ potentially emanates from the oversimplification of the 3D cloud effect by
the current bypass method, which predominantly draws from brightening regions. Nikolaeva et al. (2005)
classified the 3D cloud effect into four distinct types (brightening, shadowing, photons channeling, and
photons leaking), with only the brightening effect being investigated in this study. The shadow area, which
is not the primary focus of this research, also contributes to $X_{CO2}$ retrieval biases. The existing bypass
method cannot mitigate these biases and may even be exacerbated after the adjustment. Moreover, specific
footprints containing thin clouds that pass the cloud-screening test also exhibit $X_{CO2}$ retrieval biases. Such
footprints, near the cloud periphery or partially encompassing the cloud, might not be optimal candidates
for the parameterization (i.e. bypass) technique.
The requirement of the OCO-2 mission is to determine $X_{CO2}$ uncertainty to within 1 ppm. Setting
the true mixing ratio to the mean $X_{CO2}$ for an effective distance exceeding 15 km, we see that $X_{CO2}$ scatter
is accentuated within 15 km of clouds, as demarcated by the black markers in Fig. 11a. The mitigated $X_{CO2}$
after the spectra adjustment, represented by the red markers in Fig. 11a, exhibits reduced scatter within 15
km of clouds, a fact further corroborated by the full width at half maximum (FWHM) depicted in Fig. 11b.
Cumulatively, the bypass method aligns favorably with the baseline methodology, offering the added
benefit of small computational demands. Concurrently, our proposed mitigation method could integrate
into the current retrieval system, which deploys the 1D-RT model by introducing it as a preprocessing step
and effectively diminishes the impact of the 3D effect at the footprint level for the first time. The next step
in this process is to test this methodology on a larger dataset.
The processing time per footprint for baseline analysis is approximately 7 minutes when using 32
CPUs (AMD EPYC Processor 7713) on a cluster at the University of Colorado to simulate $1 \times 10^9$ photons
for the full experiment domain. This is contrasted with the standard retrieval time of roughly 2.5 minutes
per footprint using 16 CPUs (Intel Xeon Processor E5-2623 v3) on a local workstation. Although the 3D
computation time of 7 minutes marks a significant improvement over full-spectra 3D simulation, the
additional 3D-RT calculations required to account for the missing physics could extend the duration to
nearly 6 times that of standard retrieval on a per-footprint, per-CPU basis. Such a duration is impractical in
operational settings. Therefore, our bypass method offers a pragmatic alternative to mitigating the 3D cloud
effect while conserving computational resources. This method can be supplemented by periodic full
calculations to increase the accuracy of the bypass approach.





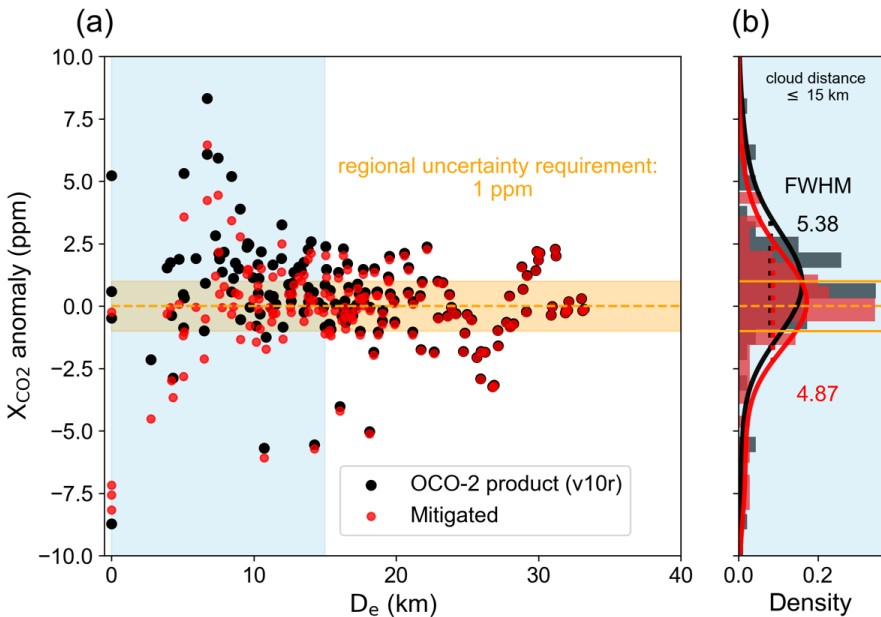

Figure 11. (a) Scatter plot comparing the $X_{CO2}$ anomaly of the OCO-2 L2 product (in black) to its value post-spectra adjustment (in red), plotted against $D_e$. The $X_{CO2}$ anomaly is defined as retrieved $X_{CO2}$ – true $X_{CO2}$, with the true $X_{CO2}$ defined by the average $X_{CO2}$ of footprints with a $D_e$ greater than 15 km (403.714 ppm in this case). The orange shade indicates the 1 ppm mission requirement. (b) Histograms and probability density functions (PDFs) for the $X_{CO2}$ anomaly of the OCO-2 L2 product (in black) and post-spectra adjustment (in red) within a 15 km $D_e$. This corresponds to the blue-shaded region in (a). The FWHM values of the PDFs of v10r and adjusted data points are 5.38 and 4.87, and the PDF averages are 0.796 and -0.178, respectively. The average change in $X_{CO2}$ after the spectra adjustment for De less than 15 km is -1.131 ppm.

With the utilization of the bypass method, we aim to assess how the 3D cloud effect impacts $X_{CO2}$ measurements across various cloud distribution patterns. As illustrated in Fig. 12, we examined four distinct cloud distribution scenarios within the same spatial domain. These patterns allowed us to evaluate the regional biases in $X_{CO2}$ retrievals. Moving from the leftmost panel to the rightmost panel, we observe a sequential reduction in regional mean $D_e$: from 36.86 km to 17.90 km, 15.60 km, and 12.08 km, respectively. Corresponding cloud fractions for these regions are 0.002, 0.004, 0.077, and 0.059.

Importantly, as cloud distances decrease and cloud fractions increase, we note a corresponding escalation in regional $X_{CO2}$ overestimations. Specifically, the regional overestimations transition from 0.096 ± 0.081 ppm to 0.380 ± 0.135 ppm, 0.559 ± 0.246 ppm, and 0.747 ± 0.377 ppm, respectively. This relationship underscores the significance of the 3D cloud effect on $X_{CO2}$ measurements. Notably, regions characterized by more uniformly distributed "popcorn" clouds exhibit a higher regional bias due to their shorter $D_e$. Therefore, our analysis highlights the importance of comprehending the 3D cloud effect, particularly in regions known for their "popcorn" cloud patterns, both over land and ocean.






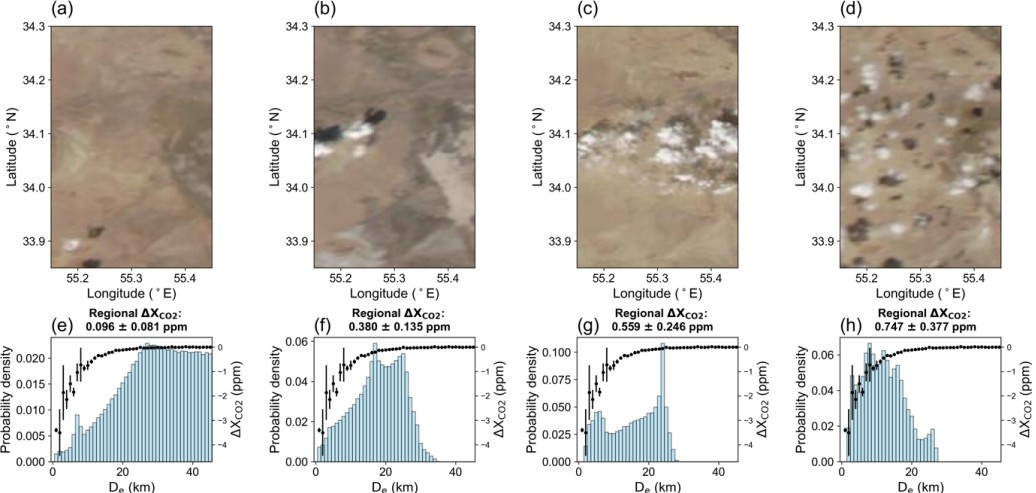

Figure 12. Examination of diverse cloud distributions within the selected region. Upper row (a-d):
Worldview true-color snapshots from MODIS Aqua captured on (a) 14 April 2021, (b) 30 November 2014,
(c) 3 September 2016, and (d) 2 August 2017. Lower row (e-h): Histograms showcasing $D_e$ for the
corresponding dates mentioned above. The error bars depict the $\Delta X_{CO2}$ variations at respective $D_e$. From
the left to the right panel, the regional mean $D_e$ diminishes sequentially from 36.86 km to 17.90 km, 15.60
km, and 12.08 km. Correspondingly, the cloud fractions for these cases are 0.002, 0.004, 0.077, and 0.059.

## 6. Conclusions

We documented the EaR³T-OCO radiance simulator, an automated tool for calculating spectral radiances
observed by NASA's OCO-2 satellite. In contrast to the standard forward model, EaR³T-OCO considers
the scene context of a given footprint. This is a prerequisite to account for the missing physics in the context-
agnostic operational retrieval that stems from clouds in the vicinity of an OCO-2 sounding. We then used
the simulator to undo the effects of such clouds by reversing the perturbations relative to the clear sky that
they exert on the observed radiances. In essence, the observed radiance spectra were mapped back to what
they *would have been* in the *absence* of clouds in the vicinity of a footprint. This radiance mapping is done
based upon the difference between simulated 1D and 3D radiance calculations (3D perturbations). After
this mapping, the standard $X_{CO2}$ retrieval can then be applied. In this way, we introduced a physics-based
mitigation of 3D-RT effects on trace gas spectroscopy products, which has been previously regarded as
intractable for real-world applications such as this.
Since 'brute-force' 3D fully spectral calculations would be computationally prohibitive, we
introduced a physics-based acceleration approach with only a few representative wavelengths, where the
3D-RT spectral perturbations for the three bands are captured by linear fit parameters (slope and offset).
These six parameters are much more cost-effective to simulate than the full spectra, especially while
moving towards operational mitigation of 3D-RT effects. This acceleration method is distinct from
previously published methods, including approaches to "freeze" photon paths for various wavelengths
(Emde et al., 2011, and Iwabuchi and Okamura, 2017). More importantly, it allows the parameterization of





the six spectral perturbation parameters *themselves* as a function of macroscopic scene parameters, which,
if successful, bypasses 3D-RT calculations altogether while retaining the core physics. This approach may
bring 3D-RT mitigation within reach for operational application, not just for OCO-2 and OCO-3 but also
for other spectroscopy missions – even with limited computational resources, so long as co-located imagery
exists to estimate effective cloud distance. In our paper, we tested the imagery-derived effective cloud
distance as the basis for this bypass option and found that it leads to almost the same results in terms of
mitigation quality as the baseline method that does entail 3D calculations.
Further validation of this bypass option is necessary across a larger variety of scenes before it can
be applied in practice. While the bypass method does capture the significant modulators of the 3D cloud
effects, including surface reflectance and sun-sensor geometry, it is not granular enough to consider detailed
scene variables such as cloud top height, cloud morphology, or aerosol load. Also, our study focused on
nadir observations. Further investigation into these factors, especially for target and glint mode, is required.
Meanwhile, using our baseline approach provided by our simulator should work across all scenes since it
considers all the scene-specific details.
Our findings highlight the impact of cloud distributions on regional $CO_2$ assessments and
emphasize the importance of aerosols and footprint size for the accuracy of upcoming $CO_2$ measurement
satellites, such as MicroCarb, CO2M, and GOSAT-GW. In general, our research elucidates the 3D cloud
perturbation on *spectroscopy* with high spectral resolution (trace gas retrievals) as opposed to *spectrometry*
(cloud and aerosol imagery retrievals) where 3D effects are traditionally studied more extensively. We now
understand that 3D effects are best addressed at the radiance level because this is where the physics that is
missing in standard retrievals is operating. We also understand that the effects are spectrally dependent,
with cloud morphology, band-specific surface reflectance, and aerosol properties acting as the primary
drivers. Our work can become the stepping stone toward more accurate and efficient trace gas retrievals
even in complex scenes, ultimately bringing spaceborne trace gas retrievals to a more accurate level of
accuracy. It will improve current flux inversions (especially over the cloud-prone Amazon) and other
applications.

## 7.    Future work

This research emphasizes the substantial impact of aerosols, particularly on the 3D effect parameters of the
$O_2$-A band. Concurrently, the solar zenith angle and surface albedo emerge as critical determinants for these
3D effect parameters. With additional simulations, there is an opportunity to incorporate the impacts of
aerosols, solar zenith angle, and surface albedo into the existing parameterization framework. The current
investigation concentrates on scenarios over land in Nadir mode. We will develop a similar
parameterization (as in Table 3) for ocean and land in Glint mode.

## Code availability

The EaR³T code (Chen et al., 2023) is available at https://github.com/hong-chen/er3t, and the EaR³T-OCO
code is available at https://github.com/ywchen-tw/OCO-2.



## Data availability

Version 10r of OCO-2 data can be accessed at https://oco2.gesdisc.eosdis.nasa.gov/data/OCO2_DATA. The MODIS-related data from data collection 6.1, including MYD02QKM, MYD02HKM, MYD021KM, MYD03, MYD06, MYD04_L2, and MCD43A3, are available at https://ladsweb.modaps.eosdis.nasa.gov/. We acknowledge the use of imagery from the Worldview Snapshots application (https://wvs.earthdata.nasa.gov), part of the Earth Observing System Data and Information System (EOSDIS).

## Author contribution

YWC and SS developed the conceptual framework of the presented methodology. YWC was primarily responsible for carrying out the radiance simulations and analyzing the data. HC played a supporting role in the development of the EaR$^3$T-OCO program. All authors engaged collaboratively in refining the methods, deliberating over the results, and contributed to the drafting of the original manuscript.

## Competing interests

At least one of the (co-)authors is a member of the editorial board of Atmospheric Measurement Techniques.

## Financial support

This research has been greatly benefited from the insights and resources provided by two ROSES projects, "Reducing OCO-2 regional biases through novel 3D cloud, albedo, and meteorology estimation", 20-OCOST20-0031, Susan Kulawik, PI; and "Mitigation of 3D Cloud Radiative Effects in OCO-2 and OCO-3 XCO2 Retrievals," 80NSSC21K1063, Steven Massie, PI. We also extend our appreciation to the National Aeronautics and Space Administration (grant no. 80NSSC18K0146) for developing EaR$^3$T. This work utilized the Alpine high performance computing resource at the University of Colorado Boulder. Alpine is jointly funded by the University of Colorado Boulder, the University of Colorado Anschutz, Colorado State University, and the National Science Foundation (award 2201538).

## Abbreviations

The following abbreviations are used in this manuscript:

| A-train | Earth Observing System Afternoon Constellation |
|---------|-----------------------------------------------|
| ABSCO   | Absorption coefficients                        |
| AOD     | Aerosol optical depth                          |
| CER     | Cloud effective radius                         |
| $CO_2$  | Carbon dioxide                                 |
| COT     | Cloud optical thickness                        |



| CTH | Cloud top height |
|-----|------------------|
| EaR$^3$T | Education and Research 3D Radiative Transfer Toolbox |
| FOV | Field of view |
| FWHM | Full Width at Half Maximum |
| GMAO | Global Modeling Assimilation Office |
| GOSAT | Greenhouse Gases Observing Satellite |
| L (0,1..) | Level 0, Level 1, etc. (data product) |
| MCARaTS | Monte Carlo Atmospheric Radiative Transfer Simulator |
| MODIS | Moderate-Resolution Imaging Spectroradiometer |
| $O_2$-A | Oxygen A-band |
| OCO | Orbiting Carbon Observatory |
| ppm | parts per million |
| $R^2$ | Determination coefficient |
| RT | Radiative transfer |
| $SCO_2$ | Strong $CO_2$ |
| SZA | Solar zenith angle |
| TOA | Top of atmosphere |
| VMR | Volume mixing ratio |
| $WCO_2$ | Weak $CO_2$ |
| $X_{CO2}$ | Column-averaged $CO_2$ dry air mole fraction |

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





# Appendix


## A. Supplementary figures


Appendix A contains supplementary information that complements the details of the simulation setting.
These supplementary elements cover a range of topics, from atmospheric profiles and cloud-related
parameters to radiative transfer simulations.

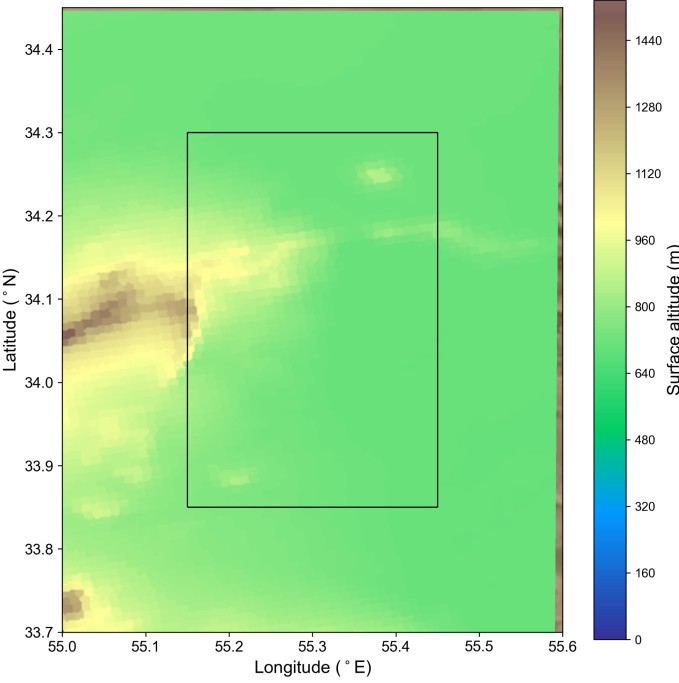

Figure A1. Contour plot showcasing surface height from the MODIS MYD03 file for the outer simulation
domain and the inner analysis domain.





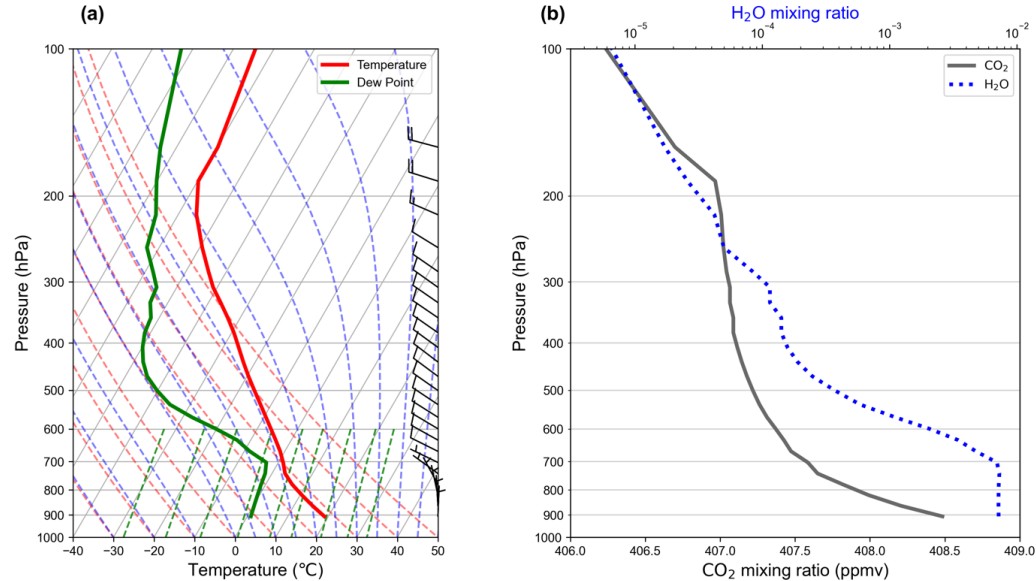

Figure A2. (a) Skew-T diagram of the atmosphere and (b) $CO_2$ and $H_2O$ VMR profiles of Fig. 1 scene.

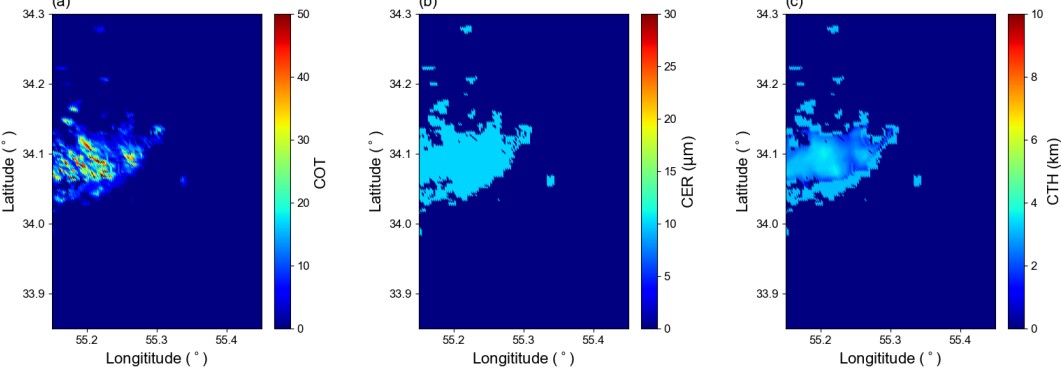

Figure A3. The cloud optical thickness (a), cloud liquid effective radius (b), and cloud top height (c) for the
3D simulation at 650 nm.



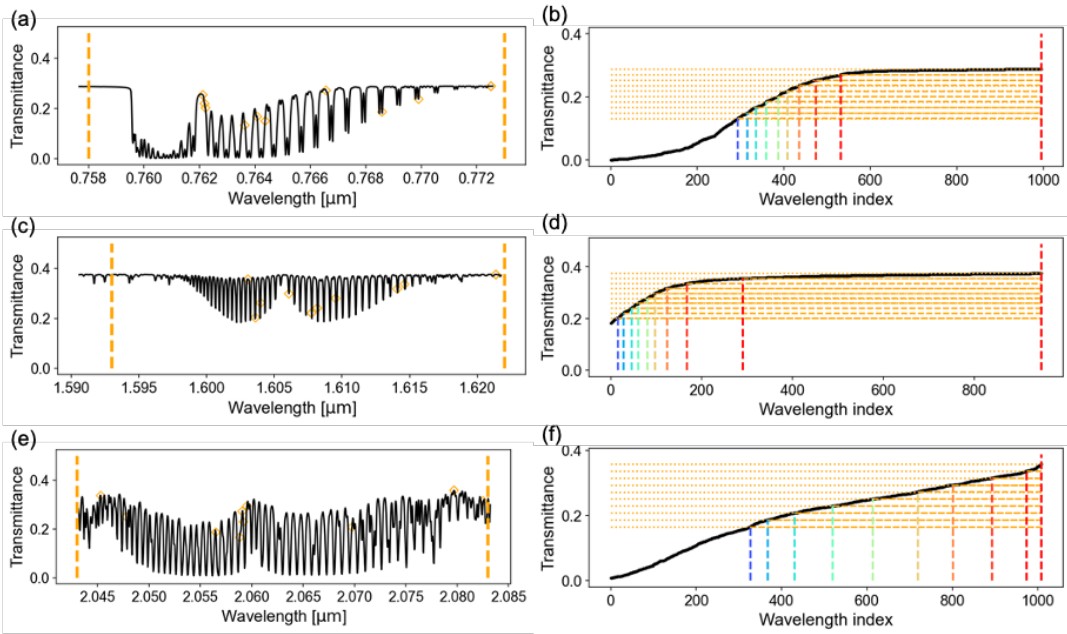

Figure A4. Simulated transmittance of (a) $O_2$-A, (c) $WCO_2$, and (e) $SCO_2$ bands derived from the atmospheric structure presented in Fig. A2. Right panels present the sorted transmittance with the selected wavelength index for (b) $O_2$-A, (d) $WCO_2$, and (f) $SCO_2$ bands. The orange markers on the left panels denote corresponding selected wavelengths shown in the right panels.



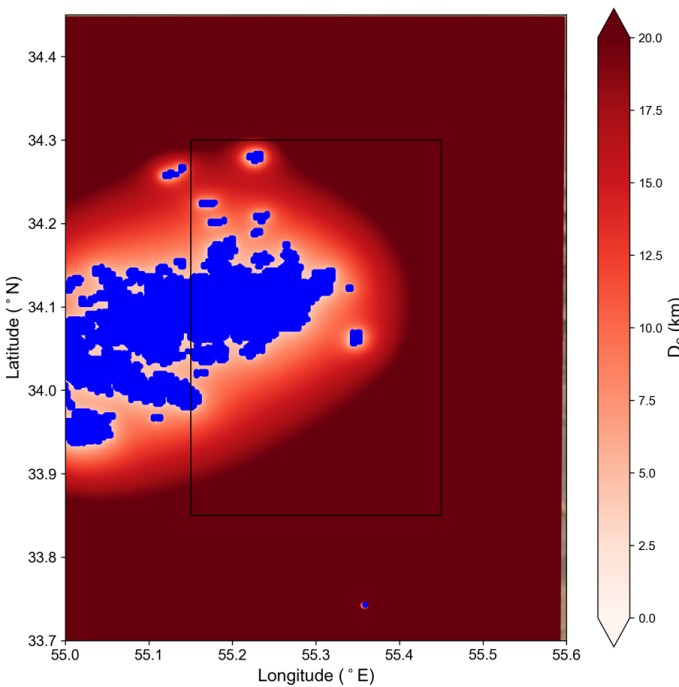

Figure A5. Distribution of the effective cloud distance, with blue dots marking the positions of the clouds. The black rectangle designates the analysis domain, while the entire domain represents the region of the RT simulation.



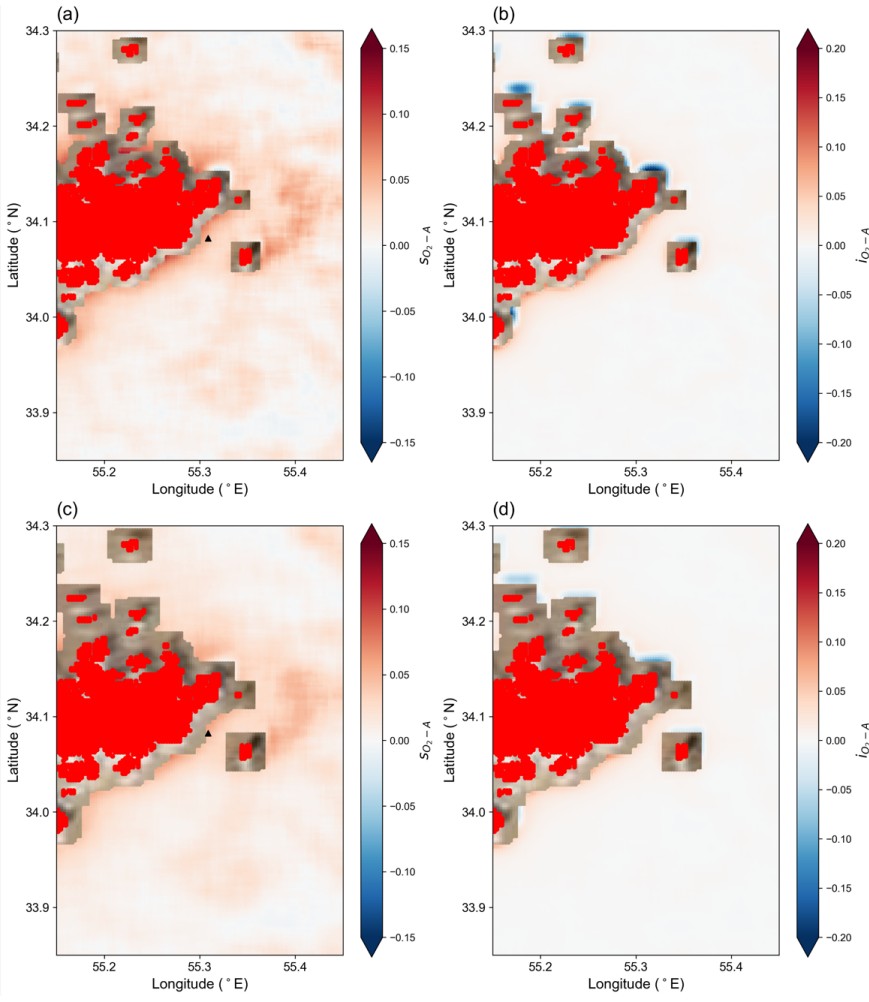

Figure A6. Distribution of $s$ (left) and $i$ (right) for the $O_2$-A band computed using average radiance from the (a, b) 9×9 and (c, d) 13×13 grid points. Red dots indicate the cloud pixels employed in the RT simulation.





B. Code walkthrough
The codes utilized for this study can be accessed from GitHub at: https://github.com/ywchen-
tw/OCO2. Subsequent sections specify the configuration file settings and provide an overview of
the simulation and analysis process.

a. Configuration file

The changeable parameters for the EaR$^3$T-OCO simulation are controlled by the configuration
file in CSV format. The following table describes the meaning and data type of each variable. If
the variable is not required to be specified, the default value is used.

| Parameter name | Description | Datatype | Required |
|---|---|---|---|
| descriptor | Case description | string | v |
| date | Date of interest | integer, YYYYMMDD | v |
| juld | Julian date of the year | integer | |
| | | | |
| pngwesn | Region for retrieving the MODIS RGB image | 4 floats | v |
| subdomain | Region for analysis | 4 floats | v |
| | | | |
| path_sat_data | directory of satellite files | string | v |
| l2 | File name of geolocated $X_{CO2}$ retrieval results data | string | |
| lt | File name of OCO-2 Level 2 bias-corrected $X_{CO2}$ and other select fields from the full-physics retrieval aggregated as daily files | string | |
| l1b | File name of calibrated, geolocated OCO-2 science spectra | string | |
| dia | File name of geolocated $X_{CO2}$ retrieval results plus algorithm diagnostic information | string | |
| met | File name of OCO-2 Level 2 meteorological parameters interpolated from global assimilation model for each sounding | string | |
| imap | File name of geolocated retrieved values of $X_{CO2}$ and fluorescence generated by the IMAP-DOAS algorithm | string | |
| co2prior | File name of OCO-2 Level 2 $CO_2$ prior based on $CO_2$ monthly flask record, global meteorology, and age of air | string | |
| sol | File name of the solar spectra | string | |
| | | | |
| nx | Interval of selected wavelength | Integer, default = 5 | |
| Trn_min | Minimum ratio of the largest transmittance for | float, $0 \leq$ Trn_min $< 1$, | |



| | wavelength selection | default = 0 | |
|---|---|---|---|
| abs_interpolation | Option for doing the interpolation | "single", "linear", or "trilinear", default = "trilinear" | |
| | | | |
| _aerosol | Add a homogeneous aerosol layer | TRUE or FALSE | |
| asy | Aerosol asymmetry parameter | float | |
| | | | |
| cth_thick | Cloud top height for thick clouds (km) | float | |
| cgt_thick | Cloud geometric thickness for thick clouds (km) | float | |
| cth_thin | Cloud top height for thin clouds (km) | float | |
| cgt_thin | Cloud geometric thickness for thin clouds (km) | float | |
| cot_Nphotons | Number of photons used for COT-Ref relationship simulation | float | |
| | | | |
| path_out | Directory of output files | string | |
| o2 | File name of $O_2$-A band simulation output | string | |
| wco2 | File name of $WCO_2$ band simulation output | string | |
| sco2 | File name of $SCO_2$ band simulation output | string | |
| | | | |
| retrieval | Retrieval version | | |
| | | | |
| ref_threshold | Radiance threshold @ 470 nm as cloudy pixel | float | v |
| modis_650_N_photons | Number of photons used for 650 nm simulation | integer | v |
| oco_N_photons | Number of photons used for OCO 3 bands simulation | integer | v |






b. Preprocess

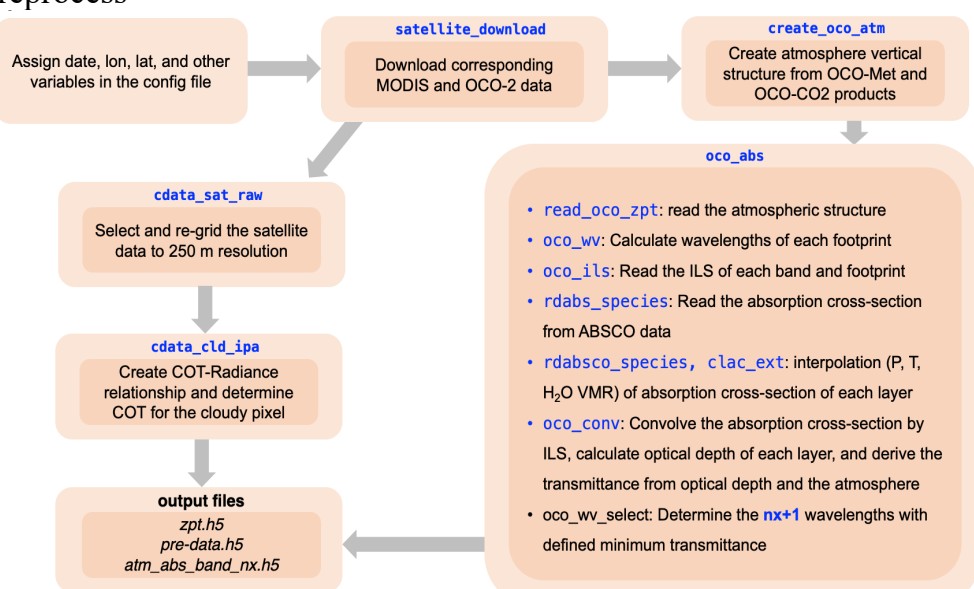


The oco_simulation.py code is the main code for data acquisition and radiance simulation. Here we
focus on the first part of the code (the preprocess function) dealing with the data download and
preprocessing.

1. satellite_download function: This function accesses the configuration file variables,
subsequently downloading the pertinent MODIS and OCO-2 data as dictated by the
specified date and geolocation details.
2. create_oco_atm function: Leveraging OCO-2 data, this function constructs vertical
profiles for temperature, water vapor, and wind.
3. oco_abs function: Based on the atmospheric profiles, this function computes the optimal
absorption coefficients for all three OCO-2 bands, subsequently identifying the
wavelengths for simulation.
4. cdata_sat_raw function: Extracts data from the MODIS and OCO-2 files, then
restructures this data to a 250 m resolution.
5. cdata_cld_ipa function: Using the EaR³T simulator, this function establishes a COT-
radiance relationship and designates a COT for every grid point.

Upon completion of the preprocessing stage, the system will generate the following files:

● *zpt.h5*: Details the vertical atmospheric structure.
● *pre-data.h5*: Contains information on cloud and radiance.
● *atm_abs_band_nx.h5*: Captures data on the absorption coefficients.

c. Simulation



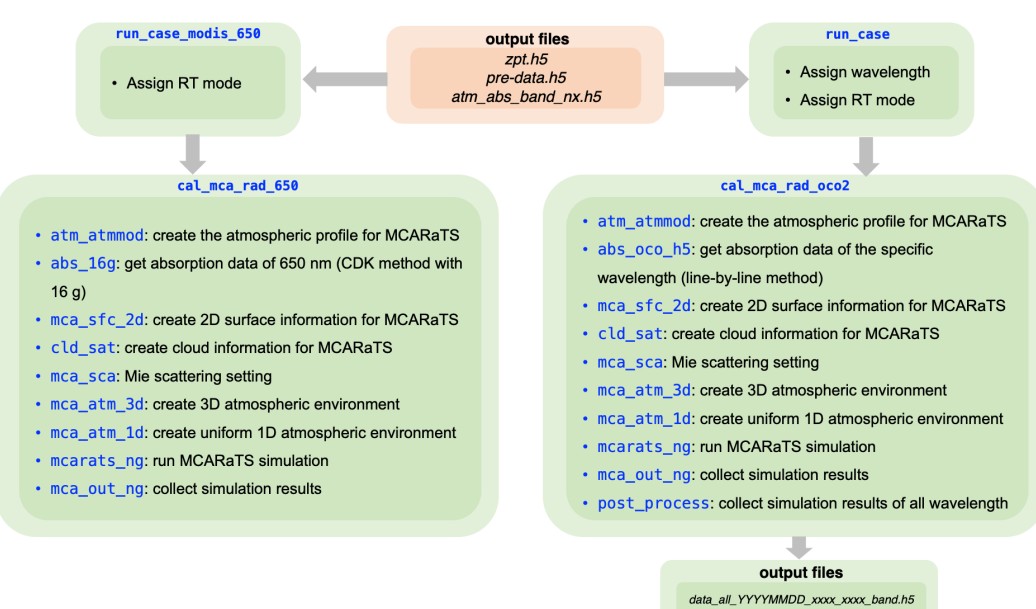

The second part of the oco_simulation.py code (run_case_modis and run_case functions) is
primarily concerned with radiance simulation.

1. **run_case_modis** function: This function drives the EaR3T simulator to simulate
radiance at 650 nm, operating in either the IPA mode or the 3D mode. It employs the
correlated-k distribution method as detailed in Chen et al. (2023). Upon completion,
simulation results are stored in the ***post_data.h5*** file.
2. **run_case_oco** function: This function activates the EaR$^3$T simulator to simulate
radiance for each designated wavelength, as identified by the **oco_abs** function. It utilizes
the IPA mode for clear-sky simulations and the 3D mode for real-world conditions. The
respective simulation outcomes for each band are archived as
***data_all_YYYYMMDD_xxxx_xxxx_band.h5***.

d. Postprocess





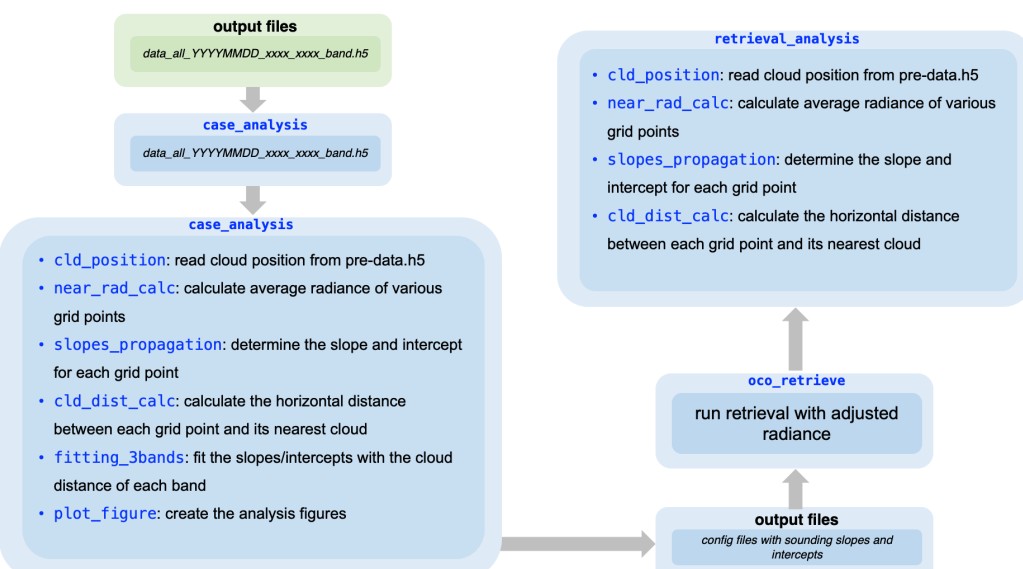

Following the radiance simulation, we proceeded with an analysis leveraging the case_analysis.py code, and subsequently executed oco_retrieve.py to extract the mitigated $X_{CO2}$.

1. **case_analysis** function: This function accesses the output files from the radiance simulation and computes the mean radiance across various average sizes, a process managed by the **near_rad_calc** function. Next, the **slopes_propogation** function determines the slope and intercept for each grid point. With the help of the **weighted_cld_dist_calc** function, the effective cloud distance for each grid point is gauged based on cloud positioning. The **fitting_3bands** function determines the most suitable fitting coefficients for the 3D parameters and the effective cloud distance. Following this, the effective cloud distance for every footprint is established and used to derive the corresponding parameterized slopes and intercepts. All results are consolidated in the configuration file.

2. **oco_retrieve** function: Initially, this function adjusts the radiance of the footprint in line with the set slopes and intercepts. It then triggers the OCO retrieval algorithm with the modified spectra to obtain the mitigated $X_{CO2}$.

3. **case_retrieval_analysis** function: This function reviews the output of the mitigated XCO2 and juxtaposes the findings with the OCO-2 L2 product.