# Peer review of "Mitigation of satellite OCO-2 CO2 biases in the vicinity of clouds 1"

_EGUsphere, 2024_

## Community Comment (CC1)

Comment on egusphere-2024-1936:

This manuscript develops a method for the mitigation of 3D radiative transfer effects on retrievals of carbon dioxide concentration from the Orbiting Carbon Observatory satellites. The novelty of this work is that it provides a pathway for physics-based mitigation of 3D radiative transfer effects using parameterizations that can be applied operationally.
I enjoyed reading this paper, and I have several comments and suggestions detailed below:

As I understand it, all of the forward modelling of the OCO bands in this paper utilizes the linear approximation suggested by Schmidt et al. (in prep). For this paper, it is important that we know how the error in the linear approximation propagates into uncertainty in the relationship between $\Delta X_{CO2}$ and the radiances i.e., how accurate is the reference calculation?
At the moment, Section 2 states the result of Schmidt et al. (in prep) but doesn't provide much physical justification for the linear approximation itself. I think this section would benefit from a short paragraph discussing approximate acceleration methods for 3D RT such as (Partain et al., 2000; Doicu et al, 2020) in comparison to exact calculations like Emde et al. (2011), so that the strengths/weaknesses (accuracy vs. speed) of the linear approximation can be contextualized.

The mitigation parameterization is based on simulated scenes derived from observations. Due to weak atmospheric scattering, the 3D enhancement effect studied here depends primarily on cloud-surface interactions. These will strongly depend on the geometric distance between cloud and surface (i.e., cloud base height and thickness). At the moment, the methodology doesn't state how the cloud base height is retrieved from the MODIS observations to form a synthetic cloud field, or its uncertainty. This procedure's uncertainty will feed into the simulations and affect how the intercept and slope parameters scale with effective distance. It would be good to address this within the manuscript as it will affect both the baseline and bypass approaches.

For the parameterization, it might be beneficial to have a generalized distance that doesn't just take into account horizontal distance but rather the 3D distance of a surface point from cloud base (or side). For isotropic scatterers, the downwelling flux impinging on a point on the surface would scale with the inverse square of this distance, so square distance weighting seems like a good choice as used in the study. Along with that, not all clouds are equally bright, and their 3D enhancement should increase with overall cloud brightness. It might be useful to have a generalized distance that includes weighting by cloud reflectance. This might help the parameterization/bypass approach generalize more effectively.

The vertical distribution of aerosol will also influence the distance scaling of the slope and intercept parameters. Currently, the study examines aerosol within the cloud layer and states that it localizes enhancements to regions closer to cloud due to reduced free paths. The effect of an elevated aerosol layer may differ. Higher-altitude scattering layers tend to

increase the horizontal distance over which 'adjacency' effects occur (Minomura et al., 2001). I think it would be worthwhile to discuss the role of the vertical distribution of the aerosol.

The issues of cloud base height and aerosol don't seem insurmountable at least for measurements acquired in vicinity of A-train sensors. I think it would be beneficial to provide a sketch of how these additional measurements can be used to constrain these other factors and develop an operational parameterization.

References:

Partain, P. T., A. K. Heidinger, and G. L. Stephens (2000), High spectral resolution atmospheric radiative transfer: Application of the equivalence theorem, *J. Geophys. Res.*, 105(D2), 2163–2177, doi:10.1029/1999JD900328.

Doicu, A.; Efremenko, D.S.; Trautmann, T. A Spectral Acceleration Approach for the Spherical Harmonics Discrete Ordinate Method. *Remote Sens.* **2020**, *12*, 3703. https://doi.org/10.3390/rs12223703

Minomura, Mitsuo, Hiroaki Kuze, and Nobuo Takeuchi. "Adjacency Effect in the Atmospheric Correction of Satellite Remote Sensing Data: Evaluation of the Influence of Aerosol Extinction Profiles." *Optical Review* 8, no. 2 (March 1, 2001): 133–41. https://doi.org/10.1007/s10043-001-0133-2.

---

## Author Comment (AC3)

Dear reviewer,

We appreciate your insightful comments and suggestions. Below, we provide detailed responses to each point raised. In this document, the reviewer's comments and suggestions are highlighted in red, our responses are in blue, and references to the original manuscript content are in black.

We understand your concerns regarding the limited validation in the manuscript, particularly the inclusion of only one case study. While additional cases would indeed strengthen the validation for operational use, the primary aim of this paper is to introduce and demonstrate the mitigation approach we have developed. This work serves as an introduction to new physics-based mitigation methods for OCO retrievals, distinct from existing statistical bias correction methods.

We appreciate your suggestion to focus more on the results. However, as this paper introduces new methods, we believe it is essential to provide a detailed description of the methodology and setup, in addition to demonstrating the methods. To address your comment, we will add a 'manuscript guide' at the end of the introduction in the revised version of the manuscript, which clearly separates the sections of the manuscript that are intended to introduce the methodology vs. the sections that present preliminary results. Again, the emphasis of this manuscript is to bring a bias-correction to real-world data, and that requires the introduction of a rather extensive radiative transfer tool.

In response to the feedback, we included two additional cases applying the same bypass parameters and a comparison with the baseline method in this response document on page 2. Due to manuscript length considerations, we will add the first case as an example in the appendix. We remain committed to enhancing this methodology and will continue to expand validation in future work.

General comments:

Overall, the subject of the study is very compelling and a significant contribution to the community. Especially considering future upcoming green house gas missions.

Thank you for your positive feedback on our research. We appreciate your recognition of its relevance for future missions. We have carefully considered your comments and provided detailed responses to address your concerns.

Major Comments

However, the study misses depth in how far the 3D cloud bias correction has been investigated.

We acknowledge the extensive efforts made by the community to address 3D cloud bias, such as recent advancements using machine learning approaches (e.g., applying the machine learning bias correction from Mauceri et al. (2023) on V11 data product). However, the methods adopted by the OCO team so are not fully physics-based, which means that they do not operate at the radiance level (where cloud-induced 3D-RT effects introduce perturbations), instead focusing on the product level in a statistical manner. Here, we operate at the radiance *and* footprint level for real-world satellite spectroscopy data – to our knowledge the first time that this has been done. The most closely related publications to our study are the ones by Emde et al. (2022) and Merrelli et al. (2015) since they studied the impact of cloud-induced spectroscopic perturbations on the products. However, neither of these studies worked with real-world data, focusing on synthetic data instead. The value of our study lies in presenting a mitigation strategy that directly addresses the 3D cloud bias from a physical (radiance) perspective, focusing on the mechanisms behind these biases rather than relying solely on empirical corrections (as done in previous, statistical studies). In this sense, we would actually say that our manuscript *adds* depth over previous studies. We might have missed what the reviewer is referring to specifically, perhaps details of the current retrieval algorithm. However, since it is stated in the manuscript, we probably do not need to point out here that the physics of the current operational algorithm does not account for horizontal photon transport and therefore by definition misses an important piece of reality, which we bring back with our work.

**Major concern is that the developed approach has been applied only to a single, hand-picked scene. This is simply not enough to make any guesses towards the performance of the approach when applied operationally.**

We understand where this concern is coming from, and are glad that the reviewer is pointing out this perception. Indeed, this paper starts out with a specific scene, and we acknowledge that more cases are needed to fully evaluate the method's performance for operational use. However, the primary goal of this study is to describe the methodology, and in that regard, the scene we used is simply used for illustration. We did not specifically hand-pick a scene because 'it worked." In reality, we have tried this method on several scenes, and the selected case simply provided a clear illustration to demonstrate the mechanics of the approach. However, this does not mean that the other cases did *not* work. To show this, we include two other cases in this response. Because of considerations regarding manuscript length, we will add only the first additional case as an example in the appendix. In future work, we plan to apply the method to a larger set of scenes to assess its robustness and operational applicability, but that future work is distinct from our intention here, which lays the ground work and describes the methodology. On this note: Future version of the algorithm will come with improvements, part of which were motivated by the reviewer comments to the release version of our code, documented with this publication.

To address your comment on validation, we have included two additional cases from the same month and general geographic area as the case in Fig. 2. These cases apply the bypass mitigation method based on parameters outlined in Table R1 (Table 2 in the manuscript) and are compared to the baseline method as validation examples (shown below). 1D-RT and 3D-RT simulations were conducted for these two cases to derive the correct slope and intercept parameters. Thus, we can evaluate the bypass mitigation based on Table R1, with the comparisons illustrated in Figs. R3 and R6.

The results from case 1 indicate that the bypass method yields a mitigation trend similar to that of the baseline method, although with lower magnitudes. The difference between Fig. R2a and b is potentially due to differences in surface altitude and albedo, solar geometry, AOD, and other environmental factors. This case demonstrates promising results, yet adjustments to the bypass parameters with more scene variables are necessary for effective operational use.

In contrast, case 2 shows less favorable performance of the bypass method compared to the baseline method. In case 2, the baseline method reveals a weaker correlation between $\Delta X_{CO2}$ and effective cloud distance, likely due to confounding factors, such as surface albedo effects. This indicates that the bypass method may be less effective in mixed or complex cloud-surface conditions. Given the length constraints of the current manuscript, we have decided to add only case 1 as an example in the appendix.

Table R1. The same table as Table 2 in the manuscript. Amplitude and e-folding distances for **s** and **i** fittings of the simulation with a homogeneous aerosol layer in the $O_2$-A, $WCO_2$, and $SCO_2$ bands for 1.0 km geometric cloud thickness of low clouds.

| | Slope | | | Intercept | | |
|---|---|---|---|---|---|---|
| | $s_{O_2-A}$ | $s_{WCO_2}$ | $s_{SCO_2}$ | $i_{O_2-A}$ | $i_{WCO_2}$ | $i_{SCO_2}$ |
| $a_s$ or $a_i$ | 0.457 ± 0.094 | 0.123 ± 0.037 | 0.250 ± 0.041 | 0.755 ± 0.327 | 0.648 ± 0.227 | 0.847 ± 0.406 |
| $d_s$ or $d_i$ (km) | 3.82 ± 0.44 | 5.04 ± 0.89 | 4.58 ± 0.78 | 2.69 ± 0.32 | 2.91 ± 0.31 | 2.35 ± 0.33 |

Additional case 1:

[Figure]

Figure R1. Satellite true-color imagery of MODIS Aqua from NASA Worldview on 5 October 2019 with (a) $X_{CO_2}$ in OCO-2 level 2 data, (b) mitigated $X_{CO_2}$ retrieved from the adjusted spectra and (c) difference between the mitigated and original $X_{CO_2}$ values.

[Figure]

Figure R2. (a) Relationship of $\Delta X_{CO_2}$ with $D_e$ based on parameterized slopes and intercepts from the bypass method in Table 2. (b) Corresponding relationship using slopes and intercepts derived from the baseline approach for Fig. R3.

[Figure]

Figure R3. (a) Scatter plot comparing the $X_{CO2}$ anomaly of the OCO-2 L2 product (in black) to its value post-spectra adjustment (in red) for the case shown in the figure above, plotted against $D_e$. The $X_{CO2}$ anomaly is defined as retrieved $X_{CO2}$ – true $X_{CO2}$, with the true $X_{CO2}$ defined by the average $X_{CO2}$ of footprints with a $D_e$ greater than 15 km (405.96 ppm in this case). The orange shade indicates the 1 ppm mission requirement. (b) Histograms and probability density functions (PDFs) for the $X_{CO2}$ anomaly of the OCO-2 L2 product (in black) and post spectra adjustment (in red) within a 15 km $D_e$. This corresponds to the blue-shaded region in (a). The FWHM values of the PDFs of v10r and adjusted data points are 5.25 and 4.28, and the PDF averages are 0.93 and 0.18, respectively. The average change in $X_{CO2}$ after the spectra adjustment for De less than 15 km is -0.86 ppm.

Additional case 2:

[Figure]

Figure R4. Satellite true-color imagery of MODIS Aqua from NASA Worldview on 18 October 2018 with (a) $X_{CO2}$ in OCO-2 level 2 data, (b) mitigated $X_{CO2}$ retrieved from the adjusted spectra and (c) difference between the mitigated and original $X_{CO2}$ values.

[Figure]

Figure R5. (a) Relationship of $\Delta X_{CO2}$ with $D_e$ based on parameterized slopes and intercepts from the bypass method in Table 2. (b) Corresponding relationship using slopes and intercepts derived from the baseline approach for Fig. R5.

[Figure]

Figure R6. (a) Scatter plot comparing the $X_{CO2}$ anomaly of the OCO-2 L2 product (in black) to its value post-spectra adjustment (in red) for the case shown in the figure above, plotted against $D_e$. The $X_{CO2}$ anomaly is defined as retrieved $X_{CO2}$ – true $X_{CO2}$, with the true $X_{CO2}$ defined by the average $X_{CO2}$ of footprints with a $D_e$ greater than 15 km (405.69 ppm in this case). The orange shade indicates the 1 ppm mission requirement. (b) Histograms and probability density functions (PDFs) for the $X_{CO2}$ anomaly of the OCO-2 L2 product (in black) and post spectra adjustment (in red) within a 15 km $D_e$. This corresponds to the blue-shaded region in (a). The FWHM values of the PDFs of v10r and adjusted data points are 10.24 and 8.80, and the PDF averages are 0.27 and 0.20, respectively. The average change in $X_{CO2}$ after the spectra adjustment for De less than 15 km is -0.45 ppm.

The study states that it developed a software tool for the automated calculation of spectral radiances from OCO-2. However, **the automation is not exploited to analyze a representative sample size of OCO-2 observations.**

We agree that analyzing the entire dataset, or even 1% of OCO-2 observations, using full 3D-RT simulations is impractical due to the high computational cost. This constraint motivated the development of the bypass method, which aims to significantly reduce the need for extensive 3D-RT calculations. However, before the bypass method can be

applied operationally, we still need to use the tool to analyze several dozen to hundreds of cases under diverse conditions to build a more generalized parameterization. While the automation feature helps streamline radiance calculations, further validation with additional cases is a key focus for future work. We are currently in the method development stage, with larger-scale case analyses planned as the next step.

**Furthermore, for this single selected scene the strongest biases seem to be collocated with cloud shadows while the authors argue that those shadows are outside the scope of this study.** For this research to be useful to the community it needs to show that it can be generalized (e.g. various SZA, ocean (where 3D cloud effects are strongest), land surface types, different cloud types, different viewing modes (nadir and glint)).

As mentioned in lines 381-384, we refer to Massie et al. (2023), who found that relatively few cloud shadow retrievals exist in the OCO-2 Lite files due to the pre-retrieval cloud screening algorithms. In addition, B11 retrieval has improved in filtering out footprints under shadow (at least for the cases we analyzed). We don't think that the remaining few cloud-shadow footprints passing the pre-screening have the same importance as the clear-sky footprints affected by clouds in the vicinity.

We agree that demonstrating the method's applicability to a broader range of conditions, such as different solar zenith angles, surface types, cloud types, and viewing geometries (nadir and glint), is crucial for generalization. To address part of your concern, we will add a section before Section 5.5 (3D effect mitigation) discussing the impact of surface albedo (related to surface types) and solar zenith angle (SZA).

Fig. R7 and R8 present the exponential decay fitting of the slope and intercept of the O2-A band under SZA and surface albedo. The x-axes are the effective horizontal cloud distance ($D_e$), which is defined as the average distance of the pixel to the surrounding cloudy pixels, weighted by the inverse square distance to the cloudy pixel (Eq. R1, Eq. 5 in the manuscript):

$$D_e = \frac{\sum_{i \,\in\, \{\text{surrounding clouds}\}} w_i D_i}{\sum_{i \,\in\, \{\text{surrounding clouds}\}} w_i} \quad \text{(R1)}$$

The exponential decay relationships in Fig R7-8 are fitted between slope ($s$) and intercept ($i$) parameters and $D_e$ using Eq. R2-3 (Eq. 6-7 in the manuscript). The amplitude ($a_s$, $a_i$) and e-folding distance ($d_s$, $d_i$) are the fitting parameters (separate sets for $s$ and $i$). The result of amplitude and e-folding distance are presented in Table R1.

$$s = a_s \times \exp(-\frac{D_e}{d_s}) \quad \text{(R2)}$$

$$i = a_i \times \exp(-\frac{D_e}{d_i}) \quad \text{(R3)}$$

We have tested these approaches for ocean glint cases and plan to have the next paper discussing specific ocean cases since their biases behave differently than land nadir cases. However, EaR³T-OCO is already capable of simulating glint cases. The impact of cloud types and properties will be studied in the future.

- Additional text to be added in Section 5.5:

"Solar geometry and surface albedo are significant factors influencing the 3D cloud effect. Figures R9 and R10 (will be added in the appendix) illustrate how these variables impact the 3D cloud effect in the $O_2$-A band under different conditions. By combining results across various solar zenith angles and surface albedo values, we developed a two-variable linear parameterization using $a_s$ and $d_s$ (slope parameters) and $a_i$ and $d_i$ (intercept parameters). As summarized in Table R1, we observe that the amplitude of the slope and intercept is inversely proportional to surface albedo and directly proportional to the cosine of the solar zenith angle (denoted as $\mu$). Additionally, the e-folding distances of the slope are negatively proportional to both surface albedo and $\mu$, while those of the intercept are positively proportional to surface albedo and negatively proportional to $\mu$. In general, higher surface albedo reduces the 3D cloud effect, as additional photons reaching the sensor represent a smaller fraction of the total signal. Conversely, lower solar zenith angles result in a smaller amplitude but longer e-folding distance, causing the 3D effect to extend further from the clouds."

[Figure]

Figure R7. Parameterization of (a) slope and (b) intercept for $O_2$-A band with effective cloud distance, varied by solar zenith angle, while holding surface albedo and aerosol optical depth (AOD) constant.

[Figure]

Figure R8. Parameterization of (a) slope and (b) intercept for O2-A band with effective cloud distance, varied by surface albedo, while holding solar zenith angle and AOD constant.

Table R1. The parameterization of as and ds of slope and ai and di of intercept for the three OCO-2 bands. Errors represent fitting uncertainty only and may be underestimated.

|  | slope | intercept |
|---|---|---|
| $O_2$-A | $a_s = -0.34 × alb_{O2-A} + 0.57 × μ - 0.03$
$d_s = -3.2 × alb_{O2-A} - 9.9 × μ + 14.9$ | $a_i = -0.60 × alb_{O2-A} + 0.36 × μ + 0.72$
$d_i = 0.42 × alb_{O2-A} - 2.1 × μ + 5.2$ |
| $WCO_2$ | $a_s = -0.15 × alb_{WCO2} + 0.11 × μ - 0.05$
$d_s = -30.7 × alb_{WCO2} - 7.0 × μ + 27.5$ | $a_i = -2.07 × alb_{WCO2} + 1.65 × μ + 1.17$
$d_i = 0.63 × alb_{WCO2} - 1.6 × μ + 3.7$ |
| $SCO_2$ | $a_s = -0.18 × alb_{SCO2} + 0.29 × μ - 0.03$
$d_s = -22.6 × alb_{SCO2} - 21.2 × μ + 34.9$ | $a_i = -2.77 × alb_{SCO2} + 2.22 × μ + 1.14$
$d_i = 0.51 × alb_{SCO2} - 1.73 × μ + 3.35$ |

The study currently reads more like a description of the work that was performed rather than being focused on the outcome of the work. The outcome is what your reader is interested in. I would suggest picking either the 3D cloud correction based on the 3RT simulations or the 'bypass' method as the outcome of this work and explore the chosen method further (explore more scenes to better estimate performance once applied operationally).

We appreciate this input. However, this manuscript is intended to introduce a new method, and therefore intentionally reads like a description of the algorithm. Of course, we also present results, but we cannot talk about results without fully introducing the method first. Also, the two methods (baseline: full 3D-RT; bypass: parameterization based on 3D-RT) are strongly related, and it is difficult to elaborate on the bypass method without describing the baseline method. Since this is the first paper discussing

the new method, we think it is important to describe the approaches and their settings in detail as well. To address this, we will edit the introduction to state the purpose of this study more clearly – see below.

- Original text (lines 94-110) for Introduction in Section 1:
  "In this paper, we introduce the direct application of the scene-dependent slope and intercept parameters to the correction of 3D-RT biases, using a modified version of the Education and Research 3D Radiative Transfer Toolbox (EaR$^3$T; Chen et al., 2023), tailored specifically for OCO (EaR$^3$T-OCO). This tool simulates the radiance for OCO-2 footprints, using, among other data (Section 3), imagery from the MODIS on the Aqua satellite, which is approximately 6 minutes behind OCO-2 within the NASA A-Train (afternoon) satellite constellation. From these, the slope and intercept parameters for the OCO-2 footprints of a given scene are derived, then used to undo the 3D-RT perturbation in the observed radiance spectra, and subsequently in the $X_{CO2}$ retrieval. The spectral dimensionality (3x1024 for the three OCO-2 channels), and thus computational effort, are thereby greatly reduced because our methodology (Section 4) only requires a few selected wavelengths. From our results for a few scenes in different regions of the world, we develop a parameterization of slope and intercept as a function of effective cloud distance and other scene variables (Section 5). We then show that the correction of 3D-RT biases in the spectroscopy and $X_{CO2}$ retrievals works both on the footprint-by-footprint basis, and by way of the new parameterization. This parameterization not only enhances our physics-based understanding of the $X_{CO2}$ retrieval biases introduced by clouds, but also offers a computationally efficient pathway for applying these insights globally across extensive datasets. Conclusions are drawn in Section 6, and future work is discussed in Section 7. The appendix explains the functionality of EaR$^3$T-OCO."

- Revised text for Introduction in Section 1, with the main changes underlined:
  "This study introduces new physics-based mitigation approaches for addressing 3D cloud biases in OCO-2 data and demonstrates their effectiveness using real OCO-2 observations. We apply scene-dependent slope and intercept parameters directly to correct 3D-RT biases at the radiance level, using a modified version of the Education and Research 3D Radiative Transfer Toolbox (EaR$^3$T; Chen et al., 2023), tailored specifically for OCO (EaR$^3$T-OCO). This tool simulates the radiance for OCO-2 footprints, using, among other data (Section 3), to derive slope and intercept parameters. The spectral dimensionality (3x1024 for the three OCO-2 channels), and thus computational effort, are thereby greatly reduced because our methodology (Section 4) only requires a few selected wavelengths. The slope and intercept parameters are then used to undo the 3D perturbation in the observed radiance spectra and subsequently in the $X_{CO2}$ retrieval (Section 5). We further develop a parameterization of slope and intercept as a function of effective cloud distance and other scene variables to bypass the 3D-RT calculation. We then show that the correction of 3D-RT biases in the spectroscopy and $X_{CO2}$ retrievals works both on the footprint-by-footprint basis, and by way of the new

bypass method. This parameterization, or bypass method, not only enhances our physics-based understanding of the $X_{CO2}$ retrieval biases introduced by clouds, but also offers a computationally efficient pathway for applying these insights globally across extensive datasets. Conclusions are drawn in Section 6, and future work is discussed in Section 7. The appendix explains the functionality of EaR$^3$T-OCO."

Minor Comments

The paper often refers to qualitative statements that should be quantified or omitted. I pointed out individual instances below.

The abstract should be shortened and more to the point. What are the key takeaways from this study. Not necessary to expose all the 'sausage making' in the abstract.

Thank you for the suggestion. We will shorten the abstract as suggested:

- Revised abstract:

"Accurate and continuous measurements of atmospheric carbon dioxide ($CO_2$) are essential for climate change research and monitoring of emission reduction efforts. NASA's Orbiting Carbon Observatory (OCO-2/3) satellites have been deployed to measure the column-averaged $CO_2$ dry air mixing ratio ($X_{CO2}$) with a designed uncertainty of less than one ppm for regional average. Although cloudy measurements are screened out, nearby clouds can still cause retrieval biases due to limitations in the forward one-dimensional (1D) radiative transfer (RT) model used in the OCO retrieval algorithm, which does not account for the scattering from clouds near the satellites' footprints. These biases, known as three-dimensional (3D) effects, can be quantified using 3D-RT models, but they are computationally expensive, especially for hyperspectral applications like OCO-2/3. This paper employs a linear approximation for each OCO-2 spectral band to represent the 3D-RT perturbations on OCO-2 spectra and reduce computational demands. We apply these metrics calculated by 3D-RT to spectrally adjust the real measured OCO-2 radiance prior to the operational retrieval to undo cloud vicinity effects without modifying the standard OCO-2 retrieval code. Additionally, a parameterization method is developed to bypass the need for 3D-RT simulations by incorporating effective cloud distance and other scene variables. The spectral adjustment mitigates $X_{CO2}$ retrieval biases in proximity to clouds over land for two cases shown in the study – the first physics-based radiance level correction of 3D-RT effects on OCO-2/3 retrievals. While the proposed method is computationally efficient for operational use, further validation is required for diverse surface and atmospheric conditions."

I would suggest to merge section 1 and 2.

Thank you for the suggestion. We agree that section 2 could be seen as an extension of the introduction. However, section 1 is already quite extensive, and combining these sections may make the introduction overly long. By keeping sections 1 and 2 separate, we aim to maintain reader focus on the linearization of the 3D effect, which is better emphasized as a distinct section.

Specific comments by Line:

L20: quantify 'high precision' or omit

Thank you for pointing out the issue. We update the text as below:

- Original text (Lines 18-20):
  "NASA's Orbiting Carbon Observatory (OCO-2/3) satellites have been deployed to measure the column-averaged $CO_2$ dry air mixing ratio ($X_{CO2}$) with very high precision."
- Revised text, with the main changes underlined:
  "NASA's Orbiting Carbon Observatory (OCO-2/3) satellites have been deployed to measure the column-averaged $CO_2$ dry air mixing ratio ($X_{CO2}$) with a designed uncertainty of less than one ppm for regional average."

L20 – L23: Sentence starting with 'Although …' is hard to digest and should be simplified, maybe broken up.

Thank you for your comment. We will revise the sentence to improve readability, as shown below:

- Original text (Lines 20-23):
  "Although cloudy measurements are screened out, nearby clouds can still cause retrieval biases because the forward one-dimensional (1D) radiative transfer (RT) model used in the OCO retrieval algorithm does not account for the scattering induced by clouds in the vicinity of the OCO-2/3 footprints."
- Revised text:
  "While most cloudy footprints are screened out, clear-sky observations can still be biased by nearby clouds. This bias arises because the forward one-dimensional (1D) radiative transfer (RT) model used in the OCO retrieval algorithm does not account for scattering from clouds near the OCO-2/3 footprints."

L27: remove 'with two metrics (called slope and intercept)'

Thank you for the suggestion. We will edit the abstract as suggested (as shown previously).

L28: remove 'and accelerate the radiative transfer by a factor of 100'

Thank you for the suggestion. We will edit the abstract as suggested (as shown previously).

L31 – L35: Sentence starting in 'EaRT-OCO .. ' -> move out of abstract.

Thank you for the suggestion. We edited the abstract as suggested (as shown previously).

L36: remove '– the first physics-based correction of 3D-RT effects on OCO-2/3 retrievals'

We would like to clarify that the current 3D bias mitigation methods proposed by Massie et al. (2022) and Mauceri et al. (2023) for OCO are primarily statistical-based. In contrast, the methods we proposed in this study are based on a physical understanding of the mechanism difference between 1D and 3D radiative transfer.

L37-L43: shorten, simplify discussion of 'bypass' method.

Thank you for your comment. We revised the sentence as shown below:

● Original text (Lines 37-43):
"Although the accelerated 3D-RT radiance adjustment step is faster than full 3D-RT calculations for all OCO spectral bands, it still requires at least as much computational effort as the $X_{CO2}$ retrieval itself. To bypass 3D-RT altogether, the slope and intercept metrics are parameterized as a function of the weighted cloud distance of a footprint and several other scene parameters, all of which can be derived directly from Aqua-MODIS imagery. While this method is fastest and thus feasible for operational use, it requires careful validation for various surface and atmospheric conditions."
● Revised text:

"The accelerated radiance adjustment step is faster than full 3D-RT calculations but still requires similar computational effort as the $X_{CO2}$ retrieval. To avoid 3D-RT completely, the bypass method parameterizes the slope and intercept as a function of the weighted cloud distance. Although this approach is the fastest and suitable for operational use, it requires thorough validation under various surface and atmospheric conditions."

L62: quantify 'accuracy' requirement from the two cited studies.

Thank you for the comments. We add a description of their emphasis on accuracy.

● Line 61-63: Deng et al. (2016) and Crowell et al. (2018) also emphasize the significance of the level of $X_{CO2}$ accuracy for reliable $CO_2$ flux determination."
● Revised text, with the main changes underlined:

"Deng et al. (2016) and Crowell et al. (2018) highlight the importance of achieving high $X_{CO2}$ measurement accuracy for reliable $CO_2$ flux estimation. Deng et al. (2016) show

that the assimilation of GOSAT $X_{CO_2}$ data with a precision of approximately 0.5–1.0 ppm can significantly improve regional CO2 flux estimates over both land and ocean. Similarly, Crowell et al. (2018) emphasize that an $X_{CO_2}$ precision of 0.5–1.0 ppm is essential for detecting regional flux perturbations, especially in cloud-prone and high-latitude regions where $CO_2$ fluxes are difficult to constrain accurately using ground-based sensors alone."

L76: remove sentence 'The cloud-related …'

Thank you for the suggestion. We remove Line 76: "The cloud-related bias is also evident when examining individual footprints." as suggested.

L90-91: Restate comment that no 'practical strategies' have been developed to correct 3D cloud effects based on the physical understanding. The study by Mauceri et al (2023) uses physics derived variables to correct for 3D cloud biases.

We appreciate your comments on the practical strategy. You are correct that the machine learning-based method developed by Mauceri et al. (2023) is indeed a practical approach to the real data. We understand that Mauceri et al. (2023) and Massie et al. (2022) use several physics-*derived* variables, such as H3D, HC, and CSNoiseRatio. However, these variables capture the 3D cloud effect only at a single band or average of continuum wavelengths. More importantly, these metrics are not used to correct the 3D cloud effect at the *radiance level*, but operate primarily on the L2 products. While cloud distance, similar to our study, can indicate potential 3D cloud biases, these variables alone cannot fully capture the reflectance-dependent 3D cloud effect across the entire spectrum. Although Mauceri et al. (2023) apply machine learning-based corrections, it is still a *statistical* mitigation approach in nature and does not go into the fundamental physics (i.e., the radiance level). We want to emphasize that our approach is based on a footprint-by-footprint *deterministic* rather than a multi-footprint *statistical* approach, albeit with some simplifications that are noted in the original manuscript (with more detail in the revised manuscript, responding to a different reviewer). It is a radiance-only approach and different from the existing statistics method. To make this more clear, we will make edits as shown below:

- Original text (Lines 89-92):
  "Although the physical mechanism of the $X_{CO_2}$ 3D cloud retrieval bias is now largely understood, practical strategies for applying these insights to a bias correction have not been developed thus far. Mauceri et al. (2023) employed machine learning techniques to correct for 3D cloud biases using observations from the Total Carbon Column Observing Network (TCCON)."

- Revised text, with the main changes underlined:

  "Although the reflectance-dependent physical mechanisms of the $X_{CO_2}$ 3D cloud retrieval bias are now largely understood, strategies for applying these insights to bias correction

have thus far been done empirically or statistically. For example, Massie et al. (2022) proposed an empirical lookup table to correct 3D cloud biases based on a 3D metric, and Mauceri et al. (2023) used machine learning techniques combined with TCCON observations. While both approaches are operationally applicable, they rely on statistical corrections rather than true physical radiance difference of the 3D cloud effect across the entire spectrum."

L93: Please also include/cite work by Massie et al where they worked on correcting 3D cloud biases with linear fits to physics derived variables.

Thank you for the suggestion. As described above, we have included Massie et al. (2022) and discussed the physics-derived variables they used, such as H3D, HC, and CSNoiseRatio.

L106: 'on the a footprint-by-footprint '

Thank you for pointing out the typo. We change this to "on a footprint-by-footprint"

L126: specify that range 'dynamic range of interest for reflectance'

Figure 1 shows the spread of perturbations at low reflectance, which are primarily due to photon noise in the Monte Carlo RT simulations. To avoid large simulation uncertainties in this low-reflectance region, we set a transmittance threshold for each band. This threshold is defined as the minimum of (1) 40% of the maximum transmittance at each band or (2) the minimum transmittance value of each band. This ensures that the analysis focuses on the dynamic range of interest for reflectance where the simulation results are more reliable.

L142-145: Hard to follow 'Increased photon …' . Please rewrite, expand.

Thank you for pointing out the problem. We edit the sentences as below:

- Original text (Lines 142-145):
  "Increased photon path lengths from multiple scattering in 3D-RT produce non-zero perturbations (percentage differences in 1D and 3D radiances) expressed in Eq. (1). Since wavelengths with higher absorption are attenuated more than those with lower absorption, the Eq. (1) perturbations are a function of reflectance (line absorption depth), referred to later as spectral distortion."
- Revised text:
  "Multiple scattering in 3D-RT increases the photon path lengths, leading to non-zero perturbations, as expressed by Eq. (1) (percentage differences between 1D and 3D radiances). This effect is more pronounced at wavelengths with higher absorption, which are attenuated more strongly compared to wavelengths with lower absorption for

the same photon path. As a result, the perturbations vary depending on the reflectance and absorption depth, a phenomenon referred to as spectral distortion in our study."

L154: Why not use B11?

The B11 version was not publicly available when this study began, and we have encountered issues with retrieval code compilation for B11. Once these issues are resolved, we plan to update our analysis using B11 in future work. We plan to keep updating our algorithm, and switching to B11 will be one of these updates, to be documented in the next publication related to EaR$^3$T-OCO.

L244: 'To mitigate excessive computational demands, we opt to use solely the wavelengths of the first footprint.' -> how does this impact the results?

The primary difference among the eight footprints in the same swath is the instrument line shape (ILS), which can slightly influence the gas absorption optical depth. Although we have not yet run simulations with varying ILS, we expect the overall trends for the slope and intercept to remain similar. This is because the perturbations are calculated using the *differences* between footprint-level 1D-RT and 3D-RT simulations, where any variations due to ILS are irrelevant. Then, the spectral radiance perturbations are applied to other footprints, with slightly different ILS and dispersion. Since the un-perturbation based on the calculated perturbations operates in radiance (reflectance) space rather than wavelength space, small changes in spectral attributes of other footprints are not expected to have any significant impact on the algorithm. However, further analysis would be needed to confirm this. Relative to other factors such as cloud geometry, sun-sensor geometry, etc., this effect is most likely negligible. Again, this will be studied more thoroughly in forthcoming publications.

L262: how did the various reflectance thresholds influence the results.

The reflectance thresholds significantly impact cloud detection and subsequent radiance simulations. If the reflectance threshold is set too high, thin clouds may go undetected, resulting in an underestimation of cloud impact. Conversely, if the threshold is set too low, some bright surface pixels could be misclassified as clouds, leading to overestimating cloud effects. Both scenarios deviate from real conditions, making it difficult to accurately represent the 3D cloud effect and potentially bias the simulation results.

L263: why did you need to develop a new cloud detection approach?

The cloud products provided by MODIS have a spatial resolution of 1 km, which is too coarse for our simulation needs. To address this, we developed a new cloud detection approach to optimize detection specifically for the study case, ensuring higher accuracy in identifying clouds. While the method used by Chen et al. (2023) is more generalized

and suitable for a wide range of scenarios, it may miss some thin clouds. Our customized approach helps to better capture these thin cloud features, which is crucial for accurately modeling the 3D cloud effects in the selected scene.

L298: 'uncertainties' : keep in mind that the uncertainties in s, I, depend on many more factors than captured by the uncertainty in the line fit. Thus, you would underestimate the true uncertainties with that approach.

Thank you for the comment. Indeed, in Tables 1 and 2, we currently only quantify the uncertainties associated with the linear fit. We acknowledge that the true uncertainties are influenced by additional factors, such as variations in geometry, cloud properties, and aerosol characteristics. These contributions will be considered in future studies as we expand our analysis to include a wider range of conditions.

We will add a description in the table caption in response to your comment.

- Original text (line 433; lines 473-474; 503-504):
  "Table 1. Amplitude and e-folding distances for $s$ and $i$ fittings in the $O_2$-A, $WCO_2$, and $SCO_2$ bands."
  "Table 2. Amplitude and e-folding distances for $s$ and $i$ fittings of the" simulation with a homogeneous aerosol layer in the $O_2$-A, $WCO_2$, and $SCO_2$ bands."
  "Table 3. Amplitude and e-folding distances for $s$ and $i$, determined using different average grid points in simulations with a homogeneous aerosol layer for the $O_2$-A, $WCO_2$, and $SCO_2$ bands."
- Revised text, with the main changes underlined:
  "Table 1. Amplitude and e-folding distances for $s$ and $i$ fittings in the $O_2$-A, $WCO_2$, and $SCO_2$ bands. Errors represent fitting uncertainty only and may be underestimated."
  "Table 2. Amplitude and e-folding distances for $s$ and $i$ fittings of the" simulation with a homogeneous aerosol layer in the $O_2$-A, $WCO_2$, and $SCO_2$ bands. Errors represent fitting uncertainty only and may be underestimated."
  "Table 3. Amplitude and e-folding distances for $s$ and $i$, determined using different average grid points in simulations with a homogeneous aerosol layer for the $O_2$-A, $WCO_2$, and $SCO_2$ bands. Errors represent fitting uncertainty only and may be underestimated."

L307: what are those 'various processes'?

The retrieval accounts for various processes, including water vapor absorption, surface albedo variations, cloud and aerosol scattering in the 1D-RT model, radiance polarization effects, and the impact of the instrument line shape, among others.

L310: The code on Gituhub is not the code used for the operational retrieval.

Thank you for pointing out the issue. While the code on GitHub is not the same as the operational retrieval code, we have tested it using B10r (i.e., B10.0.04_sdos_testing_1) and obtained the same $X_{CO2}$ values as those in the L2 $X_{CO2}$ data. This confirms that the GitHub code is functionally equivalent to our analysis. A future version of our method could therefore easily be integrated into the data stream that uses the actual operational code (after further testing of our code with more data and updating it as needed, of course).

L320: explain terms in equation

We edit the sentence and add the description as below:

- Original text (Lines 142-145):
  "Upon deriving the 3D parameters in Section 4.3, we can convert the OCO-2 spectra using Eq. (4) with the observed radiance spectra and corresponding reflectance, slope, and intercept. Assuming the absence of 3D effects in the adjusted 1D radiance, we can employ the B10.04 retrieval algorithm with un-perturbed spectra to obtain mitigated $X_{CO2}$."
- Revised text, with the main changes underlined:
  "Upon deriving the 3D parameters in Section 4.3, we can convert the OCO-2 spectra using Eq. (4) with the observed radiance spectra ($I^{IPA}_\lambda$) and corresponding reflectance ($R^{obs}_\lambda$), slope ($s_{xy}$), and intercept ($i_{xy}$). Assuming the absence of 3D effects in the adjusted 1D radiance, we can employ the B10.04 retrieval algorithm with un-perturbed spectra to obtain mitigated $X_{CO2}$."

$$I^{IPA(adjusted)}_\lambda (x, y) = \frac{I^{IPA(obs)}_\lambda (x, y)}{\left( \frac{i_{xy} + s_{xy} \times R^{obs}_\lambda}{100\%} + 1 \right)}$$

L325: 'parameters that accurately represent'

We interpret this as a suggestion to clarify how the slope and intercept parameters are defined to represent the 3D cloud effect in OCO-2 observations accurately. The original sentence aims to convey that realistic radiance simulations near the satellite's footprint are essential for deriving these parameters precisely. Although achieving perfect accuracy in the simulations is challenging, we strive to approximate real conditions as closely as possible using MODIS observations. If further clarification was intended, we would appreciate any additional guidance.

- Original text (Lines 325-326):
  "In order to derive the slope (s) and intercept (i) parameters that accurately represent the 3D cloud effect in the OCO-2 observations, it is crucial to perform realistic radiance simulations near the satellite's footprint."

L330: Quantify 'shows a good agreement'

We will add an $R^2$ and slope for the scatter plot in the sentence.

- Original text (Line 330):
  "The heat map in Fig. 3c shows a good agreement between the simulation and observation."
- Revised text:
  "The heat map in Fig. 3c shows a good agreement between the simulation and observation with $R^2$=0.69 and a slope of 0.71."

L330: remove sentence 'As a result, …'

Thank you for the suggestion. We will remove Lines 330-331: "As a result, we are confident that the simulation at other wavelengths is able to approach the actual condition." as suggested.

L332: COT repeated twice

Thank you for pointing out the typo. One COT should be "CTH" instead.

Figure 3: How much was the COT and CER tuned to agree? Could we get the right answer for the wrong reason?

We currently apply 1D-RT reflectance-to-COT mapping and fixed CER values for low and high clouds, which can lead to biases and often overestimate COT for low radiance and underestimate it for high radiance. Accordingly, the 3D-RT cloudy pixel radiance will also differ from the observation, as shown in Fig. R9. However, since our primary focus is on the bright areas, we prioritize capturing the radiance differences over these regions rather than achieving perfect agreement for all cloudy pixels. We aim to capture the general trend in the bright areas, where minor uncertainties in cloud and aerosol setups are acceptable compared to the larger differences observed between 1D-RT and 3D-RT simulations.

However, it is important to understand how COT and CER change for the same cloud field can alter the 3D cloud effect magnitude. This is crucial to evaluate the result uncertainty. We will investigate the influence of the cloud properties in the future as well as more parameters. We appreciate your question on this topic.

[Figure]

Figure R9. A scatter comparison between Fig. 3a and 3b (Fig. R10 below) for pixels with COT greater than 0.

[Figure]

(edited Fig. 3) Figure R10. MODIS observation at 650 nm (a) and 3D radiance simulation at 650 nm by EaR³T (b). A scatter comparison between (a) and (b) is depicted in (c).

L341: how did you arrive a 11 wavelengths? What happens if you use 10 or 12? Aka, how sensitive are you to this choice? Would be a great opportunity to plot accuracy vs number of wavelengths.

The choice of 11 wavelengths was made as a compromise between computational cost and fitting accuracy. Using more wavelengths would indeed result in a better fit for the full-spectrum simulation. However, increasing the number of wavelengths significantly raises computational time and cost. We have not yet systematically evaluated the sensitivity of accuracy to the number of wavelengths, but it would be a valuable analysis to explore in future work, potentially plotting accuracy versus the number of selected wavelengths to determine the optimal balance. In addition, we plan to use ALIS (Emde et al., 2022) or the acceleration method by Iwabuchi instead of our multi-wavelength

method. Either of these might be even faster than our method, but we need to evaluate their accuracy.

Figure 5: how do the other bands look like? 5 a) looks very noisy far away from the clouds.

The observed noise far from the clouds in Figure 5a may be due to variations in surface reflectance across the region, which can affect the stability of the derived parameters. We will investigate this further to determine if additional filtering or adjustments are needed.

Here are similar figures for Fig. 5 but for $WCO_2$ and $SCO_2$ bands:

[Figure]

Figure R11. Distribution of (a) s and (b) i of WCO$_2$ band. Red dots denote the cloud pixels employed in the RT simulation.

[Figure]

Figure R12. Distribution of (a) s and (b) i of SCO$_2$ band. Red dots denote the cloud pixels employed in the RT simulation.

L463: 'bands, potentially increasing'

We interpret this as a potential suggestion to clarify how aerosols might impact the uncertainty in deriving the 3D effect parameters. The original sentence intends to highlight that aerosols can affect both the s and i of the O$_2$-A and SCO$_2$ bands. Accurate representation of aerosol optical depth (AOD) is crucial, as inaccuracies could lead to errors in capturing the 3D effects. If further clarification was intended, we would welcome additional guidance.

- Original text (Lines 462-464):
  "We illustrate that the presence of aerosols can lead to alterations in both the s and i of the O$_2$-A and SCO$_2$ bands, potentially increasing the uncertainty associated with the derivation of 3D effect parameters."

L483: state footprint sizes of upcoming satellites, name and cite those satellites

We will add the description below:

- Original text (Lines 482-484):
  "Numerous upcoming satellites for CO$_2$ remote sensing will adopt similar retrieval algorithms but feature varying footprint sizes in accordance with their specific mission objectives."

- Revised                                                                                              text:
  "Numerous upcoming satellites for $CO_2$ remote sensing will adopt similar retrieval algorithms but feature varying footprint sizes in accordance with their specific mission objectives. For example, the Copernicus Anthropogenic CO2 Monitoring Mission (CO2M) by the European Space Agency (ESA) plans to have a footprint size of 4 km$^2$ (2 km by 2 km; Kuhlmann et al., 2020). MicroCarb by the Centre National d'Etudes Spatiales (CNES) will have a larger footprint size of 40.5 km$^2$ (4.5 km by 9 km; Cansot et al., 2023)."

L487: why did you not investigate smaller footprints?

We did not investigate smaller footprints because the OCO-2/3 satellites currently offer the smallest footprint size available for $CO_2$ measurements among existing and upcoming satellite missions. However, newly proposed satellites for greenhouse gas remote sensing may feature smaller footprints, and we plan to consider this analysis in the future. We anticipate that 3D cloud effects could become more pronounced as footprint size decreases.

L490: 'of pronounced biases'

We remove the "pronounced" in the sentence.

- Original text (Lines 489-490):
  "This decline aligns with the expectation that larger footprints would mitigate the spectral distortion effect, reducing the prevalence of pronounced biases."
- Revised text:
  "This decline aligns with the expectation that larger footprints would mitigate the spectral distortion effect, reducing the magnitude of 3D cloud biases."

L490-491: not clear what changes are not significant

We edit the sentence as below:

- Original text (Lines 490-491):
  "Notably, there is no statistically significant change in $a_i$ and $d_i$ of the intercept."
- Revised text, with the main changes underlined:
  "There is no statistically significant change in $a_i$ and $d_i$ of the intercept values across different footprint sizes."

L495: 'In conclusion, future satellite missions with any …' That is a very strong statement without any quantification. This would depend on the retrieval algorithm, chosen bands, accuracy requirements, area of interest, …

We revise the statement as below:

- Original text (Lines 495-496):
  "In conclusion, future satellite missions with any footprint size must account for 3D biases to ensure accurate remote sensing of $X_{CO2}$."
- Revised text:
  "We suggest that future satellite missions, regardless of footprint size, consider accounting for 3D biases to improve the accuracy of $X_{CO2}$ retrievals. Studies need to be conducted to ensure that given the bands, footprint size, and other attributes, the retrieval error induced by 3D clouds does not exceed the respective mission requirements – as is the case for OCO-2, as this study has shown."

L500: quantify 'to substantial 3D'

We add the description as below:

- Original text (Lines 498-500):
  "Conversely, missions designed with smaller footprint sizes than OCO-2, particularly those targeting enhanced data acquisition in cloud-prone regions such as the Amazon Basin (Frankenberg et al., 2024) will be susceptible to substantial 3D cloud biases."
- Revised text, with the main changes underlined:
  "Conversely, missions designed with smaller footprint sizes than OCO-2, particularly those targeting enhanced data acquisition in cloud-prone regions such as the Amazon Basin (Frankenberg et al., 2024), will be susceptible to 3D cloud biases, which have been shown to reach -0.48 ppm for Land Nadir observations in both the northern and southern hemispheres (Massie et al., 2024)."

L501: why do 3D cloud biases need to be considered in the initial planning stage? Algorithms are typically tackled much later.

While algorithm development typically occurs at later stages, the experience from OCO-2/3 has shown that 3D cloud biases can significantly impact CO2 measurements, especially when targeting regions like the cloudy Amazon. With a decade of observations highlighting this issue, it is crucial to consider 3D cloud biases during the initial planning stages of new satellite missions—particularly for those aiming for smaller footprint sizes—so that the instrument design and mission parameters can be optimized to minimize these biases from the outset. The way this is typically done at the mission development stage is in mission or science traceability matrices (STM), which are part of every proposal. Any serious mission proposal of the future needs to show that they consider the impact of ubiquitous clouds on the derivation of $X_{CO2}$ or other trace gas products from the radiances when discussing expected uncertainties. This does not require ready-made algorithms, and merely needs to consider the literature (e.g., Massie, Mauceri, Emde, our own study).

L517: How could the bypass method deal with cloud shadows?

We don't believe that this is feasible or necessary at this point. As noted in the response on page 7, only a few cloud-shadow retrievals are present in the OCO-2 Lite files due to pre-retrieval cloud screening algorithms (Massie et al., 2023). Additionally, the B11 retrieval has improved at filtering out shadowed footprints. We believe that the few remaining cloud-shadow footprints passing the pre-screening are less significant compared to the clear-sky footprints affected by nearby clouds. However, applying the same radiance adjustment for bright areas to shadowed regions could introduce additional errors. Since footprints affected by cloud shadows constitute a relatively small portion of the overall effective OCO-2 retrievals, we believe that our approach provides a reasonable average correction for the majority of clear-sky and bright-area footprints.

L524: Quality Flag =0 or 1 are not 'best quallity' data. That would only by 0

Thank you for the comment. We edit the sentence and remove the description of "best quality" as below:

- Original text (Lines 523-524):
  "Subsequently, we determine the s and i of the three bands using the coefficients in Table 2 for all footprints that are the best quality (Quality Flag = 0 or 1) data points."
- Revised text:
  "Subsequently, we determine the s and i of the three bands using the coefficients in Table 2 for all footprints that pass the pre-screening (Quality Flag = 0 or 1) data points."

L524: How are the values in Table 2 derived for the bypass method when they don't include 3D RT calculations.

We want to clarify that values in Table 2 are derived using the baseline method, which involves 3D-RT calculations for specific scenarios. The bypass method is then parameterized based on these baseline results. Our approach is to analyze various solar and viewing geometries, as well as different cloud and aerosol properties, using the baseline method. Once we establish these relationships, we can derive a generalized bypass method that can be applied under a wide range of conditions without additional 3D-RT simulations.

Figure 9: Not sure if b) is improved compared to a) outside of the cloud shadow area.

Thank you for pointing out the concern. Fig. 9c shows that footprints over both the south and north sides of clouds have a decrease in $X_{CO_2}$. This means that footprints over clear-sky areas do reduce the 3D bias.

L570 – L573: You state a problem with thin or partial clouds for the bypass method. How would an operational algorithm deal with that?

Footprints containing thin or partial clouds pose additional challenges beyond 3D photon scattering, such as elevated reflectance at the continuum wavelength compared to clear-sky conditions. An operational algorithm would need to account for these effects by either incorporating additional parameters (e.g., cloud optical depth or cloud fraction) or using more complex correction models to differentiate between 3D scattering biases and direct cloud contamination. This would ensure that the bypass method remains effective in mixed or thin cloud conditions.

L585: remove 'on a cluster at the University of Colorado'

Thank you for the suggestion. We will change as suggested and leave the statement in the acknowledgments.

L592: You state that the bypass method can be supplemented by periodic full calculations. How would that work in detail? When do you run them, how do you use their results to improve the results?

The timing and frequency of performing full 3D-RT calculations would depend on how sensitive the bypass method's parameters are to changes in key state variables, such as solar and viewing geometry, cloud and aerosol properties, and surface albedo. The first step is to establish a generalized bypass method that captures the influence of these variables. If the derived parameters are found to be highly sensitive to changes in these conditions, then more frequent recalibration using full 3D-RT simulations would be necessary to maintain the accuracy of the bypass method.

In practice, this could involve periodically running full 3D-RT simulations for a subset of representative scenarios (e.g., different seasons, surface types, or cloud conditions) and updating the bypass parameterization accordingly. These recalibrated parameters would then be applied to operational retrievals, ensuring that the bypass method remains robust over time.

Figure 12. Where does the XCO2 in those scenes come from?

The result shown in Fig. 12 is derived using the bypass parameterization from Table 2. The goal of this figure is to illustrate how variations in cloud distribution can lead to different cloud-induced biases. We started with the OCO-2 radiance data presented in Figure 2 and applied radiance adjustments to introduce 3D cloud biases. This approach allows us to explore the impact of 3D cloud biases for different effective cloud distances and assess how cloud distribution influences the retrieved $X_{CO2}$ values.

L630: 'We documented the …' -> The main manuscript does not contain any documentation of the toolbox. Would remove that statement.

Thank you for the comments. We will remove Lines 630-631: "We documented the EaR³T-OCO radiance simulator, an automated tool for calculating spectral radiances observed by NASA's OCO-2 satellite." in the conclusion.

L671: 'more accurate level of accuracy'?

We revise the sentence to emphasize accuracy improvements near clouds:

- Original text (Line 670-672):
  "Our work can become the stepping stone toward more accurate and efficient trace gas retrievals even in complex scenes, ultimately bringing spaceborne trace gas retrievals to a more accurate level of accuracy."
- Revised text:
  "Our work can become the stepping stone toward more accurate and efficient trace gas retrievals even in complex scenes, ultimately bringing spaceborne trace gas retrievals in the vicinity of clouds to their planned accuracy."

L672: remove last sentence 'It will improve …' Your study did not show any information to draw that conclusion.

Thank you for the comment. We change the sentence as below:

- Original text (Lines 672-673):
  "It will improve current flux inversions (especially over the cloud-prone Amazon) and other applications."
- Revised text:
  "If implemented operationally, the bypass method has the potential to improve $X_{CO2}$ accuracy in cloud-prone areas, such as the Amazon, which could, in turn, enhance the accuracy of flux inversions."

L685: GitHub for OCO-2 toolbox leads to a 404 page not found

- Thank you for pointing this out. We will update the link from "https://github.com/ywchen-tw/OCO-2" to "https://github.com/ywchen-tw/OCO2" to resolve the issue.

Figure A3. Why is cloud effective radius only one fixed number for the whole scene?

Thank you for these questions, and we apologize for not making this clearer in the manuscript. For this study, we chose to keep the cloud effective radius (CER) constant for simplicity, which is why it appears uniform in Fig. A3c. Specifically, we assigned CER values of 10 µm for low clouds and 25 µm for high clouds. In future work, we intend to incorporate MODIS-derived CER values to better capture spatial variability.

In response to your other questions, we have clarified the description in line 265 as follows:

- Original text (Line 265):
  "Once the cloudy pixels are identified, we retrieve the cloud top height (CTH) and cloud effective radius (CER) of the nearest location from the MODIS MYD02 cloud file and assign them to each cloudy grid point."
- Revised text:
  "Once the cloudy pixels are identified, we retrieve the cloud top height (CTH) of the nearest location from the MODIS MYD02 cloud file and assign it to each cloudy grid point. The cloud effective radius (CER) is manually set to 10 μm for low clouds and 25 μm for high clouds in this study. In future versions, we plan to use the actual MODIS CER values to capture more realistic variations."

Figure A6. Would remove. There is not much information here beyond what one would expect.

Thank you for the suggestion. We will remove it as suggested.

References:

- Kuhlmann, G., Brunner, D., Broquet, G., and Meijer, Y.: Quantifying $CO_2$ emissions of a city with the Copernicus Anthropogenic $CO_2$ Monitoring satellite mission, Atmos. Meas. Tech., 13, 6733–6754, https://doi.org/10.5194/amt-13-6733-2020, 2020.
- Cansot, E., Pistre, L., Castelnau, M., Landiech, P., Georges, L., Gaeremynck, Y., and Bernard P.: MicroCarb instrument, overview and first results, Proc. SPIE 12777, International Conference on Space Optics — ICSO 2022, 1277734 (12 July 2023); https://doi.org/10.1117/12.269033.

---

## Author Response (AR1)

**Point-by-Point Response to RC1**

Dear reviewer,

Thank you for your insightful comments and constructive suggestions. We understand your concerns regarding the limited validation presented in the manuscript. The primary aim of this paper is to introduce new physics-based mitigation techniques for OCO retrievals, which differ fundamentally from existing statistics-based bias correction methods.

Advancement of this methodology and an expanded validation effort are planned for future work. In response to your feedback, we have included two additional case studies that apply the same bypass parameters and provide a comparison with the baseline method in this response document on page 4. Given the manuscript length, we included only the first case of these two samples in the appendix.

Your suggestion to streamline the abstract by focusing on the two primary mitigation strategies is well taken. We shortened the abstract as suggested while retaining the description of the EaR$^3$T-OCO setup within the manuscript. A detailed explanation of both the methodology and experimental setup is essential for clearly conveying the approach and demonstrating its application.

We notice your concerns about the applicability of our method in the presence of shadows. Since the OCO retrieval algorithm pre-screens most cloud-shadow footprints over land, we focus our mitigation efforts on clear-sky and non-shadow footprints. These footprints constitute most nadir observations and are therefore more critical for accurate retrievals than cloud-shadow footprints. A detailed response addressing this concern is provided in the following statements.

Below, we provide detailed responses to each point raised. The comments and suggestions are highlighted in red, responses are in blue, and references to the original manuscript content appear in black.

General comments:

**Comment 01:** The paper presents a step forward to mitigate biases of retrieved CO2 concentrations from satellite observations (OCO-2/3) due to scattering of radiation by clouds in clear regions. 3D scattering effects are approximated by a linear fit of the radiances against selected wavelengths of a given spectral window.

**Response 01:** Thank you for this summary, which accurately captures our work.

**Comment 02:** Based on another ongoing work, the authors claim that 3D scattering effects for a specific scene can be described by the slope and the intercept of the fit. These two parameters can be used to adjust the cloud contaminated observations to corresponding clear sky spectra.

**Response 02:** To illustrate, we reference Fig. 2 from Schmidt et al. (2019) (references at the end of this document), which shows three distinct types of footprints with the 3D cloud effect. The characteristic correlation pattern forms three branches: clear-sky pixels with cloud in the vicinity (green), footprints in cloud shadow (blue) and cloudy pixels (red). For OCO-2/3 land nadir footprints, our analysis focuses on the clear-sky pixels (green), where we apply a linear fit to derive the slope and intercept parameters.

[Figure]

Figure R1. (Fig 2 from Schmidt et al., 2019) Reflectance enhancement (spectral perturbation) as a function of the reflectance for three distinct types of footprints.

**Comment 03:** In a following step an exponential fit of the slope and the intercept against the effective distance to clouds is performed so that the adjustment of the spectra can be done without 3D radiative transfer (3D RT) simulations for each pixel. In order to avoid 3D RT calculations completely the so-called bypass method is also presented, which uses only observational data to obtain the slope and the intercept as a function of distance from the cloud and "other scene parameters" (the second is not shown in the paper).

**Response 03:** In our study, we utilize observational data from MODIS, including cloud and aerosol properties, to better capture the true radiance distribution. However, deriving the slope and intercept requires direct comparisons between 1D and 3D simulations. Without these simulations, it would not be possible to accurately determine the relationship between radiance and cloud distance. Based on the case shown in this study, we also simulate the radiance for different solar zenith angles, surface albedo, and aerosol optical depth to evaluate the impact of these scene parameters on 3D cloud effects.

**Comment 04:** The basic method using the 3DRT simulations requires cloud input data (cloud optical thickness, effective radius and cloud top height from MODIS observations), surface albedo (also from MODIS) and several assumptions (which are not fully described). It is rather difficult to obtain the realistic setup and therefore not straightforward to obtain the slope and intercept parameters.

**Response 04:** The realistic setup for a radiance simulation is indeed always difficult. However, the short time difference between OCO-2 and MODIS-Aqua makes it more straightforward to build up the simulation environment. Since OCO-2 and MODIS Aqua are on the same orbit with only a 6-minute time difference, the MODIS observations provide a highly realistic environmental setup for OCO-2, ensuring a high degree of consistency between the two datasets. Some future missions, such as CO2M, also aim to have an imager directly alongside the spectrometer to obtain more simultaneous cloud information. Therefore, we think our approach is useful.

   While we obtain cloud optical thickness, effective radius, cloud top height, surface albedo, and aerosol properties from MODIS, some additional assumptions are necessary to complete the 3D radiative transfer (3D-RT) simulations. Specifically, MODIS does not provide cloud geometric thickness or the mixing layer height for aerosols, which requires us to make reasonable assumptions based on typical values and the literature. To make the assumptions clear, we added the following sentences in the simulation setup: "This study makes the assumption of fixed cloud geometric thickness (1 km for cloud top height smaller than 4 km and cloud base at 3 km for cloud top height greater than 4 km), which could lead to some uncertainties in the RT simulations. Further investigation of the impact of cloud properties on the 3D cloud effect is needed in the future."

**Comment 05:** The bypass method, which derives the parameters only from the observed radiances would be more practicable and it does not rely on other retrieval algorithms and assumptions on input to RT simulations, therefore this is in my opinion a promising approach.

**Response 05:** However, we want to clarify that the bypass method is a parameterized *version* of the baseline method, which derives from the result of the (more complicated) RT simulations. That is to say, the baseline method will be more accurate, but also more time-consuming compared to the bypass method. While we aim to eventually apply the bypass method *independently* to OCO-2 footprints without the need for additional RT simulations, further refinement is required. At this stage, we need to perform more RT simulations using the baseline method to build a robust, generalized bypass model that can reliably capture a wide range of scenarios.

**Comment 06:** The methods are demonstrated for a particular scene, showing the retrieved CO2 concentration of the original OCO retrieval versus the adapted retrieval. For this particular case I see that the adapted retrieval gives smaller CO2 concentrations (up to 6ppm) in the cloud shadow (for which the correction is not designed for as mentioned in the text). For in-scattering, there is almost no difference as far as I can see in Fig.9. The corrected results are similar for both methods, the baseline and the bypass approaches. The truth is not known, and the authors therefore can not prove that they achieve an improvement. A way to validate the approach would be to use synthetic observations with known input concentration.

**Response 06:** As mentioned in lines 596-598 in the revised manuscript, we refer to Massie et al. (2023), who found that relatively few cloud shadow retrievals exist in the OCO-2 Lite files due to the pre-retrieval cloud screening algorithms. Footprints affected by cloud shadows present additional complexities beyond the typical 3D cloud bias, such as significant surface albedo variations and reduced $CO_2$ absorption, which complicate the analysis further. Our current study therefore focuses exclusively on mitigating 3D cloud biases in areas with cloud-induced enhanced illumination, rather than addressing the complexities of cloud shadow regions. We agree that demonstrating the method's applicability to a broader range of conditions, such as different solar zenith angles, surface types, cloud types, and viewing geometries (nadir and glint), is crucial for generalization. This will be an essential direction for future research to ensure that the method can be reliably extended to diverse scenarios.

To clarify the results in Figure 9, we have included Figure 9c, which highlights the differences after applying the mitigation strategies. We change the scale of the color bar to make the difference clearer as below:

[Figure]

(edited Fig. 9) Figure R2. Replicated from Fig. 9 in the manuscript, with an updated color bar scale in panel (c) adjusted to -2 to 2 ppm to better highlight $\Delta X_{CO2}$.

We also want to state again that the bypass method is the parameterization version of the baseline method. As a result, the corrected results are similar for this case. Regarding the "unknown truth" issue, we acknowledge the limitations of using real observations without a reference. This challenge is common in OCO-2 3D cloud effect research. To address this, we followed established approaches from previous studies (Massie et al., 2021, 2022; Mauceri et al., 2023) by comparing footprints near and far from clouds, using those farther from clouds as a proxy for unbiased $X_{CO2}$ values. The suggestion to validate using synthetic observations is certainly well taken, but not the emphasis of this particular study, which focused on real-world data instead of synthetic data. Besides, such a study has previously been conducted by Emde et al. (2022). Of course, when working with satellite data, there is no ground truth data. However, the anomalies in Fig. R2a suggest that the $X_{CO2}$ in the vicinity of clouds is too high. This is also suggested by a companion publication (Schmidt et al., 2019, Figure 3, included below), which shows that positive slopes of the kind shown in Figure R1 (green) in the vicinity of clouds lead to positive $X_{CO2}$ bias. Furthermore, we show that our method removes at least part of this positive bias (negative values near clouds in Figure R2c.

It is, technically, possible to directly compare OCO-2 retrievals with data from the ground stations at TCCON. However, this would require the use of target mode, which has its own limitations (here, we focused on nadir-only observations). In the future, we plan to look at target observations as well.

[Figure]

Figure R3. (Adapted from Fig. 3 from Schmidt et al., 2019) Dependence of the $X_{CO2}$ retrieval when applying perturbations to the strong or weak $CO_2$ band spectra. Positive perturbations in either band introduce a positive retrieval bias in $X_{CO2}$, while negative perturbations result in an underestimation of $X_{CO2}$.

**Comment 07:** Certainly it is important to correct cloud effects in trace gas retrievals, therefore I think that the topic of the paper is appropriate for AMT. However, more validation is needed and I suggest to revise the paper in this respect. Further, several clarifications regarding the setup of the methods are required, for example it is not explained how cloud shadows are excluded.

**Response 07:** We agree that additional validation is necessary before our method can be applied operationally. As we explained previously, the primary objective of this paper is to introduce the methodology we developed and to demonstrate its potential effectiveness. We plan to conduct more extensive validation studies in the

future, incorporating a broader range of cases, including both land and ocean observations. Please also see our response to the previous point: Aside from TCOON (not applicable here), satellite retrievals of $X_{CO2}$ (and many other geophysical products) only rarely come with ground truth. But looking at Figure R2, it is obvious that the near-cloud bias is at least significantly reduced by our method. As mentioned above, we plan to extend our study to target mode observations in the future, at which point we can use TCOON for a direct validation of our method.

To address your comment on validation to some degree, we have included two additional cases from the same month and general geographic area as the case in Fig. 2. These cases apply the bypass mitigation method based on parameters outlined in Table R1 (Table 2 in the manuscript) and are compared to the baseline method as validation examples (shown below). 1D-RT and 3D-RT simulations were conducted for these two cases to derive the correct slope and intercept parameters. Thus, we can evaluate the bypass mitigation based on Table R1, with the comparisons illustrated in Figs. R4 and R7.

The results from case 1 indicate that the bypass method yields a mitigation trend similar to that of the baseline method, although with lower magnitudes (i.e., not as accurate as the baseline method). This could be due to differences in surface altitude and albedo, solar geometry, AOD, and other environmental factors. This case demonstrates promising results, yet adjustments to the bypass parameters with more scene variables are necessary for effective operational use. For now, we therefore recommend using the baseline method.

In contrast, case 2 shows less favorable performance of the bypass method compared to the baseline method. In case 2, the baseline method reveals a weaker correlation between $\Delta X_{CO2}$ and effective cloud distance, likely due to confounding factors, such as surface albedo effects. This indicates that the bypass method may be less effective in mixed or complex cloud conditions. Given the length constraints of the current manuscript, we have decided to add only case 1 as an example in the appendix.

Regarding your concern about footprints under cloud shadows, we do not exclude footprints that fall under cloud shadows. Instead, we apply the radiance adjustment to all footprints that pass the quality test. In other words, if the retrieval algorithm's pre-screening does not exclude a footprint—even if it is located in a cloud shadow—our radiance adjustment will still be applied, despite the method being primarily designed for bright areas.

Table R1. The same table as Table 2 in the manuscript. Amplitude and e-folding distances for $s$ and $i$ fittings of the simulation with a homogeneous aerosol layer in the $O_2$-A, $WCO_2$, and $SCO_2$ bands for 1.0 km geometric cloud thickness of low clouds.

| | Slope | | | Intercept | | |
|---|---|---|---|---|---|---|
| | $s_{O_2-A}$ | $s_{WCO_2}$ | $s_{SCO_2}$ | $i_{O_2-A}$ | $i_{WCO_2}$ | $i_{SCO_2}$ |
| $a_s$ or $a_i$ | $0.457 \pm 0.094$ | $0.123 \pm 0.037$ | $0.250 \pm 0.041$ | $0.755 \pm 0.327$ | $0.648 \pm 0.227$ | $0.847 \pm 0.406$ |
| $d_s$ or $d_i$ (km) | $3.82 \pm 0.44$ | $5.04 \pm 0.89$ | $4.58 \pm 0.78$ | $2.69 \pm 0.32$ | $2.91 \pm 0.31$ | $2.35 \pm 0.33$ |

Additional case 1:

[Figure]

Figure R3. Satellite true-color imagery of MODIS Aqua from NASA Worldview on 5 October 2019 with (a) $X_{CO2}$ in OCO-2 level 2 data, (b) mitigated $X_{CO2}$ retrieved from the adjusted spectra and (c) difference between the mitigated and original $X_{CO2}$ values.

[Figure]

Figure R4. (a) Relationship of $\Delta X_{CO2}$ with $D_e$ based on parameterized slopes and intercepts from the bypass method in Table 2. (b) Corresponding relationship using slopes and intercepts derived from the baseline approach for Fig. R3.

[Figure]

Figure R5. (a) Scatter plot comparing the $X_{CO2}$ anomaly of the OCO-2 L2 product (in black) to its value post-spectra adjustment (in red) for the case shown in the figure above, plotted against $D_e$. The $X_{CO2}$ anomaly is defined as retrieved $X_{CO2}$ – true $X_{CO2}$, with the true $X_{CO2}$ defined by the average $X_{CO2}$ of footprints with a $D_e$ greater than 15 km (405.96 ppm in this case). The orange shade indicates the 1 ppm mission requirement. (b) Histograms and probability density functions (PDFs) for the $X_{CO2}$ anomaly of the OCO-2 L2 product (in black) and post spectra adjustment (in red) within a 15 km $D_e$. This corresponds to the blue-shaded region in (a). The FWHM values of the PDFs of v10r and adjusted data points are 5.25 and 4.28, and the PDF averages are 0.93 and 0.18, respectively. The average change in $X_{CO2}$ after the spectra adjustment for De less than 15 km is -0.86 ppm.

Additional case 2:

[Figure]

Figure R6. Satellite true-color imagery of MODIS Aqua from NASA Worldview on 18 October 2018 with (a) $X_{CO2}$ in OCO-2 level 2 data, (b) mitigated $X_{CO2}$ retrieved from the adjusted spectra and (c) difference between the mitigated and original $X_{CO2}$ values.

[Figure]

Figure R7. (a) Relationship of $\Delta X_{CO2}$ with $D_e$ based on parameterized slopes and intercepts from the bypass method in Table 2. (b) Corresponding relationship using slopes and intercepts derived from the baseline approach for Fig. R6.

[Figure]

Figure R8. (a) Scatter plot comparing the $X_{CO2}$ anomaly of the OCO-2 L2 product (in black) to its value post-spectra adjustment (in red) for the case shown in the figure above, plotted against $D_e$. The $X_{CO2}$ anomaly is defined as retrieved $X_{CO2}$ – true $X_{CO2}$, with the true $X_{CO2}$ defined by the average $X_{CO2}$ of footprints with a $D_e$ greater than 15 km (405.69 ppm in this case). The orange shade indicates the 1 ppm mission requirement. (b)

Histograms and probability density functions (PDFs) for the $X_{CO_2}$ anomaly of the OCO-2 L2 product (in black) and post spectra adjustment (in red) within a 15 km $D_e$. This corresponds to the blue-shaded region in (a). The FWHM values of the PDFs of v10r and adjusted data points are 10.24 and 8.80, and the PDF averages are 0.27 and 0.20, respectively. The average change in $X_{CO_2}$ after the spectra adjustment for De less than 15 km is -0.45 ppm.

**Changes in manuscript:** The discussion contents have been added to the original manuscript. Please refer to lines 942-948 and Fig. A7 to A9 of the revised manuscript.

Specific comments:

**Comment 08:** - The abstract is relatively long, it could be focused a more on the two mitigation strategies and technical details about EaR3T-OCO could be shortened

**Response 08:** Thank you for the suggestions. We revised the abstract to make it more concise by focusing on the two mitigation strategies and moving the technical details about EaR³T-OCO to the methodology and appendix, as recommended.

- Revised abstract (Lines 16-33 in the revised manuscript):

"Accurate and continuous measurements of atmospheric carbon dioxide ($CO_2$) are essential for climate change research and monitoring of emission reduction efforts. NASA's Orbiting Carbon Observatory (OCO-2/3) satellites have been deployed to infer the column-averaged $CO_2$ dry air mixing ratio (XCO2) from passive spectroscopy, with a designed uncertainty of less than one ppm for the regional average. This accuracy is often not met in cloudy regions because clouds in the vicinity of a footprint introduce biases in the $X_{CO_2}$ retrievals. These arise from limitations in the one-dimensional (1D) forward radiative transfer (RT) model, which does not capture the spectral radiance perturbations introduced by clouds adjacent to a clear footprint. Our paper introduces a three-dimensional (3D) RT pipeline to explicitly account for these effects in real-world satellite observations. This is done by ingesting collocated imagery and reanalysis products to calculate the cloud-induced perturbations at the footprint level. To make that computationally feasible, a simple approximation for their spectral dependence is used. The calculated perturbations are then used to reverse (undo) the cloud vicinity effects at the radiance level, at which point the standard 1D OCO-2 retrieval code can be applied without modifications. For two cases over land, we demonstrate that this approach indeed reduces the $X_{CO_2}$ anomalies near clouds. We also characterize the dependence of the XCO2 footprint-level bias on the distance from clouds and other key scene parameters, such as surface reflectance. Although this dependence may be specific to cloud type, aerosols, and other factors, we illustrate how it could be parameterized to bypass our physics-based 3D-RT pipeline for use in an operational framework. In the future, we intend to explore this possibility by applying our tool to a variety of scenes over land and ocean."

**Comment 09:** - l.24 "These biases, referred to as the three-dimensional (3D) effects, can be quantified effectively using 3D-RT calculations, but these are computationally expensive, especially for hyperspectral applications (e.g., OCO-2/3)."

The authors refer later to the ALIS method for spectral Monte Carlo simulations. It is true that 3D RT is generally expensive but with ALIS spectral simulations are almost as fast as monochromatic simulations (see ALIS, Emde et al., 2011)

**Response 09:** We agree that techniques like ALIS are valuable advancements for accelerating 3D-RT calculations, making spectral simulations more efficient. Iwabuchi also developed a spectral acceleration technique. In this work, we chose our own, less sophisticated, acceleration method, but the main emphasis of

the work was not on acceleration, and more on application to real-world data. In the future, we envision to leave it to the user to use either ALIS, or the Iwabuchi method, or ours, to correct real-world spaceborne spectroscopy data. In addition, another focus of this work is to demonstrate that the bypass method has the potential to operationally mitigate 3D biases *without* the need for running 3D-RT simulations at all, provided sufficient cloud information is available (e.g., effective cloud distance).

**Comment 10:** - l.44 "For the case we analyzed, both the 3D-RT calculation method and the parametric bypass method successfully corrected XCO2 biases, which exceeded 2 ppm at the footprint level, and reached up to 0.7 ppm in the regional average."

How do you know the the correction is successful, you get a difference but you do not know the true CO2 concentration?

An approach to validate the retrieval is to use synthetic data, as shown in the cited work by Kylling et al. 2022 for TROPOMI. A more systematic study on the mitigation of cloud scattering for TROPOMI is shown in Hu et al. 2022, who use 2D RT simulations to derive fits to correct retrieved airmass factors and validate this approach using realistic synthetic data based on Large Eddy simulations. The synthetic data (Emde et al. 2022) they are using includes also $O_2A$ band spectra and could possibly also be used to validate the mitigation approach for $CO_2$ retrievals.

**Response 10:** Determining true $CO_2$ concentrations without simultaneous in-situ measurements, such as airborne observations, poses a significant challenge. Observations from TCCON stations offer an alternative for validating the mitigation method, though precise nadir views at these stations are very uncommon because of the relative locations of the TCCON stations relative to the sunsynchronous satellite orbits. This "unknown truth" issue is a common limitation in OCO-2 3D cloud effect studies due to the scarcity of ground-based and airborne observations available for validation. Consequently, many studies compare footprints near and far from clouds, using those farther from clouds as reference values (Massie et al., 2021, 2022; Mauceri et al., 2023). We use a similar approach, considering footprints with larger effective cloud distances as being relatively free of cloud bias.

We agree that synthetic data is a valuable tool for validating satellite retrieval algorithms. However, it is nearly impossible to perfectly replicate all atmospheric interactions exactly as they occur in the real atmosphere. Therefore, testing the mitigation approach on real observations remains essential for capturing the complexities of actual atmospheric conditions. In addition, Figure R2 (and others) does show that our bias correction goes into the right direction. Quantifying this with ground truth will need to be left for further studies that include target observations and TCCON stations (as discussed in response to an earlier point). Nonetheless, we will keep using synthetic data as we have in previous work, as suggested. Again, the exclusive use of synthetic data only cannot replace the application to real-world data, where unforeseen effects such as sensor performance, unknown aerosol layers, geolocation inaccuracies etc. may play a role that cannot be captured by synthetic observations only. Our own synthetic data (e.g., from Schmidt et al., 2019) convinced us that we had to go to real-world data next, and that is the primary focus of *this* manuscript.

**Comment 11:** - l. 46 "We find that the biases depend most strongly on the cloud field morphology and surface reflectance, but also on secondary factors such as aerosol layers and sun-sensor geometry."

I assume with cloud field morphology, you mean the weighted distance to the clouds. The paper does not show how the bias depends on surface reflectance and on sun-sensor geometry. The impact of cloud scattering will certainly increase with increasing solar zenith angle but also for slant viewing angles (here only nadir view is shown). Also the cloud geometrical thickness is probably important.

**Response 11:** By "cloud field morphology," we refer to the distribution of clouds, including cloud types, cloud top height (CTH), cloud base height, and cloud optical thickness (COT), all of which can influence the magnitude of the 3D cloud effect.

We agree that solar and viewing geometry also play a significant role in determining the extent of these biases. We acknowledge that the impact of other geometrical factors, such as solar zenith angle, is not explicitly shown in the current paper. We added a section before Section 5.5 (3D effect mitigation) discussing the impact of surface albedo and solar zenith angle (SZA). Additionally, we plan to conduct a more extensive investigation in future work to encompass these and other relevant factors.

To illustrate the effect of solar zenith angle and surface albedo, we provide parameterization figures for the $O_2$-A band across different conditions:

[Figure]

Figure R9. Parameterization of (a) slope and (b) intercept for $O_2$-A band with effective cloud distance, varied by solar zenith angle, while holding surface albedo and aerosol optical depth (AOD) constant.

[Figure]

Figure R10. Parameterization of (a) slope and (b) intercept for $O_2$-A band with effective cloud distance, varied by surface albedo, while holding solar zenith angle and AOD constant.

Combining these results across different solar zenith angles and surface albedo values allows us to develop a two-variable linear parameterization of $a_s$ and $d_s$ (slope parameters) and $a_i$ and $d_i$ (intercept parameters). As summarized in Table R1 below, we find that the amplitude of slope and intercept is inversely proportional to surface albedo and directly proportional to the cosine of SZA (denoted as $\mu$). Additionally, the e-folding

distances of the slope are negatively proportional to both surface albedo and $\mu$, while those of the intercept are positively proportional to surface albedo and negatively proportional to $\mu$.

Table R1. The parameterization of $a_s$ and $d_s$ of slope and $a_i$ and $d_i$ of intercept for the three OCO-2 bands.

|  | slope | intercept |
|---|---|---|
| $O_2$-A | $a_s = -0.34 \times alb_{O2\text{-}A} + 0.57 \times \mu - 0.03$
 $d_s = -3.2 \times alb_{O2\text{-}A} - 9.9 \times \mu + 14.9$ | $a_i = -0.60 \times alb_{O2\text{-}A} + 0.36 \times \mu + 0.72$
 $d_i = 0.42 \times alb_{O2\text{-}A} - 2.1 \times \mu + 5.2$ |
| $WCO_2$ | $a_s = -0.15 \times alb_{wCO2} + 0.11 \times \mu - 0.05$
 $d_s = -30.7 \times alb_{wCO2} - 7.0 \times \mu + 27.5$ | $a_i = -2.07 \times alb_{wCO2} + 1.65 \times \mu + 1.17$
 $d_i = 0.63 \times alb_{wCO2} - 1.6 \times \mu + 3.7$ |
| $SCO_2$ | $a_s = -0.18 \times alb_{SCO2} + 0.29 \times \mu - 0.03$
 $d_s = -22.6 \times alb_{SCO2} - 21.2 \times \mu + 34.9$ | $a_i = -2.77 \times alb_{SCO2} + 2.22 \times \mu + 1.14$
 $d_i = 0.51 \times alb_{SCO2} - 1.73 \times \mu + 3.35$ |

**Comment 12:** - Eq.1: Remove 100%, because 100%=1. Or multiply by 100 and say that the unit of the perturbation is in per cent.

**Response 12:** We removed 100% in Eq. 1 to simplify the expression, as the perturbation will then be represented in unit form.

**Comment 13:** Fig.1: Is the fitted line obtained by fitting the blue dots or the grey dots? You should show that both fits result in the same slope and intercept (you could include the two fitting lines and the corresponding equations).

**Response 13:** Thank you for the suggestion. We modified Fig. 1 and have both fitting lines as shown below:

[Figure]

(edited version of Fig. 1) Figure R11. Example of the linear relationship between perturbation and reflectance. The grey dots represent the complete wavelength range, while the red dots indicate the subset selected for the $O_2$-A band simulation. The black and red lines represent the linear fit of the grey and red dots, respectively.

**Comment 14:** l. 146: "The intercept is related to the often-reported increase of reflectance near clouds, or decrease in shadows, whereas the slope accounts for spectroscopic effects."

This interpretation is correct as long as the spectral dependence of scattering can be neglected which is true for small spectral windows.

**Response 14:** We added the description in the sentence that this is true when the spectral window is small, which can be applied to OCO spectral windows as below:

●       Revised text (Lines 302-304 in the revised manuscript), with the main changes underlined:

"The intercept is related to the often-reported increase of reflectance near clouds, or decrease in shadows, whereas the slope accounts for spectroscopic effects for a small spectral range, where the scattering effect can be considered spectral independent."

**Comment 15:** Fig.2: It looks as if the main differences between the retrievals are in the cloud shadow region. Could you also show the image without the CO2 concentration included to see whether there is a cloud shadow at the place with higher CO2 concentrations?

**Response 15:** Here are the MODIS-Aqua images (a) without and (b) with OCO-2 footprints overlapped. Note that there is a six-minute time difference between the two satellites. Again, we want to emphasize that footprints over cloud shadows are out of the scope of this research.

[Figure]

[Figure]

Figure R12. Satellite true-color imagery of MODIS Aqua from NASA Worldview on 18 October 2018, (a) without and (b) with OCO-2 retrieved $X_{CO_2}$ overlaid.

**Comment 16:** l. 267: "To determine the cloud optical thickness (COT) of each pixel, we run the RT model over several COT and derive the COT-radiance relationship by ourselves to ensure the radiance consistency in 1D-RT simulation."

This is not so clear. It means you do not use the cloud optical thickness as from the MODIS retrieval algorithm but an adjusted optical thickness that is needed as input for the 1D RT simulation to be consistent with the observed radiance? Isn't this exactly the same as the MODIS retrieval?

The retrieved optical thickness is of course biased by 3D effects because generally the reflectance is smaller in 3D simulations compared to 1D for the same vertical optical thickness due to photon leakage on the cloud sides. That means that the COT is underestimated, but this is not what you mean here?

**Response 16:** Converting radiance to COT follows the method detailed in Appendix C2 of Chen et al. (2023). We do not use the MODIS COT directly because the MODIS cloud identification algorithm may overlook small, isolated clouds, leading to significant radiance discrepancies for nearby OCO footprints. Instead, we apply an optimized cloud detection radiance threshold tailored for each scene and then perform the radiance-to-COT mapping accordingly.

Regarding the concern about the underestimation of COT, we acknowledge that using 1D-RT-based radiance-to-COT mapping can result in biases, where COT may indeed be overestimated for low radiances and underestimated for high radiances due to the 3D effects. Since our primary area of interest is in bright regions, we focus on minimizing radiance discrepancies over these areas rather than on correcting the biases at cloudy pixels. In essence, the clouds in our calculations serve as source of additional diffuse illumination, which is not as strongly dependent on COT than, say, the reflectance. Still, we acknowledge that we may be under- or overestimating the amount of diffuse illumination. Therefore, in the future, integrating COT correction techniques could further refine COT mapping. For instance, Nataraja et al. (2022) developed a neural network-based COT correction using 650 nm reflectance, which could be a potential approach to incorporate.

**Comment 17:** l.281: "MCARaTS iteratively traces the path of each photon and calculates the distribution of photons based on the final probability."

What do you mean with "distribution of photons" and "final probability"?

**Response 17:** MCARaTS traces each of the photons ($10^9$ photons for this study) and records their paths within the simulation domain to determine where they are absorbed, scattered, or transmitted. This process results in a distribution of photon interactions across the simulation area. By dividing the number of photons reaching a particular region by the total number of photons simulated, we obtain a probability distribution, which is then used to calculate the resulting radiance.

**Comment 18:** l. 287: "The mean radiance and the standard deviation are then calculated from three runs to estimate the uncertainty."

Three samples are not sufficient to estimate the standard deviation. Why not running more simulations with less photons to get a better estimate?

**Response 18:** We agree that using only three runs may not provide a robust estimate of the standard deviation. However, due to the high computational cost of each MCARaTS simulation, we opted for three runs with a larger number of photons ($10^9$ per run) to ensure stable mean radiance values. Running more simulations with fewer photons could increase the noise, affecting the reliability of the radiance estimates. In future work, we will explore optimizing the balance between the number of photons per run and the number of runs to achieve a more precise estimate of the standard deviation while maintaining computational efficiency.

**Comment 19:** Eq. 4: Remove 100% which is equal to 1.

**Response 19:** We modified Eq. 4 as suggested.

**Comment 20:** l. 328: "Fig. 3a-b presents the 3D-RT simulation and MODIS observation of 650 nm using the COT, CER, and CTH shown in Fig. A3."

It is not fully clear how the cloud input is created from the MODIS data.

Why is CER seems constant (looks like this in Fig. A3)? Please provide the value of CER. How is the cloud vertically constructed? What is the cloud base height? What is the sun-observer geometry?

**Response 20:** We apologize for not making this clearer in the manuscript. For this study, we chose to keep the cloud effective radius (CER) constant for simplicity, which is why it appears uniform in Fig. A3c. Specifically, we assigned CER values of 10 μm for low clouds and 25 μm for high clouds. In future work, we intend to incorporate MODIS-derived CER values to better capture spatial variability.

Regarding cloud structure and geometry, clouds are vertically constructed with cloud top height (CTH) derived from MODIS data, as shown in Fig. A3a. The cloud base height (CBH) is set to 1 km for low clouds and 3 km for higher clouds. The sun-observer geometry is matched to OCO-2 observation conditions, using an average viewing angle of 0.31˚ and an SZA of 48.5˚ across all OCO-2 footprints in the area of interest.

In response to your other questions, we have clarified the description in line 265 as follows:

●     Original text (Line 265):
"Once the cloudy pixels are identified, we retrieve the cloud top height (CTH) and cloud effective radius

(CER) of the nearest location from the MODIS MYD02 cloud file and assign them to each cloudy grid point."

● Revised text (Lines 1430-1433 in the revised manuscript):
"Once the cloudy pixels are identified, we retrieve the cloud top height (CTH) of the nearest location from the MODIS MYD02 cloud file and assign it to each cloudy grid point. The cloud effective radius (CER) is manually set to 10 μm for low clouds and 25 μm for high clouds in this study. In future versions, we plan to use the actual MODIS CER values to capture more realistic variations."

**Comment 21:** Can the method also be applied for ice clouds? Is it valid over ocean?

**Response 21:** We have not yet tested the method on ice clouds, so its applicability in such cases remains uncertain. Over ocean surfaces, OCO-2 is operated in glint mode, not in nadir mode. We only studied the nadir mode in this manuscript. In glint, the $CO_2$ bias behaves differently due to factors such as ocean glint reflection and specific scattering processes. Our preliminary result (not in this manuscript) shows that the bypass method may also work for ocean cases. However, additional investigations are needed to adapt and validate the method for oceanic conditions.

**Comment 22:** How is the spectral albedo generated from MODIS data? A dataset and a method to generate hyperspectral surface albedo data from MODIS data is presented in Roccetti et al. 2024, could this be included in your model?

**Response 22:** For the surface albedo in this study, we used both the *"brdf_reflectance"* from the OCO-2 Level 2 data and the MODIS MCD43A3 data. The spatial distribution of surface albedo over the area of interest is derived from the MCD43A3 product and then scaled using the OCO-2 BRDF reflectance values. Currently, we assume a wavelength-independent surface albedo within each band range. Incorporating hyperspectral surface albedo data, such as that presented in Roccetti et al. (2024), is an excellent suggestion. Unfortunately the publication was not available when we developed our method. However, we expect that it will enhance the accuracy of our EaR[3]T-OCO simulator, particularly for surface types with significant spectral variability over small wavelength ranges. We plan on implementing this improvement in future versions of the model. Thank you for the excellent suggestion!

**Comment 23:** l. 330: "The heat map in Fig. 3c shows a good agreement between the simulation and observation. As a result, we are confident that the simulation at other wavelengths is able to approach the actual condition."

More tests needed to draw this conclusion. How do other bands compare? At least one image in NIR should be shown.

**Response 23:** To address your request, we have included the simulations for all three channels overlaid with OCO footprint observations, as shown below:

[Figure]

Figure R13. Simulated continuum radiance overlaid with OCO-2 observed radiance (c) for the (a) $O_2$-A, (b) $WCO_2$, and (c) $SCO_2$ band.

**Comment 24:** Most of the points in the image correspond to clear sky. How is the correlation for the cloudy pixels only? This would show better whether the cloud input is realistic.

**Response 24:** We recognize that our radiance-to-COT mapping, which is based on 1D-RT, can lead to an overestimation of COT for low radiance values and an underestimation for high radiance values due to 3D effects. For example, comparing the cloudy pixels between our simulation and observation presents an underestimated radiance. Since our primary focus is on bright areas, where radiance values are typically higher, we prioritize minimizing radiance discrepancies in these regions rather than at the cloudy pixels. As mentioned above, integrating COT correction techniques could further refine COT mapping in the future.

[Figure]

Figure R14. A scatter comparison between Fig. 3a and 3b (Fig. R15 below) for pixels with COT greater than 0.

**Comment 25:** I think "scatter plot" is a better name than "heat map". Could you also include a colorbar?

**Response 25:** Thank you for your suggestion. We updated the Fig.3 as below with changing the heatmap to scatterplot and adding color bar:

[Figure]

(edited Fig. 3) Figure R15. MODIS observation at 650 nm (a) and 3D radiance simulation at 650 nm by EaR$^3$T (b). A scatter comparison between (a) and (b) is depicted in (c).

**Comment 26:** l. 345: "Employing a reduced number of wavelengths, uniformly distributed across the reflectance space, effectively minimizes computational demands while still permitting the derivation of **s** and **i** for the linear relationships within each band."

How are the wavelengths selected? Do you use the same set of wavelengths for all scenes?

**Response 26:** We describe the wavelength selection approach in Lines 340-343. For each band, we first calculate the average transmittance of the region of interest using representative values for solar and viewing geometries, surface albedo, and surface height. To avoid running simulations at wavelengths with extremely low transmittance, we define a minimum threshold for each band, set as the lower of (1) 40% of the band's maximum transmittance or (2) the band's minimum transmittance. As a result, the selected wavelengths for different scenes could be different.

Once the applicable wavelengths are identified, we sort them by transmittance and select a fixed number (e.g., 11) that are uniformly distributed within the transmittance space, as shown in Fig. 4a. This selection ensures that we capture the key spectral variations while minimizing computational cost. The same method is applied for each scene, but the exact set of wavelengths may vary slightly depending on the scene-specific atmospheric conditions.

●     Original text (Lines 340-343):
"To balance computational demands with accuracy, we selected 11 wavelengths evenly distributed over the high 60% transmittance based on sorted clear-sky transmittance for further RT simulation (depicted in Fig. 4 as an example, the transmittance of full spectra is presented in Fig. A4). The transmittance is calculated based on the atmosphere profile, $\theta_s$, etc."

**Comment 27:** Have you tested whether this relationship remains linear in different cloud situations? Clouds can shield the lower atmosphere so that due to the presence of clouds the amount of CO2 absorption is significantly decreased. I would expect non-linearity effects and would like to understand better why this relation should always be linear.

**Response 27:** Thank you for the question. We have observed that when reflectance becomes too low, the linearity of the relationship can deviate significantly. In other words, the main linear part would start from a certain reflectance, the mechanism of which remains unclear. This is one of the reasons why we established a transmittance threshold to exclude wavelengths associated with extremely low reflectance, thereby minimizing non-linear effects.

In cloud-shadowed areas, clouds can shield the lower atmosphere and reduce the total reflected radiance, introducing a negative intercept term. The ratio of $CO_2$ absorption to the total reflected radiance would remain relatively similar to clear-sky conditions, though the slope behaves differently under shadowed conditions. In some cloud-shadowed areas, however, we also observe non-linear behavior, which will require further investigation to fully understand. Nevertheless, our study primarily focuses on bright areas where the linear relationship remains more stable. The dynamics within cloudier or shadowed regions fall outside the scope of this analysis and would require a separate investigation to address potential non-linearities fully.

**Comment 28:** l. 365: 1km² -> 1.25km²?

**Response 28:** The area is 1 km² because the 5x5 grid points correspond to a 1 km x 1 km region, with each side of the grid representing 1 km in length.

**Comment 29:** l. 366: "We excluded the data if the 25 nearest grid points contained cloud pixels used in the RT simulation."

Does this mean you consider only completely clear sky pixels? What about partially cloudy pixels for which CO2 retrievals are also performed, if the cloud fraction is not too high?

**Response 29:** Partially cloudy footprints can pass the pre-screening process, but they are much fewer in number compared to clear-sky footprints close to clouds. In this paper, we focus primarily on footprints located in clear-sky regions to isolate the impact of 3D cloud scattering. Partially cloudy footprints are more complex, as they can be affected by additional factors beyond scattering, such as cloud fraction and sub-pixel cloud variability. Addressing these would require further investigation, which is outside the scope of this study.

**Comment 30:** l. 381: "Though it is instructive to discuss both cloud brightening and cloud shadowing effects, Massie et al. (2023) determined that there are relatively few cloud shadow retrievals in the OCO-2 Lite files, since many observations impacted by shadowing are screened by the OCO-2 pre-retrieval cloud screening algorithms. Thereafter, bright area analyses are the primary focus of our study."

As mentioned before, in the specific scene that you present it seems that you obtain largest differences in a cloud shadow region?

**Response 30:** While some large negative-bias footprints do indeed appear in shadowed regions, the majority of positive-bias footprints are located in clear-sky areas. In our investigations, we observed that the version 11 dataset more effectively screens out these negative-bias footprints compared to version 10. We will move toward version 11 retrieval in the future. Footprints under cloud shadows encounter more complexities beyond cloud-induced photon scattering, such as surface albedo variations and reduced $CO_2$ absorption, which are not fully addressed by our current bright area mitigation approach. As a result, applying this method to cloud-shadowed regions may unintentionally worsen the bias. Our study focuses on mitigating biases in bright, clear-sky regions, while addressing cloud-shadowed areas would require a distinct and targeted strategy.

**Comment 31:** l. 387: "we identified an exponential decay relationship between the 3D cloud effect parameters and the effective horizontal cloud distance ($D$, Fig. 6)"

What is the "bright area" (mentioned in the caption of Fig. 6) in the images, how do you select which is bright area and which is shadow?

**Response 31:** The "bright area" is defined as the region where the reflectance calculated from the 3D-RT model is greater than that from the 1D-RT model. These areas show enhanced reflectance due to cloud-induced brightening effects. In contrast, regions where the reflectance is lower than that from the 1D-RT model are classified as cloud shadows, indicating areas where clouds reduce the reflected radiance.

**Comment 32:** I assume that this exponential decay is only valid in the bright area, because in the shadow there are abrubt changes in reflectance where the shadow ends. Is this the reason, why you say that your method is focused on the bright area?

**Response 32:** The exponential decay pattern can also be observed in shadow areas to some extent, but the relationship behaves in the opposite direction, with a decrease in reflectance instead of brightening. Our retrieval method is not optimized for footprints in shadowed regions, as these areas present additional challenges, including severe changes in reflectance at shadow boundaries, as you mentioned. These complexities would require further investigation to address effectively, so we focus primarily on bright areas in this study.

**Comment 33:** In Fig. 6 I don't see the "background shading". Could you include colors corresponding to the density of the dots as in Fig.3?

**Response 33:** Thank you for pointing out the issue. We understand that the density shading might be confusing. We replaced the density kernel with the density scatter plot as below:

[Figure]

(edited Fig. 6) Figure R16. The same figure as Fig. 6 but changing from heatmap to scatter plot.

**Comment 34:** Table 1: These parameters are valid only for the specific scene, correct? This should be clarified. Is the number of digits meaningful?

**Response 34:** Yes, the parameters in Table 1 are derived specifically from the results shown in Figures 5 and 6. Considering the uncertainty, the number of digits is not meaningful. We updated the table description to make this clearer as below:

●     Original text (Line 433):
"Table 1. Amplitude and e-folding distances for *s* and *i* fittings in the $O_2$-A, $WCO_2$, and $SCO_2$ bands."

- Revised text (Lines 659-660 in the revised manuscript):

"Table 1. Amplitude and e-folding distances for $s$ and $i$ fittings in the $O_2$-A, $WCO_2$, and $SCO_2$ bands for the simulation shown in Fig. 5."

**Comment 35:** l. 516: "However, the bypass method is less precise than conducting a 3D-RT simulation with our baseline approach to derive s and i on a pixel-by-pixel basis."

Why is it less precise? I think it could even be more precise because it does not rely on assumptions to produce the input for the 3DRT simulations.

**Response 35:** As mentioned earlier, the bypass method is a parameterized approach derived from our baseline method, which itself relies on 3D-RT simulation results. Therefore, it still inherits the assumptions made in the baseline method. While the bypass method offers a computational advantage, its precision can be lower because it uses generalized relationships rather than performing pixel-specific 3D-RT calculations, which can capture more localized variations.

**Comment 36:** l. 517: "This bypass approach also disregards the presence of cloud shadows."

Why can the bypass approach not account for the shadows? Shadows are included in the observed radiances.

**Response 36:** While it's true that shadows are present in the observed radiances, footprints over cloud-shadowed regions introduce additional complexities beyond the 3D cloud bias that the bypass method is designed to address. Adapting the bypass method to accurately handle shadowed regions would require further development and validation, which lies outside the scope of this study. Additionally, footprints over cloud shadows are rare in the retrieved land $X_{CO2}$. To avoid potential confusion, we have removed the discussion of cloud shadows from Section 5.6.

**Comment 37:** Fig. 12: Are the results shown here based on the basic approach or the bypass approach? There are many shadows in the images. Do you include a cloud mask to exclude the shadows or how are they treated?

**Response 37:** The results shown in Fig. 12 are based on the bypass parameterization listed in Table 2. The purpose of this figure is to demonstrate how variations in cloud distribution can lead to different cloud-induced biases. For this analysis, we consider the effective cloud distance for non-cloudy pixels, but we do not apply a cloud mask to exclude shadowed areas. As a result, cloud shadows are included in the images, but shadow-induced biases are not specifically addressed. Investigating these biases would require a more detailed analysis, which is beyond the scope of this study. However, we decided to remove Fig. 12 and its related discussion to streamline Section 5.6 and focus exclusively on the mitigation demonstration.

**Comment 38:** l. 647: "... it allows the parameterization of the six spectral perturbation parameters themselves as a function of macroscopic scene parameters ..."

Which are the macroscopic scene parameters?

**Response 38:** The macroscopic scene parameters include solar and viewing geometry (e.g., solar zenith angle, viewing angle), cloud properties (e.g., cloud top height, cloud optical thickness), aerosol properties, and surface albedo. For example, we have demonstrated how variations in surface albedo can influence the spectral perturbation parameters in our analysis.

**Comment 39:** Couldn't one use only the spectral radiance observations to obtain the perturbation parameters using the bypass method?

**Response 39:** We have attempted to retrieve the perturbation parameters directly from the spectral radiance observations, but the results have not been satisfactory so far. We believe this is due to an insufficient degree of freedom for independently retrieving all six perturbation parameters, as they are highly coupled with other factors such as surface albedo and aerosol optical thickness. Decoupling these variables would require additional constraints or assumptions, which complicates the retrieval process.

**Comment 40:** l. 656: "While the bypass method does capture the significant modulators of the 3D cloud effects, including surface reflectance and sun-sensor geometry, it is not granular enough to consider detailed scene variables such as cloud top height, cloud morphology, or aerosol load."

Shouldn't the bypass method, since it uses reflectances containing all 3D cloud effects, capture all modulators?

**Response 40:** While the bypass method indeed uses reflectance that inherently captures all 3D cloud effects, it simplifies the variation in perturbation parameters by grouping them based solely on effective cloud distances. This simplification does not fully account for the spread in perturbation values that may arise due to more detailed factors, such as variations in cloud top height, cloud morphology, or aerosol distributions, which are not explicitly parameterized in the bypass method.

**References**:

Schmidt, K. S., Massie, S., and Feingold, G., 2019, June. Impact of Broken Clouds on Trace Gas Spectroscopy From Low Earth Orbit. In Hyperspectral Imaging and Sounding of the Environment (Optica Publishing Group, 2019), paper HW5C-2.

Chen, H., Schmidt, K. S., Massie, S. T., Nataraja, V., Norgren, M. S., Gristey, J. J., Feingold, G., Holz, R. E., and Iwabuchi, H.: The Education and Research 3D Radiative Transfer Toolbox (EaR3T) – towards the mitigation of 3D bias in airborne and spaceborne passive imagery cloud retrievals, Atmos. Meas. Tech., 16, 1971–2000, https://doi.org/10.5194/amt-16-1971-2023, 2023.

Nataraja, V., Schmidt, S., Chen, H., Yamaguchi, T., Kazil, J., Feingold, G., Wolf, K., and Iwabuchi, H.: Segmentation-based multi-pixel cloud optical thickness retrieval using a convolutional neural network, Atmos. Meas. Tech., 15, 5181–5205, https://doi.org/10.5194/amt-15-5181-2022, 2022.

Roccetti, G., Bugliaro, L., Gödde, F., Emde, C., Hamann, U., Manev, M., Sterzik, M. F., and Wehrum, C.: Development of a HAMSTER: Hyperspectral Albedo Maps dataset with high Spatial and TEmporal Resolution, EGUsphere [preprint], https://doi.org/10.5194/egusphere-2024-167, 2024.

Emde, C., Yu, H., Kylling, A., van Roozendael, M., Stebel, K., Veihelmann, B., and Mayer, B.: Impact of 3D cloud structures on the atmospheric trace gas products from UV–Vis sounders – Part 1: Synthetic dataset for validation of trace gas retrieval algorithms, Atmos. Meas. Tech., 15, 1587–1608, https://doi.org/10.5194/amt-15-1587-2022, 2022.

Yu, H., Emde, C., Kylling, A., Veihelmann, B., Mayer, B., Stebel, K., and Van Roozendael, M.: Impact of 3D cloud structures on the atmospheric trace gas products from UV–Vis sounders – Part 2: Impact on NO2 retrieval and mitigation strategies, Atmos. Meas. Tech., 15, 5743–5768, https://doi.org/10.5194/amt-15-5743-2022, 2022.

**Point-by-Point Response to RC2**

Dear reviewer,

We appreciate your insightful comments and suggestions. Below, we provide detailed responses to each point raised. In this document, the reviewer's comments and suggestions are highlighted in red, our responses are in blue, and references to the original manuscript content are in black.

We understand your concerns regarding the limited validation in the manuscript, particularly the inclusion of only one case study. While additional cases would indeed strengthen the validation for operational use, the primary aim of this paper is to introduce and demonstrate the mitigation approach we have developed. This work serves as an introduction to new physics-based mitigation methods for OCO retrievals, distinct from existing statistical bias correction methods.

We appreciate your suggestion to focus more on the results. However, as this paper introduces new methods, we believe it is essential to provide a detailed description of the methodology and setup, in addition to demonstrating the methods. In response to your comment, we have revised the description of the bypass method to better illustrate its application in quantifying the 3D cloud effect. Additionally, we have moved the simulation settings to the appendix to help emphasize the results. Again, the emphasis of this manuscript is to bring a bias-correction to real-world data, and that requires the introduction of a rather extensive radiative transfer tool.

In response to the feedback, we included two additional cases applying the same bypass parameters and a comparison with the baseline method in this response document on the next page. Due to manuscript length considerations, we added the first case as an example in the appendix. We remain committed to enhancing this methodology and will continue to expand validation in future work.

**Comment 01:** Overall, the subject of the study is very compelling and a significant contribution to the community. Especially considering future upcoming green house gas missions.

**Response 01:** Thank you for your positive feedback on our research. We appreciate your recognition of its relevance for future missions. We have carefully considered your comments and provided detailed responses to address your concerns.

**Comment 02:** However, the study misses depth in how far the 3D cloud bias correction has been investigated.

**Response 02:** We acknowledge the extensive efforts made by the community to address 3D cloud bias, such as recent advancements using machine learning approaches (e.g., applying the machine learning bias correction from Mauceri et al. (2023) on V11 data product). However, the methods adopted by the OCO team so are not fully physics-based, which means that they do not operate at the radiance level (where cloud-induced 3D-RT effects introduce perturbations), instead focusing on the product level in a statistical manner. Here, we operate at the radiance *and* footprint level for real-world satellite spectroscopy data – to our knowledge the first time that this has been done. The most closely related publications to our study are the ones by Emde et al. (2022) and Merrelli et al. (2015) since they studied the impact of cloud-induced spectroscopic perturbations on the products. However, neither of these studies worked with real-world data, focusing on synthetic data instead. The value of our study lies in presenting a mitigation strategy that directly addresses the 3D cloud bias from a physical (radiance) perspective, focusing on the mechanisms behind these biases rather than relying solely on empirical corrections (as done in previous, statistical studies). In this sense, we would actually say that our manuscript *adds* depth over previous studies. We might have missed what the reviewer is referring to specifically, perhaps details of the current retrieval algorithm. However, since it is stated in the manuscript, we

probably do not need to point out here that the physics of the current operational algorithm does not account for horizontal photon transport and therefore by definition misses an important piece of reality, which we bring back with our work.

**Comment 03: Major concern is that the developed approach has been applied only to a single, hand-picked scene. This is simply not enough to make any guesses towards the performance of the approach when applied operationally.**

**Response 03:** We understand where this concern is coming from, and are glad that the reviewer is pointing out this perception. Indeed, this paper starts out with a specific scene, and we acknowledge that more cases are needed to fully evaluate the method's performance for operational use. However, the primary goal of this study is to describe the methodology, and in that regard, the scene we used is simply used for illustration. We did not specifically hand-pick a scene because 'it worked." In reality, we have tried this method on several scenes, and the selected case simply provided a clear illustration to demonstrate the mechanics of the approach. However, this does not mean that the other cases did *not* work. To show this, we include two other cases in this response. Because of considerations regarding manuscript length, we added only the first additional case as an example in the appendix. In future work, we plan to apply the method to a larger set of scenes to assess its robustness and operational applicability, but that future work is distinct from our intention here, which lays the ground work and describes the methodology. On this note: Future version of the algorithm will come with improvements, part of which were motivated by the reviewer comments to the release version of our code, documented with this publication.

To address your comment on validation, we have included two additional cases from the same month and general geographic area as the case in Fig. 2. These cases apply the bypass mitigation method based on parameters outlined in Table R1 (Table 2 in the manuscript) and are compared to the baseline method as validation examples (shown below). 1D-RT and 3D-RT simulations were conducted for these two cases to derive the correct slope and intercept parameters. Thus, we can evaluate the bypass mitigation based on Table R1, with the comparisons illustrated in Figs. R3 and R6.

The results from case 1 indicate that the bypass method yields a mitigation trend similar to that of the baseline method, although with lower magnitudes. The difference between Fig. R2a and b is potentially due to differences in surface altitude and albedo, solar geometry, AOD, and other environmental factors. This case demonstrates promising results, yet adjustments to the bypass parameters with more scene variables are necessary for effective operational use.

In contrast, case 2 shows less favorable performance of the bypass method compared to the baseline method. In case 2, the baseline method reveals a weaker correlation between $\Delta X_{CO2}$ and effective cloud distance, likely due to confounding factors, such as surface albedo effects. This indicates that the bypass method may be less effective in mixed or complex cloud-surface conditions. Given the length constraints of the current manuscript, we have decided to add only case 1 as an example in the appendix.

Table R1. The same table as Table 2 in the manuscript. Amplitude and e-folding distances for **s** and **i** fittings of the simulation with a homogeneous aerosol layer in the $O_2$-A, $WCO_2$, and $SCO_2$ bands for 1.0 km geometric cloud thickness of low clouds.

| | Slope | | | Intercept | | |
|---|---|---|---|---|---|---|
| | $s_{O_2-A}$ | $s_{WCO_2}$ | $s_{SCO_2}$ | $i_{O_2-A}$ | $i_{WCO_2}$ | $i_{SCO_2}$ |

| $a_s$ or $a_i$ | $0.457 \pm 0.094$ | $0.123 \pm 0.037$ | $0.250 \pm 0.041$ | $0.755 \pm 0.327$ | $0.648 \pm 0.227$ | $0.847 \pm 0.406$ |
|---|---|---|---|---|---|---|
| $d_s$ or $d_i$ (km) | $3.82 \pm 0.44$ | $5.04 \pm 0.89$ | $4.58 \pm 0.78$ | $2.69 \pm 0.32$ | $2.91 \pm 0.31$ | $2.35 \pm 0.33$ |

Additional case 1:

[Figure]

Figure R1. Satellite true-color imagery of MODIS Aqua from NASA Worldview on 5 October 2019 with (a) $X_{CO2}$ in OCO-2 level 2 data, (b) mitigated $X_{CO2}$ retrieved from the adjusted spectra and (c) difference between the mitigated and original $X_{CO2}$ values.

[Figure]

Figure R2. (a) Relationship of $\Delta X_{CO2}$ with $D_e$ based on parameterized slopes and intercepts from the bypass method in Table 2. (b) Corresponding relationship using slopes and intercepts derived from the baseline approach for Fig. R3.

[Figure]

Figure R3. (a) Scatter plot comparing the $X_{CO2}$ anomaly of the OCO-2 L2 product (in black) to its value post-spectra adjustment (in red) for the case shown in the figure above, plotted against $D_e$. The $X_{CO2}$ anomaly is defined as retrieved $X_{CO2}$ – true $X_{CO2}$, with the true $X_{CO2}$ defined by the average $X_{CO2}$ of footprints with a $D_e$ greater than 15 km (405.96 ppm in this case). The orange shade indicates the 1 ppm mission requirement. (b) Histograms and probability density functions (PDFs) for the $X_{CO2}$ anomaly of the OCO-2 L2 product (in black) and post spectra adjustment (in red) within a 15 km $D_e$. This corresponds to the blue-shaded region in (a). The FWHM values of the PDFs of v10r and adjusted data points are 5.25 and 4.28, and the PDF averages are 0.93 and 0.18, respectively. The average change in $X_{CO2}$ after the spectra adjustment for De less than 15 km is -0.86 ppm.

Additional case 2:

[Figure]

Figure R4. Satellite true-color imagery of MODIS Aqua from NASA Worldview on 18 October 2018 with (a) $X_{CO2}$ in OCO-2 level 2 data, (b) mitigated $X_{CO2}$ retrieved from the adjusted spectra and (c) difference between the mitigated and original $X_{CO2}$ values.

[Figure]

Figure R5. (a) Relationship of $\Delta X_{CO2}$ with $D_e$ based on parameterized slopes and intercepts from the bypass method in Table 2. (b) Corresponding relationship using slopes and intercepts derived from the baseline approach for Fig. R5.

[Figure]

Figure R6. (a) Scatter plot comparing the $X_{CO2}$ anomaly of the OCO-2 L2 product (in black) to its value post-spectra adjustment (in red) for the case shown in the figure above, plotted against $D_e$. The $X_{CO2}$ anomaly is defined as retrieved $X_{CO2}$ – true $X_{CO2}$, with the true $X_{CO2}$ defined by the average $X_{CO2}$ of footprints with a $D_e$ greater than 15 km (405.69 ppm in this case). The orange shade indicates the 1 ppm mission requirement. (b)

Histograms and probability density functions (PDFs) for the $X_{CO2}$ anomaly of the OCO-2 L2 product (in black) and post spectra adjustment (in red) within a 15 km $D_e$. This corresponds to the blue-shaded region in (a). The FWHM values of the PDFs of v10r and adjusted data points are 10.24 and 8.80, and the PDF averages are 0.27 and 0.20, respectively. The average change in $X_{CO2}$ after the spectra adjustment for De less than 15 km is -0.45 ppm.

**Changes in manuscript:** The discussion contents have been added to the original manuscript. Please refer to lines 942-948 and Fig. A7 to A9 of the revised manuscript.

**Comment 04:** The study states that it developed a software tool for the automated calculation of spectral radiances from OCO-2. However, the automation is not exploited to analyze a representative sample size of OCO-2 observations.

**Response 04:** We agree that analyzing the entire dataset, or even 1% of OCO-2 observations, using full 3D-RT simulations is impractical due to the high computational cost. This constraint motivated the development of the bypass method, which aims to significantly reduce the need for extensive 3D-RT calculations. However, before the bypass method can be applied operationally, we still need to use the tool to analyze several dozen to hundreds of cases under diverse conditions to build a more generalized parameterization. While the automation feature helps streamline radiance calculations, further validation with additional cases is a key focus for future work. We are currently in the method development stage, with larger-scale case analyses planned as the next step.

**Comment 05:** Furthermore, for this single selected scene the strongest biases seem to be collocated with cloud shadows while the authors argue that those shadows are outside the scope of this study. For this research to be useful to the community it needs to show that it can be generalized (e.g. various SZA, ocean (where 3D cloud effects are strongest), land surface types, different cloud types, different viewing modes (nadir and glint)).

**Response 05:** As mentioned in lines 381-384, we refer to Massie et al. (2023), who found that relatively few cloud shadow retrievals exist in the OCO-2 Lite files due to the pre-retrieval cloud screening algorithms. In addition, B11 retrieval has improved in filtering out footprints under shadow (at least for the cases we analyzed). We don't think that the remaining few cloud-shadow footprints passing the pre-screening have the same importance as the clear-sky footprints affected by clouds in the vicinity.

We agree that demonstrating the method's applicability to a broader range of conditions, such as different solar zenith angles, surface types, cloud types, and viewing geometries (nadir and glint), is crucial for generalization. To address part of your concern, we added a section before Section 5.5 (3D effect mitigation) discussing the impact of surface albedo (related to surface types) and solar zenith angle (SZA).

Fig. R7 and R8 present the exponential decay fitting of the slope and intercept of the O2-A band under SZA and surface albedo. The x-axes are the effective horizontal cloud distance ($D_e$), which is defined as the average distance of the pixel to the surrounding cloudy pixels, weighted by the inverse square distance to the cloudy pixel (Eq. R1, Eq. 5 in the manuscript):

$$D_e = \frac{\sum_{i \in \{\text{surrounding clouds}\}} w_i D_i}{\sum_{i \in \{\text{surrounding clouds}\}} w_i} \quad (R1)$$

The exponential decay relationships in Fig R7-8 are fitted between slope ($s$) and intercept ($i$) parameters and $D_e$ using Eq. R2-3 (Eq. 6-7 in the manuscript). The amplitude ($a_s$, $a_i$) and e-folding distance ($d_s$, $d_i$) are the fitting parameters (separate sets for $s$ and $i$). The result of amplitude and e-folding distance are presented in Table R1.

$$s = a_s \times \exp(-\frac{D_e}{d_s}) \text{ (R2)}$$

$$i = a_i \times \exp(-\frac{D_e}{d_i}) \text{ (R3)}$$

We have tested these approaches for ocean glint cases and plan to have the next paper discussing specific ocean cases since their biases behave differently than land nadir cases. However, EaR$^3$T-OCO is already capable of simulating glint cases. The impact of cloud types and properties will be studied in the future.

●      Additional text to be added in new Section 5.5 (Lines 770-782 in the revised manuscript):

"Solar zenith angle and surface albedo are significant factors influencing the 3D cloud effect, represented by a parameterized set of 12 relationships. To investigate their impact on the 3D cloud effect, we also kept several variables constant, including COT, CER, CTH, cloud position, AOD, and atmospheric conditions, as the setup used in Section 5.3, and manually changed the SZA and surface reflectance. Figures R9 and R10 (will be added in the appendix) illustrate how these variables impact the 3D cloud effect in the $O_2$-A band under different conditions. Combining results across various solar zenith angles and surface albedo values, we developed a two-variable linear parameterization using $a_s$ and $d_s$ (slope parameters) and $a_i$ and $d_i$ (intercept parameters). As summarized in Table R1, we observe that the amplitude of the slope and intercept is inversely proportional to surface albedo and directly proportional to the cosine of the solar zenith angle (denoted as μ). Additionally, the e-folding distances of the slope ($d_s$) and are negatively proportional to both surface albedo and μ, while those of the intercept are positively proportional to surface albedo and negatively proportional to μ. In general, higher surface albedo reduces the 3D cloud effect, as additional photons reaching the sensor represent a smaller fraction of the total signal. Conversely, lower solar zenith angles result in a smaller amplitude but longer e-folding distance, causing the 3D effect to extend further from the clouds."

[Figure]

Figure R7. Parameterization of (a) slope and (b) intercept for $O_2$-A band with effective cloud distance, varied by solar zenith angle, while holding surface albedo and aerosol optical depth (AOD) constant.

[Figure]

Figure R8. Parameterization of (a) slope and (b) intercept for O2-A band with effective cloud distance, varied by surface albedo, while holding solar zenith angle and AOD constant.

Table R1. The parameterization of as and ds of slope and ai and di of intercept for the three OCO-2 bands. Errors represent fitting uncertainty only and may be underestimated.

| | slope | intercept |
|---|---|---|
| $O_2$-A | $a_s = -0.34 \times alb_{O2\text{-}A} + 0.57 \times \mu - 0.03$
 $d_s = -3.2 \times alb_{O2\text{-}A} - 9.9 \times \mu + 14.9$ | $a_i = -0.60 \times alb_{O2\text{-}A} + 0.36 \times \mu + 0.72$
 $d_i = 0.42 \times alb_{O2\text{-}A} - 2.1 \times \mu + 5.2$ |
| $WCO_2$ | $a_s = -0.15 \times alb_{WCO2} + 0.11 \times \mu - 0.05$
 $d_s = -30.7 \times alb_{WCO2} - 7.0 \times \mu + 27.5$ | $a_i = -2.07 \times alb_{WCO2} + 1.65 \times \mu + 1.17$
 $d_i = 0.63 \times alb_{WCO2} - 1.6 \times \mu + 3.7$ |
| $SCO_2$ | $a_s = -0.18 \times alb_{SCO2} + 0.29 \times \mu - 0.03$
 $d_s = -22.6 \times alb_{SCO2} - 21.2 \times \mu + 34.9$ | $a_i = -2.77 \times alb_{SCO2} + 2.22 \times \mu + 1.14$
 $d_i = 0.51 \times alb_{SCO2} - 1.73 \times \mu + 3.35$ |

**Changes in manuscript:** We have added a new section discussing the above contents to the original manuscript. Please refer to lines 770-782 and Table 4 of the revised manuscript.

**Comment 06:** The study currently reads more like a description of the work that was performed rather than being focused on the outcome of the work. The outcome is what your reader is interested in. I would suggest picking either the 3D cloud correction based on the 3RT simulations or the 'bypass' method as the outcome of this work and explore the chosen method further (explore more scenes to better estimate performance once applied operationally).

**Response 06:** We appreciate this input. However, this manuscript is intended to introduce a new method, and therefore intentionally reads like a description of the algorithm. Of course, we also present results, but we cannot talk about results without fully introducing the method first.  Also, the two methods (baseline: full 3D-RT; bypass: parameterization based on 3D-RT) are strongly related, and it is difficult to elaborate on the bypass method without describing the baseline method. Since this is the first paper discussing the new method, we think it is important to describe the approaches and their settings in detail as well. To address this, we edited the introduction to describe the bypass method as a way to quantify 3D cloud effects – see below. In addition,

we shortened the main part of the manuscript by moving the simulation settings to the appendix and streamlining the overall description.

- Original text (lines 80-88) for Introduction in Section 1:

"Schmidt et al. (2024) explain that lateral photon transport can be understood as missing physics in the operational OCO algorithm, and any adjustments for discrepancies between 1D-RT and 3D-RT could introduce additional inaccuracies in $X_{CO2}$ retrieval. Although advances have been made in expediting high-resolution 3D-RT simulations by using the same photon paths for various wavelengths (Emde et al., 2011; Iwabuchi and Okamura, 2017), the computational demands of such models have still hindered their operational application. Schmidt et al. (2024) introduced the 3D-RT radiance perturbation as the percentage difference between the 3D and 1D radiance simulations. This radiance perturbation is found to be linear over the relevant dynamic range of reflectance, which allows a simple representation of the perturbation as slope and intercept for each of the three OCO-2 bands. The details will be described in Section 2."

- Revised text for Introduction in Section 1, with the main changes underlined:

"Schmidt et al. (2019) explain that lateral photon transport represents missing physics in the operational OCO algorithm, and any adjustments for differences between 1D-RT and 3D-RT could introduce additional inaccuracies in $X_{CO2}$ retrieval. Evaluating these differences requires a high-resolution 3D-RT model capable of simulating spectra across multiple wavelengths. Recent advancements have accelerated high-resolution 3D-RT simulations for multi-wavelength applications. For instance, Partain et al. (2000) introduced an enhanced implementation of the equivalence theorem, which decouples scattering and absorption calculations, allowing for accurate spectral integration without repeated multiple-scattering computations for Monte Carlo models. Emde et al. (2011) developed the Absorption Lines Importance Sampling (ALIS) technique, which efficiently computes high-resolution polarized spectra by leveraging Monte Carlo photon tracing across multiple wavelengths simultaneously. Iwabuchi and Okamura (2017) also adopted a similar way of using the same photon paths for various wavelengths to accelerate multi-wavelength 3D-RT simulation. Doicu et al. (2020) accelerated the Spherical Harmonics Discrete Ordinate Method (SHDOM) 3D-RT model, which is different from Monte-Carlo-based 3D radiative transfer models, by combining the correlated k-distribution method with dimensionality reduction techniques, such as principal component analysis.

While these acceleration methods have the potential to improve the accuracy of trace gas retrievals by taking into account missing physics (horizontal photon transport), current operational retrievals still do not use true 3D-RT in trace gas retrieval processes. Here, we adopt the approach introduced by Schmidt et al. (2019) as a practical method to approximate the 1D-RT and 3D-RT differences in spectral radiance observations, building on the concept of 3D-RT radiance perturbations, defined as the spectral percentage difference between 3D and 1D radiance simulations. These simulations in this study are performed by a modified version of the Education and Research 3D Radiative Transfer Toolbox (EaR³T; Chen et al., 2023), tailored specifically for OCO (EaR³T-OCO). The 3D perturbations proved to be a linear function of the radiance (or reflectance) itself across the relevant dynamic range of reflectance, which allows its representation by a simple slope and intercept for each of the three OCO-2 bands, with further details provided in Section 2. We also introduce a "*bypass*" parameterization that relates slopes and intercepts to factors such as cloud distance and scene reflectance, enabling the quantification of 3D cloud effects under varying conditions."

**Changes in manuscript:** The original contents in lines 80-88 have been modified. Please refer to lines 163-178 of the revised manuscript.

Minor Comments

**Comment 07:** The paper often refers to qualitative statements that should be quantified or omitted. I pointed out individual instances below.

The abstract should be shortened and more to the point. What are the key takeaways from this study. Not necessary to expose all the 'sausage making' in the abstract.

**Response 07:** Thank you for the suggestion. We shortened the abstract as suggested:

● Revised abstract (Lines 16-33 in the revised manuscript):

"Accurate and continuous measurements of atmospheric carbon dioxide ($CO_2$) are essential for climate change research and monitoring of emission reduction efforts. NASA's Orbiting Carbon Observatory (OCO-2/3) satellites have been deployed to infer the column-averaged $CO_2$ dry air mixing ratio ($X_{CO2}$) from passive spectroscopy, with a designed uncertainty of less than one ppm for the regional average. This accuracy is often not met in cloudy regions because clouds in the vicinity of a footprint introduce biases in the $X_{CO2}$ retrievals. These arise from limitations in the one-dimensional (1D) forward radiative transfer (RT) model, which does not capture the spectral radiance perturbations introduced by clouds adjacent to a clear footprint. Our paper introduces a three-dimensional (3D) RT pipeline to explicitly account for these effects in real-world satellite observations. This is done by ingesting collocated imagery and reanalysis products to calculate the cloud-induced perturbations at the footprint level. To make that computationally feasible, a simple approximation for their spectral dependence is used. The calculated perturbations are then used to reverse (undo) the cloud vicinity effects at the radiance level, at which point the standard 1D OCO-2 retrieval code can be applied without modifications. For two cases over land, we demonstrate that this approach indeed reduces the $X_{CO2}$ anomalies near clouds. We also characterize the dependence of the $X_{CO2}$ footprint-level bias on the distance from clouds and other key scene parameters, such as surface reflectance. Although this dependence may be specific to cloud type, aerosols, and other factors, we illustrate how it could be parameterized to bypass our physics-based 3D-RT pipeline for use in an operational framework. In the future, we intend to explore this possibility by applying our tool to a variety of scenes over land and ocean."

**Comment 08:** I would suggest to merge section 1 and 2.

**Response 08:** Thank you for the suggestion. We agree that section 2 could be seen as an extension of the introduction. However, section 1 is already quite extensive, and combining these sections may make the introduction overly long. By keeping sections 1 and 2 separate, we aim to maintain reader focus on the linearization of the 3D effect, which is better emphasized as a distinct section.

Specific comments by Line:

**Comment 09:** L20: quantify 'high precision' or omit

**Response 09:** Thank you for pointing out the issue. We update the text as below:

● Original text (Lines 18-20):
"NASA's Orbiting Carbon Observatory (OCO-2/3) satellites have been deployed to measure the column-averaged $CO_2$ dry air mixing ratio ($X_{CO2}$) with very high precision."

● Revised text (Lines 17-19 in the revised manuscript), with the main changes underlined:
"NASA's Orbiting Carbon Observatory (OCO-2/3) satellites have been deployed to infer the column-averaged $CO_2$ dry air mixing ratio ($X_{CO2}$) from passive spectroscopy, with a designed uncertainty of less than one ppm for the regional average."

**Comment 10:** L20 – L23: Sentence starting with 'Although …' is hard to digest and should be simplified, maybe broken up.

**Response 10:** Thank you for your comment. We revised the sentence to improve readability, as below:

●       Original text (Lines 20-23):

"Although cloudy measurements are screened out, nearby clouds can still cause retrieval biases because the forward one-dimensional (1D) radiative transfer (RT) model used in the OCO retrieval algorithm does not account for the scattering induced by clouds in the vicinity of the OCO-2/3 footprints."

●       Revised text (Lines 19-22 in the revised manuscript):

"This accuracy is often not met in cloudy regions because clouds in the vicinity of a footprint introduce biases in the $X_{CO2}$ retrievals. These arise from limitations in the one-dimensional (1D) forward radiative transfer (RT) model, which does not capture the spectral radiance perturbations introduced by clouds adjacent to a clear footprint."

**Comment 11:** L27: remove 'with two metrics (called slope and intercept)'

**Response 11:** Thank you for the suggestion. We edited the abstract as suggested (as shown previously).

**Comment 12:** L28: remove 'and accelerate the radiative transfer by a factor of 100'

**Response 12:** Thank you for the suggestion. We edited the abstract as suggested (as shown previously).

**Comment 13:** L31 – L35: Sentence starting in 'EaRT-OCO .. ' -> move out of abstract.

**Response 13:** Thank you for the suggestion. We edited the abstract as suggested (as shown previously).

**Comment 14:** L36: remove '– the first physics-based correction of 3D-RT effects on OCO-2/3 retrievals'

**Response 14:** We would like to clarify that the current 3D bias mitigation methods proposed by Massie et al. (2022) and Mauceri et al. (2023) for OCO are primarily statistical-based. In contrast, the methods we proposed in this study are based on a physical understanding of the mechanism difference between 1D and 3D radiative transfer.

**Comment 15:** L37-L43: shorten, simplify discussion of 'bypass' method.

**Response 15:** Thank you for your comment. We revised the sentence as shown below:

●       Original text (Lines 37-43):

"Although the accelerated 3D-RT radiance adjustment step is faster than full 3D-RT calculations for all OCO spectral bands, it still requires at least as much computational effort as the $X_{CO2}$ retrieval itself. To bypass 3D-RT altogether, the slope and intercept metrics are parameterized as a function of the weighted cloud distance of a footprint and several other scene parameters, all of which can be derived directly from Aqua-MODIS imagery. While this method is fastest and thus feasible for operational use, it requires careful validation for various surface and atmospheric conditions."

●       Revised text (Lines 29-31 in the revised manuscript):

"We also characterize the dependence of the $X_{CO2}$ footprint-level bias on the distance from clouds and other key scene parameters, such as surface reflectance. Although this dependence may be specific to cloud type,

aerosols, and other factors, we illustrate how it could be parameterized to bypass our physics-based 3D-RT pipeline for use in an operational framework."

**Comment 16:** L62: quantify 'accuracy' requirement from the two cited studies.

**Response 16:** We add a description of their emphasis on accuracy.

- Line 61-63: Deng et al. (2016) and Crowell et al. (2018) also emphasize the significance of the level of $X_{CO2}$ accuracy for reliable $CO_2$ flux determination."

- Revised text (Lines 103-108 in the revised manuscript), with the main changes underlined:

"Deng et al. (2016) and Crowell et al. (2018) highlight the importance of achieving high $X_{CO2}$ measurement accuracy for reliable $CO_2$ flux estimation. Deng et al. (2016) show that the assimilation of GOSAT $X_{CO2}$ data with a precision of approximately 0.5–1.0 ppm can significantly improve regional CO2 flux estimates over both land and ocean. Similarly, Crowell et al. (2018) emphasize that an $X_{CO2}$ precision of 0.5–1.0 ppm is essential for detecting regional flux perturbations, especially in cloud-prone and high-latitude regions where $CO_2$ fluxes are difficult to constrain accurately using ground-based sensors alone."

**Comment 17:** L76: remove sentence 'The cloud-related …'

**Response 17:** Thank you for the suggestion. We remove Line 76: "The cloud-related bias is also evident when examining individual footprints." as suggested.

**Comment 18:** L90-91: Restate comment that no 'practical strategies' have been developed to correct 3D cloud effects based on the physical understanding. The study by Mauceri et al (2023) uses physics derived variables to correct for 3D cloud biases.

**Response 18:** We appreciate your comments on the practical strategy. You are correct that the machine learning-based method developed by Mauceri et al. (2023) is indeed a practical approach to the real data. We understand that Mauceri et al. (2023) and Massie et al. (2022) use several physics-*derived* variables, such as H3D, HC, and CSNoiseRatio. However, these variables capture the 3D cloud effect only at a single band or average of continuum wavelengths. More importantly, these metrics are not used to correct the 3D cloud effect at the *radiance level*, but operate primarily on the L2 products. While cloud distance, similar to our study, can indicate potential 3D cloud biases, these variables alone cannot fully capture the reflectance-dependent 3D cloud effect across the entire spectrum. Although Mauceri et al. (2023) apply machine learning-based corrections, it is still a *statistical* mitigation approach in nature and does not go into the fundamental physics (i.e., the radiance level). We want to emphasize that our approach is based on a footprint-by-footprint *deterministic* rather than a multi-footprint *statistical* approach, albeit with some simplifications that are noted in the original manuscript (with more detail in the revised manuscript, responding to a different reviewer). It is a radiance-only approach and different from the existing statistics method. To make this more clear, we made edits as shown below:

- Original text (Lines 89-92):

"Although the physical mechanism of the $X_{CO2}$ 3D cloud retrieval bias is now largely understood, practical strategies for applying these insights to a bias correction have not been developed thus far. Mauceri et al. (2023) employed machine learning techniques to correct for 3D cloud biases using observations from the Total Carbon Column Observing Network (TCCON)."

- Revised text (Lines 166-171 in the revised manuscript), with the main changes underlined:

"Although the reflectance-dependent physical mechanisms of the $X_{CO_2}$ 3D cloud retrieval bias are now largely understood, strategies for applying these insights to bias correction have thus far been done empirically or statistically. For example, Massie et al. (2022) proposed an empirical lookup table to correct 3D cloud biases based on a 3D metric, and Mauceri et al. (2023) used machine learning techniques combined with TCCON observations. While both approaches are operationally applicable, they rely on statistical corrections rather than true physical radiance difference of the 3D cloud effect across the entire spectrum."

**Comment 19:** L93: Please also include/cite work by Massie et al where they worked on correcting 3D cloud biases with linear fits to physics derived variables.

**Response 19:** Thank you for the suggestion. As described above, we have included Massie et al. (2022) and discussed the physics-derived variables they used, such as H3D, HC, and CSNoiseRatio.

**Comment 20:** L106: 'on the a footprint-by-footprint '

**Response 20:** Thank you for pointing out the typo. We have revised the paragraph in the introduction, and the sentence has been removed.

**Comment 21:** L126: specify that range 'dynamic range of interest for reflectance'

**Response 21:** Figure 1 shows the spread of perturbations at low reflectance, which are primarily due to photon noise in the Monte Carlo RT simulations. To avoid large simulation uncertainties in this low-reflectance region, we set a transmittance threshold for each band. This threshold is defined as the minimum of (1) 40% of the maximum transmittance at each band or (2) the minimum transmittance value of each band. This ensures that the analysis focuses on the dynamic range of interest for reflectance where the simulation results are more reliable.

**Comment 22:** L142-145: Hard to follow 'Increased photon …' . Please rewrite, expand.

**Response 22:** Thank you for pointing out the problem. We edit the sentences as below:

- Original text (Lines 142-145):

"Increased photon path lengths from multiple scattering in 3D-RT produce non-zero perturbations (percentage differences in 1D and 3D radiances) expressed in Eq. (1). Since wavelengths with higher absorption are attenuated more than those with lower absorption, the Eq. (1) perturbations are a function of reflectance (line absorption depth), referred to later as spectral distortion."

- Revised text (Lines 297-302 in the revised manuscript):

"Multiple scattering in 3D-RT increases the photon path lengths and enhances absorption across different wavelengths, leading to non-zero perturbations, as expressed by Eq. (1) (percentage differences between 1D and 3D radiances). This effect is more pronounced at wavelengths with higher absorption, which experience greater attenuation compared to those with weaker absorption for the same photon path length. As a result, the perturbations vary depending on the reflectance and absorption depth, a phenomenon referred to as spectral distortion in our study."

**Comment 23:** L154: Why not use B11?

**Response 23:** The B11 version was not publicly available when this study began, and we have encountered issues with retrieval code compilation for B11. Once these issues are resolved, we plan to update our analysis using B11 in future work. We plan to keep updating our algorithm, and switching to B11 will be one of these updates, to be documented in the next publication related to EaR³T-OCO.

**Comment 24:** L244: 'To mitigate excessive computational demands, we opt to use solely the wavelengths of the first footprint.' -> how does this impact the results?

**Response 24:** The primary difference among the eight footprints in the same swath is the instrument line shape (ILS), which can slightly influence the gas absorption optical depth. Although we have not yet run simulations with varying ILS, we expect the overall trends for the slope and intercept to remain similar. This is because the perturbations are calculated using the *differences* between footprint-level 1D-RT and 3D-RT simulations, where any variations due to ILS are irrelevant. Then, the spectral radiance perturbations are applied to other footprints, with slightly different ILS and dispersion. Since the un-perturbation based on the calculated perturbations operates in radiance (reflectance) space rather than wavelength space, small changes in spectral attributes of other footprints are not expected to have any significant impact on the algorithm. However, further analysis would be needed to confirm this. Relative to other factors such as cloud geometry, sun-sensor geometry, etc., this effect is most likely negligible. Again, this will be studied more thoroughly in forthcoming publications.

**Comment 25:** L262: how did the various reflectance thresholds influence the results.

**Response 25:** The reflectance thresholds significantly impact cloud detection and subsequent radiance simulations. If the reflectance threshold is set too high, thin clouds may go undetected, resulting in an underestimation of cloud impact. Conversely, if the threshold is set too low, some bright surface pixels could be misclassified as clouds, leading to overestimating cloud effects. Both scenarios deviate from real conditions, making it difficult to accurately represent the 3D cloud effect and potentially bias the simulation results.

**Comment 26:** L263: why did you need to develop a new cloud detection approach?

**Response 26:** The cloud products provided by MODIS have a spatial resolution of 1 km, which is too coarse for our simulation needs. To address this, we developed a new cloud detection approach to optimize detection specifically for the study case, ensuring higher accuracy in identifying clouds. While the method used by Chen et al. (2023) is more generalized and suitable for a wide range of scenarios, it may miss some thin clouds. Our customized approach helps to better capture these thin cloud features, which is crucial for accurately modeling the 3D cloud effects in the selected scene.

**Comment 27:** L298: 'uncertainties' : keep in mind that the uncertainties in s, I, depend on many more factors than captured by the uncertainty in the line fit. Thus, you would underestimate the true uncertainties with that approach.

**Response 27:** Indeed, in Tables 1 and 2, we currently only quantify the uncertainties associated with the linear fit. We acknowledge that the true uncertainties are influenced by additional factors, such as variations in geometry, cloud properties, and aerosol characteristics. These contributions will be considered in future studies as we expand our analysis to include a wider range of conditions.

We added a description in the table caption in response to your comment.

- Original text (line 433; lines 473-474; 503-504):

"Table 1. Amplitude and e-folding distances for *s* and *i* fittings in the $O_2$-A, $WCO_2$, and $SCO_2$ bands."

"Table 2. Amplitude and e-folding distances for *s* and *i* fittings of the" simulation with a homogeneous aerosol layer in the $O_2$-A, $WCO_2$, and $SCO_2$ bands."

"Table 3. Amplitude and e-folding distances for *s* and *i*, determined using different average grid points in simulations with a homogeneous aerosol layer for the $O_2$-A, $WCO_2$, and $SCO_2$ bands."

- Revised text (Lines 659-660; lines 703-704; lines 747-748 in the revised manuscript), with the main changes underlined:

"Table 1. Amplitude and e-folding distances for *s* and *i* fittings in the $O_2$-A, $WCO_2$, and $SCO_2$ bands. Errors represent fitting uncertainty only and may be underestimated."

"Table 2. Amplitude and e-folding distances for *s* and *i* fittings of the" simulation with a homogeneous aerosol layer in the $O_2$-A, $WCO_2$, and $SCO_2$ bands. Errors represent fitting uncertainty only and may be underestimated."

"Table 3. Amplitude and e-folding distances for *s* and *i*, determined using different average grid points in simulations with a homogeneous aerosol layer for the $O_2$-A, $WCO_2$, and $SCO_2$ bands. Errors represent fitting uncertainty only and may be underestimated."

**Comment 28:** L307: what are those 'various processes'?

**Response 28:** The retrieval accounts for various processes, including water vapor absorption, surface albedo variations, cloud and aerosol scattering in the 1D-RT model, radiance polarization effects, and the impact of the instrument line shape, among others.

**Comment 29:** L310: The code on Gituhub is not the code used for the operational retrieval.

**Response 29:** Thank you for pointing out the issue. While the code on GitHub is not the same as the operational retrieval code, we have tested it using B10r (i.e., B10.0.04_sdos_testing_1) and obtained the same $X_{CO2}$ values as those in the L2 $X_{CO2}$ data. This confirms that the GitHub code is functionally equivalent to our analysis. A future version of our method could therefore easily be integrated into the data stream that uses the actual operational code (after further testing of our code with more data and updating it as needed, of course).

**Comment 30:** L320: explain terms in equation

**Response 30:** We edit the sentence and add the description as below:

- Original text (Lines 142-145):

"Upon deriving the 3D parameters in Section 4.3, we can convert the OCO-2 spectra using Eq. (4) with the observed radiance spectra and corresponding reflectance, slope, and intercept. Assuming the absence of 3D effects in the adjusted 1D radiance, we can employ the B10.04 retrieval algorithm with un-perturbed spectra to obtain mitigated $X_{CO2}$."

- Revised text (Lines 397-401 in the revised manuscript), with the main changes underlined:

"Upon deriving the 3D parameters in Section 4.3, we calculate the adjusted OCO-2 spectra ($I_\lambda^{IPA\,(adjusted)}$) using Eq. (4) with the observed radiance spectra ($\underline{I_\lambda^{3D\,(obs)}}$) and corresponding reflectance ($\underline{R_\lambda^{obs}}$), slope $\underline{(s_{xy})}$, and intercept $\underline{(i_{xy})}$. Assuming the absence of 3D effects in the adjusted 1D radiance, we can employ the B10.04 retrieval algorithm to the adjusted spectra ($I_\lambda^{IPA\,(adjusted)}$) to obtain mitigated $X_{CO2}$."

$$I_\lambda^{IPA\,(adjusted)}(x, y) = \frac{I_\lambda^{IPA\,(obs)}(x, y)}{(i_{xy} + s_{xy} \times R_\lambda^{obs} + 1)}$$

**Comment 31:** L325: 'parameters that accurately represent'

**Response 31:** We interpret this as a suggestion to clarify how the slope and intercept parameters are defined to represent the 3D cloud effect in OCO-2 observations accurately. The original sentence aims to convey that realistic radiance simulations near the satellite's footprint are essential for deriving these parameters precisely. Although achieving perfect accuracy in the simulations is challenging, we strive to approximate real conditions as closely as possible using MODIS observations. If further clarification was intended, we would appreciate any additional guidance.

- Original text (Lines 325-326):

"In order to derive the slope (s) and intercept (i) parameters that accurately represent the 3D cloud effect in the OCO-2 observations, it is crucial to perform realistic radiance simulations near the satellite's footprint."

**Comment 32:** L330: Quantify 'shows a good agreement'

**Response 32:** We added an $R^2$ and slope for the scatter plot in the sentence.

- Original text (Line 330):

"The heat map in Fig. 3c shows a good agreement between the simulation and observation."

- Revised text (Lines 538-539 in the revised manuscript):

"The heat map in Fig. 3c shows a good agreement between the simulation and observation, with an $R^2$ of 0.69 and a slope of 0.71."

**Comment 33:** L330: remove sentence 'As a result, …'

**Response 33:** Thank you for the suggestion. We removed Lines 330-331: "As a result, we are confident that the simulation at other wavelengths is able to approach the actual condition." as suggested.

**Comment 34:** L332: COT repeated twice

**Response 34:** Thank you for pointing out the typo. One COT should be "CTH" instead.

**Comment 35:** Figure 3: How much was the COT and CER tuned to agree? Could we get the right answer for the wrong reason?

**Response 35:** We currently apply 1D-RT reflectance-to-COT mapping and fixed CER values for low and high clouds, which can lead to biases and often overestimate COT for low radiance and underestimate it for high radiance. Accordingly, the 3D-RT cloudy pixel radiance will also differ from the observation, as shown in Fig. R9. However, since our primary focus is on the bright areas, we prioritize capturing the radiance differences over these regions rather than achieving perfect agreement for all cloudy pixels. We aim to capture the general trend in the bright areas, where minor uncertainties in cloud and aerosol setups are acceptable compared to the larger differences observed between 1D-RT and 3D-RT simulations.

However, it is important to understand how COT and CER change for the same cloud field can alter the 3D cloud effect magnitude. This is crucial to evaluate the result uncertainty. We will investigate the influence of the cloud properties in the future as well as more parameters. We appreciate your question on this topic.

[Figure]

Figure R9. A scatter comparison between Fig. 3a and 3b (Fig. R10 below) for pixels with COT greater than 0.

[Figure]

(edited Fig. 3) Figure R10. MODIS observation at 650 nm (a) and 3D radiance simulation at 650 nm by EaR³T (b). A scatter comparison between (a) and (b) is depicted in (c).

**Comment 36:** L341: how did you arrive a 11 wavelengths? What happens if you use 10 or 12? Aka, how sensitive are you to this choice? Would be a great opportunity to plot accuracy vs number of wavelengths.

**Response 36:** The choice of 11 wavelengths was made as a compromise between computational cost and fitting accuracy. Using more wavelengths would indeed result in a better fit for the full-spectrum simulation. However, increasing the number of wavelengths significantly raises computational time and cost. We have not yet systematically evaluated the sensitivity of accuracy to the number of wavelengths, but it would be a valuable analysis to explore in future work, potentially plotting accuracy versus the number of selected wavelengths to determine the optimal balance. In addition, we plan to use ALIS (Emde et al., 2022) or the acceleration method by Iwabuchi instead of our multi-wavelength method. Either of these might be even faster than our method, but we need to evaluate their accuracy.

**Comment 37:** Figure 5: how do the other bands look like? 5 a) looks very noisy far away from the clouds.

**Response 37:** The observed noise far from the clouds in Figure 5a may be due to variations in surface reflectance across the region, which can affect the stability of the derived parameters. We will investigate this further to determine if additional filtering or adjustments are needed.

Here are similar figures for Fig. 5 but for $WCO_2$ and $SCO_2$ bands:

[Figure]

Figure R11. Distribution of (a) s and (b) i of $WCO_2$ band. Red dots denote the cloud pixels employed in the RT simulation.

[Figure]

Figure R12. Distribution of (a) s and (b) i of $SCO_2$ band. Red dots denote the cloud pixels employed in the RT simulation.

**Comment 38:** L463: 'bands, potentially increasing'

**Response 38:** We interpret this as a potential suggestion to clarify how aerosols might impact the uncertainty in deriving the 3D effect parameters. The original sentence intends to highlight that aerosols can affect both the s and i of the $O_2$-A and $SCO_2$ bands. Accurate representation of aerosol optical depth (AOD) is crucial, as inaccuracies could lead to errors in capturing the 3D effects. If further clarification was intended, we would welcome additional guidance.

- Original text (Lines 462-464):

"We illustrate that the presence of aerosols can lead to alterations in both the s and i of the $O_2$-A and $SCO_2$ bands, potentially increasing the uncertainty associated with the derivation of 3D effect parameters."

**Comment 39:** L483: state footprint sizes of upcoming satellites, name and cite those satellites

**Response 39:** We added the description below:

- Original text (Lines 482-484):

"Numerous upcoming satellites for $CO_2$ remote sensing will adopt similar retrieval algorithms but feature varying footprint sizes in accordance with their specific mission objectives."

- Revised text (Lines 719-723 in the revised manuscript):

"Numerous upcoming satellites for $CO_2$ remote sensing will adopt similar retrieval algorithms but feature varying footprint sizes in accordance with their specific mission objectives. For example, the Copernicus Anthropogenic CO2 Monitoring Mission (CO2M) by the European Space Agency (ESA) plans to have a footprint size of 4 $km^2$ (2 km by 2 km; Kuhlmann et al., 2020). MicroCarb by the Centre National d'Etudes Spatiales (CNES) will have a larger footprint size of 40.5 $km^2$ (4.5 km by 9 km; Cansot et al., 2023)."

**Comment 40:** L487: why did you not investigate smaller footprints?

**Response 40:** We did not investigate smaller footprints because the OCO-2/3 satellites currently offer the smallest footprint size available for $CO_2$ measurements among existing and upcoming satellite missions. However, newly proposed satellites for greenhouse gas remote sensing may feature smaller footprints, and we plan to consider this analysis in the future. We anticipate that 3D cloud effects could become more pronounced as footprint size decreases.

**Comment 41:** L490: 'of pronounced biases'

**Response 41:** We remove the "pronounced" in the sentence.

- Original text (Lines 489-490):

"This decline aligns with the expectation that larger footprints would mitigate the spectral distortion effect, reducing the prevalence of pronounced biases."

- Revised text (Lines 728-729 in the revised manuscript):

"This decline aligns with the expectation that larger footprints would mitigate the spectral distortion effect, reducing the magnitude of 3D cloud biases."

**Comment 42:** L490-491: not clear what changes are not significant

**Response 42:** We edit the sentence as below:

- Original text (Lines 490-491):

"Notably, there is no statistically significant change in $a_i$ and $d_i$ of the intercept."

- Revised text (Lines 729-730 in the revised manuscript):

"No statistically significant change exists in $a_i$ and $d_i$ in the intercept values across different footprint sizes."

**Comment 43:** L495: 'In conclusion, future satellite missions with any …' That is a very strong statement without any quantification. This would depend on the retrieval algorithm, chosen bands, accuracy requirements, area of interest, …

**Response 4:** We revise the statement as below:

- Original text (Lines 495-496):

"In conclusion, future satellite missions with any footprint size must account for 3D biases to ensure accurate remote sensing of $X_{CO2}$."

- Revised text (Lines 733-737 in the revised manuscript):

"We suggest that future satellite missions, regardless of footprint size, consider accounting for 3D biases to improve the accuracy of $X_{CO2}$ retrievals. Studies need to be conducted to ensure that, given the bands, footprint size, and other attributes, the retrieval error induced by 3D clouds does not exceed the mission requirements – similar to what has been demonstrated for OCO-2 in this study."

**Comment 44:** L500: quantify 'to substantial 3D'

**Response 44:** We add the description as below:

- Original text (Lines 498-500):

"Conversely, missions designed with smaller footprint sizes than OCO-2, particularly those targeting enhanced data acquisition in cloud-prone regions such as the Amazon Basin (Frankenberg et al., 2024) will be susceptible to substantial 3D cloud biases."

- Revised text (Lines 738-741 in the revised manuscript), with the main changes underlined:
"Conversely, missions designed with smaller footprint sizes than OCO-2, particularly those targeting enhanced data acquisition in cloud-prone regions such as the Amazon Basin – where Land Nadir observations have been reported to exhibit biases up to -0.48 ppm in both hemispheres (Massie et al., 2023) – must rigorously account for 3D radiative transfer effects."

**Comment 45:** L501: why do 3D cloud biases need to be considered in the initial planning stage? Algorithms are typically tackled much later.

**Response 45:** While algorithm development typically occurs at later stages, the experience from OCO-2/3 has shown that 3D cloud biases can significantly impact CO2 measurements, especially when targeting regions like the cloudy Amazon. With a decade of observations highlighting this issue, it is crucial to consider 3D cloud biases during the initial planning stages of new satellite missions—particularly for those aiming for smaller footprint sizes—so that the instrument design and mission parameters can be optimized to minimize these biases from the outset. The way this is typically done at the mission development stage is in mission or science traceability matrices (STM), which are part of every proposal. Any serious mission proposal of the future needs to show that they consider the impact of ubiquitous clouds on the derivation of $X_{CO2}$ or other trace gas products from the radiances when discussing expected uncertainties. This does not require ready-made algorithms, and merely needs to consider the literature (e.g., Massie, Mauceri, Emde, our own study).

**Comment 46:** L517: How could the bypass method deal with cloud shadows?

**Response 46:** We don't believe that this is feasible or necessary at this point. As noted in the response on page 7, only a few cloud-shadow retrievals are present in the OCO-2 Lite files due to pre-retrieval cloud screening algorithms (Massie et al., 2023). Additionally, the B11 retrieval has improved at filtering out shadowed footprints. We believe that the few remaining cloud-shadow footprints passing the pre-screening are less significant compared to the clear-sky footprints affected by nearby clouds. However, applying the same radiance adjustment for bright areas to shadowed regions could introduce additional errors. Since footprints affected by cloud shadows constitute a relatively small portion of the overall effective OCO-2 retrievals, we believe that our approach provides a reasonable average correction for the majority of clear-sky and bright-area footprints.

**Comment 47:** L524: Quality Flag =0 or 1 are not 'best quallity' data. That would only by 0

**Response 47:** Thank you for the comment. We edit the sentence and remove the description of "best quality" as below:

- Original text (Lines 523-524):
"Subsequently, we determine the s and i of the three bands using the coefficients in Table 2 for all footprints that are the best quality (Quality Flag = 0 or 1) data points."

- Revised text (Lines 790-792 in the revised manuscript):
"To mitigate the 3D biases in $X_{CO2}$, it is essential to compute the 3D parameters (*s* and *i* for the three OCO-2 spectral bands) for all footprints that pass the pre-screening (Quality Flag = 0 or 1)."

**Comment 48:** L524: How are the values in Table 2 derived for the bypass method when they don't include 3D RT calculations.

**Response 48:** We want to clarify that values in Table 2 are derived using the baseline method, which involves 3D-RT calculations for specific scenarios. The bypass method is then parameterized based on these baseline results. Our approach is to analyze various solar and viewing geometries, as well as different cloud and aerosol properties, using the baseline method. Once we establish these relationships, we can derive a generalized bypass method that can be applied under a wide range of conditions without additional 3D-RT simulations.

**Comment 49:** Figure 9: Not sure if b) is improved compared to a) outside of the cloud shadow area.

**Response 49:** Thank you for pointing out the concern. Fig. 9c shows that footprints over both the south and north sides of clouds have a decrease in $X_{CO2}$. This means that footprints over clear-sky areas do reduce the 3D bias.

**Comment 50:** L570 – L573: You state a problem with thin or partial clouds for the bypass method. How would an operational algorithm deal with that?

**Response 50:** Footprints containing thin or partial clouds pose additional challenges beyond 3D photon scattering, such as elevated reflectance at the continuum wavelength compared to clear-sky conditions. An operational algorithm would need to account for these effects by either incorporating additional parameters (e.g., cloud optical depth or cloud fraction) or using more complex correction models to differentiate between

3D scattering biases and direct cloud contamination. This would ensure that the bypass method remains effective in mixed or thin cloud conditions.

**Comment 51:** L585: remove 'on a cluster at the University of Colorado'

**Response 51:** Thank you for the suggestion. We changed as the manuscript suggested and leave the statement in the acknowledgments.

**Comment 52:** L592: You state that the bypass method can be supplemented by periodic full calculations. How would that work in detail? When do you run them, how do you use their results to improve the results?

**Response 52:** The timing and frequency of performing full 3D-RT calculations would depend on how sensitive the bypass method's parameters are to changes in key state variables, such as solar and viewing geometry, cloud and aerosol properties, and surface albedo. The first step is to establish a generalized bypass method that captures the influence of these variables. If the derived parameters are found to be highly sensitive to changes in these conditions, then more frequent recalibration using full 3D-RT simulations would be necessary to maintain the accuracy of the bypass method.

In practice, this could involve periodically running full 3D-RT simulations for a subset of representative scenarios (e.g., different seasons, surface types, or cloud conditions) and updating the bypass parameterization accordingly. These recalibrated parameters would then be applied to operational retrievals, ensuring that the bypass method remains robust over time.

**Comment 53:** Figure 12. Where does the XCO2 in those scenes come from?

**Response 53:** The result shown in Fig. 12 is derived using the bypass parameterization from Table 2. The goal of this figure is to illustrate how variations in cloud distribution can lead to different cloud-induced biases. We started with the OCO-2 radiance data presented in Figure 2 and applied radiance adjustments to introduce 3D cloud biases. This approach allows us to explore the impact of 3D cloud biases for different effective cloud distances and assess how cloud distribution influences the retrieved $X_{CO2}$ values.

**Comment 54:** L630: 'We documented the …' -> The main manuscript does not contain any documentation of the toolbox. Would remove that statement.

**Response 54:** Thank you for the comments. We removed Lines 630-631: "We documented the EaR³T-OCO radiance simulator, an automated tool for calculating spectral radiances observed by NASA's OCO-2 satellite." in the conclusion.

**Comment 55:** L671: 'more accurate level of accuracy'?

**Response 55:** We revise the sentence to emphasize accuracy improvements near clouds:

●     Original text (Line 670-672):
"Our work can become the stepping stone toward more accurate and efficient trace gas retrievals even in complex scenes, ultimately bringing spaceborne trace gas retrievals to a more accurate level of accuracy."

●     Revised text (Lines 1056-1057 in the revised manuscript):
"Our work can become the stepping stone toward more accurate and efficient trace gas retrievals in the vicinity of clouds."

**Comment 56:** L672: remove last sentence 'It will improve …' Your study did not show any information to draw that conclusion.

**Response 56:** Thank you for the comment. We change the sentence as below:

● Original text (Lines 672-673):
"It will improve current flux inversions (especially over the cloud-prone Amazon) and other applications."

● Revised text (Lines 1057-1058 in the revised manuscript):
"Looking ahead, adapting this mitigation to operational workflows could markedly improve $X_{CO2}$ accuracy in cloud-prone regions—including the Amazon—and thereby enhance the fidelity of $CO_2$ flux inversions."

**Comment 57:** L685: GitHub for OCO-2 toolbox leads to a 404 page not found

**Response 57:** Thank you for pointing this out. We updated the link from "https://github.com/ywchen-tw/OCO-2" to "https://github.com/ywchen-tw/OCO2" to resolve the issue.

**Comment 58:** Figure A3. Why is cloud effective radius only one fixed number for the whole scene?

**Response 58:** We apologize for not making this clearer in the manuscript. For this study, we chose to keep the cloud effective radius (CER) constant for simplicity, which is why it appears uniform in Fig. A3c. Specifically, we assigned CER values of 10 μm for low clouds and 25 μm for high clouds. In future work, we intend to incorporate MODIS-derived CER values to better capture spatial variability.

In response to your other questions, we have clarified the description in line 265 as follows:

● Original text (Line 265):
"Once the cloudy pixels are identified, we retrieve the cloud top height (CTH) and cloud effective radius (CER) of the nearest location from the MODIS MYD02 cloud file and assign them to each cloudy grid point."

● Revised text (Lines 1430-1433 in the revised manuscript):
"Once the cloudy pixels are identified, we retrieve the cloud top height (CTH) of the nearest location from the MODIS MYD02 cloud file and assign it to each cloudy grid point. The cloud effective radius (CER) is manually set to 10 μm for low clouds and 25 μm for high clouds in this study. In future versions, we plan to use the actual MODIS CER values to capture more realistic variations."

**Comment 59:** Figure A6. Would remove. There is not much information here beyond what one would expect.

**Response 59:** Thank you for the suggestion. We removed it as suggested.

References:

● Kuhlmann, G., Brunner, D., Broquet, G., and Meijer, Y.: Quantifying $CO_2$ emissions of a city with the Copernicus Anthropogenic $CO_2$ Monitoring satellite mission, Atmos. Meas. Tech., 13, 6733–6754, https://doi.org/10.5194/amt-13-6733-2020, 2020.

- Cansot, E., Pistre, L., Castelnau, M., Landiech, P., Georges, L., Gaeremynck, Y., and Bernard P.: MicroCarb instrument, overview and first results, Proc. SPIE 12777, International Conference on Space Optics — ICSO 2022, 1277734 (12 July 2023); https://doi.org/10.1117/12.269033.

**Point-by-Point Response to CC1**

Dear Jesse,

We appreciate your comments and suggestions on this paper. We have worked to clarify any unclear descriptions in the manuscript and to address each of your questions. Your insights on cloud base height and the aerosol layer are especially valuable to our research. In response, we conducted additional simulations regarding cloud base height and aerosol top layer placement, which we discuss in detail in the responses below.

We also thank you for your suggestion on considering effective cloud distance with respect to vertical cloud base height and reflectance. Our main concern is that cloud-based information is not always readily available, introducing significant uncertainty in cloud distance calculations. We will continue exploring different effective cloud distances that incorporate various factors, including cloud reflectance, surface albedo, inhomogeneity, and, when possible, cloud base height.

Below, we provide our responses to each of your comments and suggestions. We have marked your comments in red, our responses in blue, and the original paper content in black.

**Comment 01:** This manuscript develops a method for the mitigation of 3D radiative transfer effects on retrievals of carbon dioxide concentration from the Orbiting Carbon Observatory satellites. The novelty of this work is that it provides a pathway for physics-based mitigation of 3D radiative transfer effects using parameterizations that can be applied operationally.

**Response 01:** Thank you for summarizing the paper. We appreciate your comments on our research.

**Comment 02:** I enjoyed reading this paper, and I have several comments and suggestions detailed below:

As I understand it, all of the forward modelling of the OCO bands in this paper utilizes the linear approximation suggested by Schmidt et al. (in prep). For this paper, it is important that we know how the error in the linear approximation propagates into uncertainty in the relationship between $\Delta X_{CO2}$ and the radiances i.e., how accurate is the reference calculation?

**Response 02:** Thank you for the question. As shown in Fig. R1 (an edited version of Fig. 1), the perturbation at low reflectance spreads largely due to the high uncertainty of Monte Carlo radiance simulations at extremely low transmittance. To avoid high uncertainty and heavy computation, we define a minimum transmittance threshold for each band, set as the lower of either (1) 40% of the band's maximum transmittance or (2) the band's minimum transmittance. This threshold reduces the potential for linear approximation errors. Additionally, increasing the number of wavelengths and photons in the simulation further stabilizes the linear approximation. These steps ensure that the linear approximation accurately represents the perturbation across the entire reflectance range within an acceptable level of uncertainty. However, to better quantify the impact of the linear approximation error, we will explore incorporating a detailed uncertainty analysis in future work, specifically assessing how deviations in the linear model propagate into $\Delta X_{CO2}$ estimates. A glimpse of this work is shown in the Figure below. It illustrates the difference in slopes and intercept values when using just a few representative wavelengths in a channel vs. all/many of them.

[Figure]

(edited version of Fig. 1) Figure R1. Example of the linear relationship between perturbation and reflectance. The grey dots represent the complete wavelength range, while the red dots indicate the subset selected for the O₂-A band simulation. The black and red lines represent the linear fit of the grey and red dots, respectively,

**Comment 03:** At the moment, Section 2 states the result of Schmidt et al. (in prep) but doesn't provide much physical justification for the linear approximation itself.

**Response 03:** Thank you for the comment. We describe the physical meaning of the linear parameters in lines 140-147: "The slope and intercept are indicative of distinct physical phenomena: a non-zero slope corresponds to wavelength-dependent variations and differences in 1D and 3D radiances, photon path lengths, and absorption. Increased photon path lengths from multiple scattering in 3D-RT produce non-zero perturbations (percentage differences in 1D and 3D radiances) expressed in Eq. (1). Since wavelengths with higher absorption are attenuated more than those with lower absorption, the Eq. (1) perturbations are a function of reflectance (line absorption depth), referred to later as spectral distortion. The intercept is related to the often-reported increase of reflectance near clouds, or decrease in shadows, whereas the slope accounts for spectroscopic effects."

When we first found this remarkable linearity (Schmidt et al., 2019), we were somewhat surprised by it, and to this day there is no rigorous physical explanation. We only have empirical evidence that 3D perturbations are linear over the *relevant* dynamic range of reflectance, which arise from extra illumination of the surface mediated by multiple scattering in clouds. The photons causing this extra illumination have a history of photon path length *different* from the directly surface-reflected radiation. Specifically, absorption is enhanced for those photons experiencing longer photon path length.

We recognize that low-reflectance wavelengths do not always align well with the fitting lines due to their much lower intensities. Therefore, we focus our analysis on higher reflectance wavelengths, where the linear approximation is more robust and provides better accuracy in the relationship between $\Delta X_{CO2}$ and radiances.

Linearity actually breaks down for the very highest absorption optical depth (reflectance, transmittance ~0). At this point, the perturbation is close to zero. This can be shown with SHDOM calculations. However, this only happens at the very highest optical thickness, which relates to wavelengths where OCO-2 does not pick up a signal. For practical purposes, the assumption of linearity seems to be valid. For the future, it would be desirable to come up with a semi-analytic explanation for the transition from linear to non-linear range as the absorption optical thickness increases.

**Comment 04:** I think this section would benefit from a short paragraph discussing approximate acceleration methods for 3D RT such as (Partain et al., 2000; Doicu et al, 2020) in comparison to exact calculations like Emde et al. (2011), so that the strengths/weaknesses (accuracy vs. speed) of the linear approximation can be contextualized.

**Response 04:** Thank you for the suggestion. We have included a few studies that have progress on accelerating multi-wavelength 3D-RT calculation. These two papers can make the introduction more comprehensive. We added a summary of both papers in the introduction as below:

● Original text (Lines 80-88):

"Schmidt et al. (2024) explain that lateral photon transport can be understood as missing physics in the operational OCO algorithm, and any adjustments for discrepancies between 1D-RT and 3D-RT could introduce additional inaccuracies in $X_{CO2}$ retrieval. Although advances have been made in expediting high-resolution 3D-RT simulations by using the same photon paths for various wavelengths (Emde et al., 2011; Iwabuchi and Okamura, 2017), the computational demands of such models have still hindered their operational application. Schmidt et al. (2019) introduced the 3D-RT radiance perturbation as the percentage difference between the 3D and 1D radiance simulations. This radiance perturbation is found to be linear over the relevant dynamic range of reflectance, which allows a simple representation of the perturbation as slope and intercept for each of the three OCO-2 bands. The details will be described in Section 2."

● Revised text (Lines 126-165 in the revised manuscript), with the main changes underlined:

"Schmidt et al. (2019) explain that lateral photon transport represents missing physics in the operational OCO algorithm, and any adjustments for discrepancies between 1D-RT and 3D-RT could introduce additional inaccuracies in $X_{CO2}$ retrieval. To evaluate the differences between 1D-RT and 3D-RT, a high-resolution, multi-wavelength 3D-RT model is essential. Recent advancements have accelerated high-resolution 3D-RT simulations for multi-wavelength applications. For instance, Partain et al. (2000) introduced an enhanced implementation of the equivalence theorem, which decouples scattering and absorption calculations, allowing for accurate spectral integration without repeated multiple-scattering computations for Monte-Carlo models. Emde et al. (2011) developed the Absorption Lines Importance Sampling (ALIS) technique, which efficiently computes high-resolution polarized spectra by leveraging Monte Carlo photon tracing across multiple wavelengths simultaneously. Iwabuchi and Okamura (2017) also adopted a similar way of using the same photon paths for various wavelengths to accelerate multi-wavelength 3D-RT simulation. Doicu et al. (2020) accelerated the Spherical Harmonics Discrete Ordinate Method (SHDOM) 3D-RT model, which is different from Monte-Carlo-based 3D radiative transfer models, by combining the correlated k-distribution method with dimensionality reduction techniques, such as principal component analysis.

While these acceleration methods have the potential to improve the accuracy of trace gas retrievals by taking into account missing physics (horizontal photon transport), current operational retrievals still do not use true 3D-RT in trace gas retrieval processes. Here, we adopt the approach introduced by Schmidt et al. (2019) as a practical method to approximate the 1D-RT and 3D-RT differences in spectral radiance observations, building on the concept of 3D-RT radiance perturbations, defined as the spectral percentage difference between 3D and 1D radiance simulations. These simulations in this study are performed by a modified version of the Education and Research 3D Radiative Transfer Toolbox (EaR³T; Chen et al., 2023), tailored specifically for OCO (EaR³T-OCO). The 3D perturbations proved to be a linear function of the radiance (or reflectance) itself across the relevant dynamic range of reflectance, which allows its representation by a simple slope and intercept for each of the three OCO-2 bands, with further details provided in Section 2. We also introduce a "*bypass*" parameterization that relates slopes and intercepts to factors such as cloud distance and scene reflectance, enabling the quantification of 3D cloud effects under varying conditions."

**Comment 05:** The mitigation parameterization is based on simulated scenes derived from observations. Due to weak atmospheric scattering, the 3D enhancement effect studied here depends primarily on cloud-surface interactions. These will strongly depend on the geometric distance between cloud and surface (i.e., cloud base height and thickness). At the moment, the methodology doesn't state how the cloud base height is retrieved from the MODIS observations to form a synthetic cloud field, or its uncertainty. This procedure's uncertainty will feed into the simulations and affect how the intercept and slope parameters scale with effective distance. It would be good to address this within the manuscript as it will affect both the baseline and bypass approaches.

**Response 05:** Indeed, this is a problem that injects uncertainty. Unfortunately, it is not easy to address since MODIS retrievals only provide cloud top height. Attempts have been made (not by our team, but other OCO-2/3 team members) to retrieve cloud geometric thickness in addition to optical thickness from the OCO-2/3 observations *themselves*, but even if this were always successful, these retrievals would only be available for OCO-2 footprint locations, and not for the wider scene context. In this study, we therefore assume a fixed geometric thickness for clouds, specifically 1 km for cloud top heights smaller than 4 km, and a cloud base height of 3 km for cloud top heights greater than 4 km. This is a starting point for illustration of the general approach. The subjective assumption can be changed once more detailed cloud information does become available.

We believe that your opinion is partially correct. A fraction of the enhanced surface illumination causing the observed perturbations does indeed stem from multiple reflections between the cloud base and the surface, especially when the cloud fraction is very low. However, the primary path of enhanced illumination does stem from multiple scattering, when considering a clear-sky patch in the *vicinity* of a cloud. For the alternate path suggested by the review (which does factor into the extra illumination to some degree), the photon path length distribution is much more comparable to direct illumination than the radiation traveling through the cloud because even multiple reflections between the cloud (base) and the surface do not enhance the photon path nearly as much as multiple scattering in an extended cloud layer. That said, the photon path length distribution change mediated by the cloud itself does depend on the thickness of the cloud, so our approximation of a fixed geometric thickness is not ideal in either event. We will add a statement to this effect in the revised paper.

Inspired by your comment, we decided to look into this issue a little deeper than we initially had. We performed a simulation that reduced the geometric cloud thickness to 0.5 km for low clouds, as shown in Table R1. Compared to the simulation of a geometric cloud thickness of 1 km in Table R2 (same as Table 2 in the manuscript), the amplitude of slope ($a_s$) decreases notably when clouds have smaller geometric thickness. This indicates that the cloud base height or geometric cloud thickness could impact the 3D effect magnitude for the same COT. It is, however, unclear whether this is caused by reflections between the surface and the cloud, or simply by changes of the multiple scattering photon paths *inside* the cloud, which we believe are the more important pathway. The relative importance of the multi-scattering within clouds and multiple reflections between the cloud base and the surface needs further studies. The good news is that with the advent of Oxygen A-Band measurements from space, cloud geometric thickness in addition to cloud top height should become more widely available than it is now.

Table R1. Amplitude and e-folding distances for $s$ and $i$ fittings of the simulation with a homogeneous aerosol layer in the $O_2$-A, $WCO_2$, and $SCO_2$ bands for 0.5 km geometric cloud thickness of low clouds.

| | Slope | | | Intercept | | |
|---|---|---|---|---|---|---|
| | $s_{O_2-A}$ | $s_{WCO_2}$ | $s_{SCO_2}$ | $i_{O_2-A}$ | $i_{WCO_2}$ | $i_{SCO_2}$ |

| | | | | | | |
|---|---|---|---|---|---|---|
| $a_s$ or $a_i$ | 0.214 ± 0.059 | 0.094 ± 0.032 | 0.150 ± 0.052 | 1.033 ± 0.349 | 1.069 ± 0.405 | 0.811 ± 0.383 |
| $d_s$ or $d_i$ (km) | 7.13 ± 1.34 | 6.86 ± 1.52 | 6.31 ± 1.41 | 2.83 ± 0.32 | 2.65 ± 0.39 | 2.49 ± 0.45 |

Table R2. The same table as Table 2 in the manuscript. Amplitude and e-folding distances for **s** and **i** fittings of the simulation with a homogeneous aerosol layer in the $O_2$-A, $WCO_2$, and $SCO_2$ bands for 1.0 km geometric cloud thickness of low clouds.

| | Slope | | | Intercept | | |
|---|---|---|---|---|---|---|
| | $s_{O_2-A}$ | $s_{WCO_2}$ | $s_{SCO_2}$ | $i_{O_2-A}$ | $i_{WCO_2}$ | $i_{SCO_2}$ |
| $a_s$ or $a_i$ | 0.457 ± 0.094 | 0.123 ± 0.037 | 0.250 ± 0.041 | 0.755 ± 0.327 | 0.648 ± 0.227 | 0.847 ± 0.406 |
| $d_s$ or $d_i$ (km) | 3.82 ± 0.44 | 5.04 ± 0.89 | 4.58 ± 0.78 | 2.69 ± 0.32 | 2.91 ± 0.31 | 2.35 ± 0.33 |

We added the following statement in Section 4.2.3 of the manuscript to clarify our views (we may shorten the exact wording, depending on your feedback) and move the simulation setting to the appendix:

- Added text (Lines 1439-1466 in the revised manuscript):

"By necessity, this study assumes fixed cloud geometric thickness (1 km for cloud top height smaller than 4 km and cloud base at 3 km for cloud top height greater than 4 km). The additional photon path caused by multiple scattering within clouds influences the magnitude of the 3D cloud effect, so the slopes are sensitive to the choice of geometric cloud thickness. Unfortunately, this parameter is not readily available from operational products. Some attempts are being made to exploit the $O_2$-A channel of OCO-2 (Zinner et al., 2019; Li et al, 2024). Once these are mature, the information will be used by our algorithm. Generally, since the vertical cloud properties can influence the magnitude and distribution of the 3D cloud effect, further investigation of the impact of cloud properties, including COT, CTH, and cloud base height, on the 3D cloud effect is recommended for future research."

**Comment 06:** For the parameterization, it might be beneficial to have a generalized distance that doesn't just take into account horizontal distance but rather the 3D distance of a surface point from cloud base (or side). For isotropic scatterers, the downwelling flux impinging on a point on the surface would scale with the inverse square of this distance, so square distance weighting seems like a good choice as used in the study.

**Response 06:** Thanks for the suggestion. We have considered incorporating vertical distance into our parameterization; however, the lack of reliable cloud base height data poses a challenge. The variation in slope and intercept parameters at the same effective cloud distance may arise from differences in vertical heights, which we currently cannot accurately quantify due to this data limitation. We will explore this approach further when more comprehensive cloud height data becomes available. That said, it remains to be resolved whether the most significant effect is cloud-surface reflections (as suggested by you) or by multiple scattering within the cloud itself (as favored by us). Most likely, it is a combination. If multiple scattering were the sole source, the vertical dimension should not matter as much in future parameterization as the horizontal. Of course, this will depend on other factors as well, for example the solar zenith angle and the surface reflectance.

**Comment 07:** Along with that, not all clouds are equally bright, and their 3D enhancement should increase with overall cloud brightness. It might be useful to have a generalized distance that includes weighting by cloud reflectance. This might help the parameterization/bypass approach generalize more effectively.

**Response 07:** This is an interesting thought. We appreciate this idea and agree that the parameterization can be done better by considering more factors, such as cloud brightness. It remains to be seen, however, whether the enhanced surface illumination stemming from clouds does scale with brightness. Our impression is that what matters more is the general cloud context (i.e., the geometric distribution), but this is, again, somewhat subjective and will be studied more systematically in the future. Using cloud brightness as one of the input parameters for future bypass methods will be a good line of investigation. Our approach is merely an initial attempt to bypass 3D-RT.

**Comment 08:** The vertical distribution of aerosol will also influence the distance scaling of the slope and intercept parameters. Currently, the study examines aerosol within the cloud layer and states that it localizes enhancements to regions closer to cloud due to reduced free paths. The effect of an elevated aerosol layer may differ. Higher-altitude scattering layers tend to increase the horizontal distance over which 'adjacency' effects occur (Minomura et al., 2001). I think it would be worthwhile to discuss the role of the vertical distribution of the aerosol.

**Response 08:** Thank you for this insightful comment. In response, we have provided an additional simulation with the aerosol top layer set at 2 km, which is lower than the case discussed in Section 5.3 (3.1 km), while keeping other conditions consistent with the setup in Section 5.3. Table R3 presents the resulting slope and intercept values. Compared to Table R1 (above) and Table 1 in the manuscript (which did not include surface aerosols), $a_s$ decreases and $d_s$ across all three bands. This suggests that the relative height between the aerosol top layer and cloud base and top heights can influence slope parameterization.

Table R3. Amplitude and e-folding distances for *s* and *i* fittings of the simulation with a homogeneous aerosol layer in the $O_2$-A, $WCO_2$, and $SCO_2$ bands for the top of aerosol layer at 2 km. Consider this table relative to Table R1.

| | Slope | | | Intercept | | |
|---|---|---|---|---|---|---|
| | $s_{O_2-A}$ | $s_{WCO_2}$ | $s_{SCO_2}$ | $i_{O_2-A}$ | $i_{WCO_2}$ | $i_{SCO_2}$ |
| $a_s$ or $a_i$ | $0.106 \pm 0.051$ | $0.072 \pm 0.022$ | $0.114 \pm 0.040$ | $0.933 \pm 0.424$ | $0.531 \pm 0.192$ | $0.574 \pm 0.316$ |
| $d_s$ or $d_i$ (km) | $8.90 \pm 2.72$ | $8.14 \pm 1.61$ | $7.76 \pm 1.68$ | $2.43 \pm 0.38$ | $3.03 \pm 0.42$ | $2.48 \pm 0.48$ |

We added a paragraph in Section 5.3 discussing the potential impact of aerosol vertical distribution on the 3D cloud effect:

●    Added text (Lines 708-715 in the revised manuscript):

"Currently, we assume an even distribution of aerosols in the boundary layer in our radiance simulations. However, as Minomura et al. (2001) demonstrate, the effect of aerosols on radiance scattering can vary significantly depending on vertical distribution—particularly when surface albedo differences are pronounced, or the aerosol layer is low. In contrast, elevated aerosol layers can extend the horizontal range of adjacency effects, potentially altering the scaling of slope and intercept parameters. This is also applicable to spectroscopy. Consequently, non-uniform vertical aerosol distributions or uncertainties in boundary layer height could introduce variability in evaluating 3D cloud effects. Vertical aerosol and cloud distribution information, such

as the data from CALIPSO on the A-train, could be beneficial for improving the accuracy of simulations, but they are not implemented in the initial software release. ”

**Comment 09:** The issues of cloud base height and aerosol don't seem insurmountable at least for measurements acquired in vicinity of A-train sensors. I think it would be beneficial to provide a sketch of how these additional measurements can be used to constrain these other factors and develop an operational parameterization.

**Response 09:** Indeed! As stated above, CALIPSO could be used to constrain the vertical distribution of aerosols, whereas the oxygen A-Band on OCO-2 itself could be used to estimate geometric cloud thickness, the two greatest 'caveats' of this study that you pointed out, and that we fully acknowledge. However, this is only the initial software release and the first associated paper. Reconstructing the real cloud, aerosol, and surface fields at the level of detail suggested by you and other reviewers have been beyond the scope of this initial step. It is also important to note that LIDAR data are not always available for every satellite track; for instance, there was no CALIPSO data available for the same track in the case examined in this study.

We added the following text in Section 5.3 (with the previous response):

- Added text (Lines 714-715 in the revised manuscript):

"Vertical aerosol and cloud distribution information, such as the data from CALIPSO on the A-train and from the Oxygen A-Band of OCO-2 will be considered in future releases of the algorithm. They are expected to improve the predictions of the slope and intercept parameters, and thus make the mitigation of cloud vicinity effects more accurate and powerful."

References:

Schmidt, K. S., Massie, S., and Feingold, G., 2019, June. Impact of Broken Clouds on Trace Gas Spectroscopy From Low Earth Orbit. In Hyperspectral Imaging and Sounding of the Environment (Optica Publishing Group, 2019), paper HW5C-2.

Emde, C., Yu, H., Kylling, A., van Roozendael, M., Stebel, K., Veihelmann, B., and Mayer, B.: Impact of 3D cloud structures on the atmospheric trace gas products from UV–Vis sounders – Part 1: Synthetic dataset for validation of trace gas retrieval algorithms, Atmos. Meas. Tech., 15, 1587–1608, https://doi.org/10.5194/amt-15-1587-2022, 2022.

Partain, P. T., A. K. Heidinger, and G. L. Stephens (2000), High spectral resolution atmospheric radiative transfer: Application of the equivalence theorem, J. Geophys. Res., 105(D2), 21632177, doi:10.1029/1999JD900328.

Doicu, A.; Efremenko, D.S.; Trautmann, T. A Spectral Acceleration Approach for the Spherical Harmonics Discrete Ordinate Method. Remote Sens. 2020, 12, 3703. https://doi.org/10.3390/rs12223703

Minomura, Mitsuo, Hiroaki Kuze, and Nobuo Takeuchi. "Adjacency Effect in the Atmospheric Correction of Satellite Remote Sensing Data: Evaluation of the Influence of Aerosol Extinction Profiles." Optical Review 8, no. 2 (March 1, 2001): 133–41. https://doi.org/10.1007/s10043-0010133-2.